# PolyGraph Discrepancy: A Classifier-Based Metric for Graph Generation

**Markus Krimmel**[*]    **Philip Hartout**[*]    **Karsten Borgwardt**    **Dexiong Chen**

Max Planck Institute of Biochemistry, Martinsried, Germany

`{krimmel,hartout,borgwardt,dchen}@biochem.mpg.de`

[*]Equal contribution

## Abstract

Existing methods for evaluating graph generative models primarily rely on Maximum Mean Discrepancy (MMD) metrics based on graph descriptors. While these metrics can rank generative models, they do not provide an absolute measure of performance. Their values are also highly sensitive to extrinsic parameters, namely kernel and descriptor parametrization, making them incomparable across different graph descriptors. We introduce PolyGraph Discrepancy (PGD), a new evaluation framework that addresses these limitations. It approximates the Jensen-Shannon (JS) distance of graph distributions by fitting binary classifiers to distinguish between real and generated graphs, featurized by these descriptors. The data log-likelihood of these classifiers approximates a variational lower bound on the JS distance between the two distributions. Resulting scores are constrained to the unit interval $[0, 1]$ and are comparable across different graph descriptors. We further derive a theoretically grounded summary score that combines these individual metrics to provide a maximally tight lower bound on the distance for the given descriptors. Thorough experiments demonstrate that PGD provides a more robust and insightful evaluation compared to MMD metrics. A reference implementation of PGD is available on GitHub at  `BorgwardtLab/polygraph-benchmark`.

## 1 Introduction

Graph generative models (GGMs) are seeing wider adoption across scientific domains, from retrosynthesis (Somnath et al., 2021) and social network modeling (Bojchevski et al., 2018) to the discovery of novel drugs and materials (Liu et al., 2024; Kelvinius et al., 2025). However, progress in this field is increasingly bottlenecked by the lack of robust methods for evaluating generated graphs (Thompson et al., 2022; O'Bray et al., 2022).

This evaluation challenge is not unique to graphs. In image generation, the community has largely converged on pretrained embeddings paired with distribution distances, such as Inception-v3 coupled with Fréchet distance yielding the widely used Fréchet Inception distance (FID) (Heusel et al., 2017), or DinoV2 and density estimation producing the Feature Likelihood Divergence (FLD) (Jiralerspong et al., 2023). While these approaches provide standardized metrics adapted to other fields such as materials (Kelvinius et al., 2025), video (Unterthiner et al., 2019), and audio (Kilgour et al., 2019), limitations remain (Barratt & Sharma, 2018). As an alternative, the classifier two-sample test (C2ST) (Lopez-Paz & Oquab, 2017) recasts evaluation as a supervised classification task, turning classifier performance into evaluation metrics. To date, the applicability of these approaches to graph-structured data has not yet been explored.

The *de facto* standard for evaluating GGMs is to compute the Maximum Mean Discrepancy (MMD) (Borgwardt et al., 2006; Gretton et al., 2006; 2012) between distributions of hand-crafted graph descriptors (e.g., degrees, Laplacian spectra, etc.) on a small set of synthetic and real-world graphs (You et al., 2018). While convenient, this approach has critical limitations. The MMD value depends on the choice of kernel and descriptor, so (i) a single reported value without context carries no absolute notion of the goodness of fit of the generative model, and (ii) rankings are *sensitive to descriptor and kernel parameter choice* (O'Bray et al., 2022), with no way of obtaining a single

Table 1: Comparison of Maximum Mean Discrepancy and PolyGraph Discrepancy. Normalized kernels bound $MMD^2$ to $[0, 2]$, but this is seldom adopted in GGM evaluation. Regardless, the interpretation of a specific value remains kernel-dependent even within this bounded range. Note that while MMD values can be computed for different descriptors, the resulting scores are not comparable across kernels or feature spaces.

| Property | MMD | PGD |
|---|---|---|
| Range | $[0, \infty)$ | $[0, 1]$ |
| Intrinsic Scale | ✗ | ✓ |
| Descriptor Comparison | ✗ | ✓ |
| Single Ranking | ✗ | ✓ |

ranking across descriptors for consistent and systematic model comparison. These issues persist regardless of how much data is available. In practice, rigorous GGM evaluation is further hampered by (iii) the small sample sizes typical of current GGM benchmarks (20–40 test graphs), which introduce *substantial bias and variance* into MMD estimates (Krimmel et al., 2026).

We introduce PolyGraph Discrepancy (PGD), a novel evaluation framework that estimates the Jensen-Shannon distance (JSD) (Endres & Schindelin, 2003) between true and generated graph distributions using *probabilistic classification* instead of kernel-based distances. A discriminator is trained to distinguish real from generated graphs using standard graph descriptors, where the classifier's data log-likelihood provides a lower bound on the JSD. This yields scores in $[0, 1]$ that are directly *comparable across descriptors*. Taking the maximum over descriptors gives the tightest available bound while identifying the most informative descriptor. Our formulation of PGD uses TabPFN (Hollmann et al., 2025), a fast, hyperparameter-free discriminator, making it robust and simple to use. Empirically, we show that PGD monotonically tracks synthetic data perturbations, strongly correlates with model training progress, and accurately captures generated graph quality. It also produces robust rankings across representative GGMs. Table 1 summarizes the advantages of PGD over MMD.

Our work makes four primary contributions:

- **A rigorous reassessment of MMD for GGM evaluation.** We empirically show that standard MMD estimators are plagued by high bias and variance at typical benchmark sizes (20-40 graphs), leading to unreliable model rankings, and we provide actionable remedies.
- **PolyGraph Discrepancy (PGD): an estimate of the JSD distance between distributions.** We propose a method to derive interpretable evaluation scores by approximating variational lower bounds on the JSD via probabilistic discrimination on graph descriptors.
- **A comprehensive empirical validation.** We show that PGD tracks data perturbations monotonically and correlates strongly with training dynamics of state-of-the-art models. We also provide comprehensive PGD-based benchmark results across synthetic and real-world graphs, including molecules.
- **An open-source library to advance GGM evaluation.** We release the `PolyGraph` library, including implementations of PGD, MMD estimators, and new, larger benchmark datasets (SBM-L, LOBSTER-L, PLANAR-L), to facilitate more robust and reproducible future research.

## 2 RELATED WORK

Here we present related work on the evaluation of graph generative models and classifier-based evaluation for general generative models.

**Evaluation of Graph Generative Models.** The evaluation of GGMs has largely been shaped by methods based on the MMD (Gretton et al., 2012). You et al. (2018) first proposed computing the MMD between generated and real graph distributions using a Wasserstein Gaussian kernel on a set of graph descriptors, including degree histograms, clustering coefficients, and orbit counts. To reduce the computational cost of this method, Liao et al. (2019) introduced a simpler kernel formulation using a Gaussian kernel with the squared total variation distance, which gained widespread adoption (Martinkus et al., 2022; Vignac et al., 2023; Chen et al., 2025). However, this simplified kernel was shown to be indefinite and highly sensitive to hyperparameter choices (O'Bray et al.,

2022). Subsequent work has focused on correcting these flaws, either by modifying the kernel to ensure it is positive definite (O'Bray et al., 2022) or by employing standard RBF kernels with automated hyperparameter tuning (Thompson et al., 2022; Sriperumbudur et al., 2009). A parallel research effort has concentrated on identifying more expressive graph descriptors for use within the MMD framework. The initial set of statistics was augmented with the graph Laplacian spectrum by Liao et al. (2019), and more recently, graph neural networks (GNNs) have been used as powerful graph featurizers (Thompson et al., 2022; Shirzad et al., 2022). These improvements address the reliability of MMD estimation, but a more fundamental limitation remains: *since MMD has no inherent, kernel-independent scale, it is difficult to assess whether newly proposed descriptors are better suited for discriminating between real and generated graphs.* PGD, however, is comparable across descriptors and thus explicitly quantifies their discriminative power.

Departing from MMD, other evaluation paradigms have been proposed. Southern et al. (2023) used tools from topological data analysis, featurizing graphs via persistent homology and comparing distributions based on their average persistence landscapes. In a different direction, Martinkus et al. (2022) introduced synthetic benchmark datasets (Planar and SBM) that allow for judging the structural validity of individual graph samples–such as planarity. The small size of the synthetic datasets and the resulting variance in MMD estimates were criticized by Krimmel et al. (2026). *We expand on these observations and propose concrete techniques for quantifying the uncertainty in GGM evaluation metrics, addressing a critical need for more reliable and reproducible evaluations.*

**Classifier-Based Evaluation.** One relevant family of metrics used for generative model evaluation is derived from the classifier two-sample test (C2ST) (Lopez-Paz & Oquab, 2017). This work proposes to discriminate generated from reference samples via binary classification and repurpose the resulting accuracy as a measure for the separability of the generated and reference distributions. By extension, it assesses the quality of the generative model.

The MMD can also be viewed through this lens, as it corresponds to the optimal linear risk of a kernel classifier (Sriperumbudur et al., 2009; Gretton & Jitkrittum, 2016). Generative adversarial networks (GANs) (Goodfellow et al., 2014; Li et al., 2015; Bińkowski et al., 2018; Arjovsky et al., 2017) also leverage a classifier's output, not just for training but also for evaluation, where classifier-based divergences (including MMD) have been shown to correlate well with the perceptual quality of generated images (Im et al., 2018).

Despite the success of these methods in other domains, their application to graph generation has been limited. While some work has used fixed multi-class classifiers on generated graphs to measure performance (Liu et al., 2019), classifiers that discriminate between real and generated graphs have not been explored beyond the MMD framework. *Our work addresses this gap, proposing a novel classifier-based evaluation framework for GGMs that provides scores that are (i) absolute, (ii) comparable across different graph descriptors, and (iii) capable of estimating lower bounds on certain probability metrics.*

## 3 Preliminaries

In this section, we review two divergences, MMD and the Jensen-Shannon (JS) divergence, from a unified variational perspective: the optimal performance of a discriminator tasked with distinguishing between two distributions. We first discuss MMD, interpreting it as the linear risk of a classifier in a reproducing kernel Hilbert space (RKHS) (Sriperumbudur et al., 2009). We highlight its limitations in the context of graph generation, primarily its lack of an absolute scale, which motivates our subsequent review of the JS distance as a foundation for more interpretable, classifier-based evaluation metrics such as the PolyGraph Discrepancy.

### 3.1 MMD and its Interpretation as Classification Risk

Given two probability distributions $P$ and $Q$ over a space $\mathcal{X}$ (in our case, the space of graphs) and a kernel $k : \mathcal{X} \times \mathcal{X} \to \mathbb{R}$, the squared MMD is defined as:

$$\text{MMD}^2(P, Q, k) := \mathbb{E}_{x,x' \sim P}[k(x,x')] - 2\mathbb{E}_{x \sim P, y \sim Q}[k(x,y)] + \mathbb{E}_{y,y' \sim Q}[k(y,y')]. \qquad (1)$$

The MMD can be expressed as the distance between the mean embeddings of $P$ and $Q$ in the RKHS $\mathcal{H}$ induced by $k$. This framing leads to a variational formulation where the MMD is precisely the

optimal linear classification risk achievable by a discriminator in the unit ball of $\mathcal{H}$ (Sriperumbudur et al., 2009). We refer to Appendix D for a detailed derivation.

A fundamental limitation of MMD for model evaluation is its lack of an absolute scale (O'Bray et al., 2022). Even within a single kernel family such as the RBF kernel, changing the bandwidth $\sigma$ changes the resulting MMD for the same pair of distributions, so no single value carries a kernel-independent meaning. In practice, this sensitivity is amplified by the choice of input features: with a linear kernel, for instance, simply rescaling the features rescales the MMD by the same factor. As a result, comparing MMD scores across different graph descriptors is not meaningful, and while MMD can rank models for a fixed descriptor, it provides no absolute measure of performance.

To overcome this, we turn to metrics that possess a fixed intrinsic scale, making them comparable across different graph descriptors. This leads us to the Jensen-Shannon divergence and, more generally, to the family of $f$-divergences.

### 3.2 Variational Estimation of the Jensen-Shannon Distance

The Jensen-Shannon (JS) divergence is a symmetrized version of the Kullback-Leibler (KL) divergence: $\frac{1}{2}(D_{KL}(P \parallel M) + D_{KL}(Q \parallel M))$ with $M := \frac{1}{2}(P + Q)$ being the mixture of $P$ and $Q$. It is constrained to the unit interval $[0, 1]$ and, in contrast to MMD, is independent of extrinsic parameters such as kernel choice. As extensively leveraged in GANs (Goodfellow et al., 2014), the JS divergence admits (under mild conditions) a variational formulation as the maximal data log-likelihood (up to constants) achievable by a binary classifier $D$ distinguishing between samples from $P$ and $Q$:

$$D_{JS}(P \parallel Q) = \sup_{D:\mathcal{X}\to[0,1]} \frac{1}{2}\mathbb{E}_{x\sim P}[\log_2 D(x)] + \frac{1}{2}\mathbb{E}_{x\sim Q}[\log_2(1 - D(x))] + 1. \qquad (2)$$

Importantly, the log-likelihood of *any* classifier provides a valid lower bound on the JS divergence and the bound is tightened by fitting a classifier via maximum likelihood methods. While the JS divergence is not a metric, its square root (termed the JS distance) is (Endres & Schindelin, 2003).

The JS divergence belongs to the larger family of $f$-divergences. As shown by Nguyen et al. (2010), any $f$-divergence admits a variational formulation similar to Eqn. (2). In Appendix E, we investigate the total variation (TV) distance as a possible alternative to the JS distance. Instead of log-likelihood, we show that the variational objective of the TV distance is given by the classifier's *informedness*.

## 4 PolyGraph Discrepancy: Variational Estimates of the JS Distance

Building on the variational view of divergences, we introduce PolyGraph Discrepancy (PGD), a framework for evaluating GGMs. PGD estimates the JS distance between a distribution of reference graphs and a distribution of generated graphs. The core idea is to reframe the divergence estimation as a classification task: we featurize graphs using a variety of established graph descriptors and measure how well a powerful, non-parametric classifier can distinguish between the two sets. The resulting classifier performance, measured in terms of log-likelihood, serves as a tight, empirical lower bound on the true JS divergence between the underlying graph distributions. Fig. 1 shows this procedure.

Our method proceeds in two main stages. First, we detail how to estimate a lower bound on the divergence using a *single* graph descriptor in Section 4.1. Second, we describe in Section 4.2 how to systematically combine *multiple* descriptors from a larger set to compute the final PGD, which represents the tightest lower bound from the given descriptors. We provide pseudocode in Appendix B.

### 4.1 Estimating the JS Distance with a Single Descriptor

Given a multiset of reference graphs $P_{ref}$ and generated graphs $Q_{gen}$, along with a single graph descriptor $d : \mathcal{X} \to \mathbb{R}^n$, we estimate the divergence of $P_{ref}$ and $Q_{gen}$ via featurization by $d$.

To prevent overfitting, where a classifier might perfectly memorize the training data and thus overestimate the true divergence, we randomly partition both $P_{ref}$ and $Q_{gen}$ into disjoint `fit` and `test`

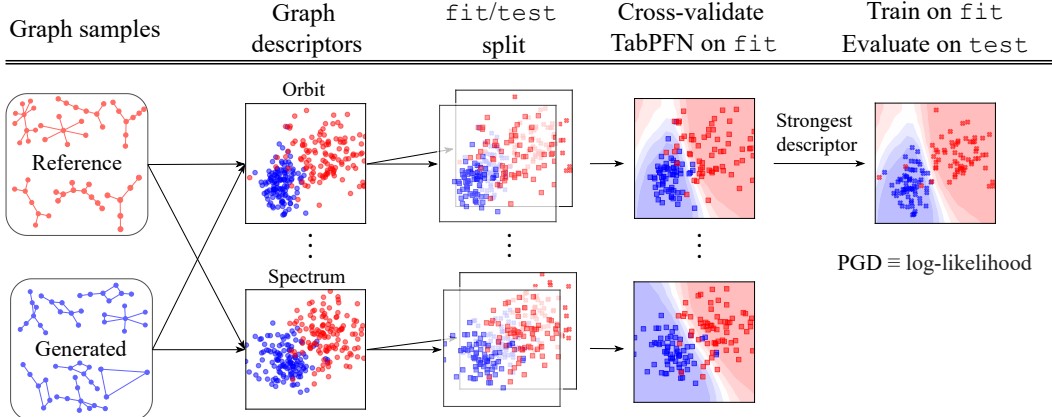

Figure 1: Computation of the PGD metric. TabPFN is trained to discriminate between generated and reference graphs based on different vectorial descriptions. The most expressive descriptor (here: orbit) is used to derive the final PGD, yielding a maximally tight lower bound on the Jensen-Shannon (JS) distance between the generated and reference graph distributions.

sets of equal size. Our goal is to approximate the supremum in Eqn. (2) by training a discriminator exclusively on the `fit` set, and computing the final divergence estimate on the held-out `test` set.

**Discriminator Choice.** An appropriate discriminator for this task must satisfy three criteria:

1. **Probabilistic:** It must output class probabilities to estimate the JS divergence via its log-likelihood objective.
2. **Efficient:** It must be fast to train, enabling rapid evaluation across many descriptors.
3. **Hyperparameter-Free:** It should be robust and require no manual tuning to ensure fair and reproducible comparisons.

These requirements rule out the training of deep neural networks with gradient descent, because it is computationally expensive and requires hyperparameter tuning. It also rules out non-probabilistic models such as decision trees and SVMs. As a result, we choose TabPFN v2.5 (Hollmann et al., 2025) in this work. TabPFN is a transformer-based model that approximates Bayesian inference over a large space of simple models consisting of Bayesian neural networks and structural causal models. It is fast (see Table 15), requires no hyperparameter tuning, and has proven to be a powerful classifier for tabular data, making it an ideal choice for our framework since our classifier operates on graph descriptors. In Appendix J, we investigate logistic regression as an alternative choice and show that TabPFN yields tighter bounds in practice. In Appendix R we show that kernel logistic regression also fits naturally into the PGD framework, allowing for the use of, *e.g.*, graph kernels (Borgwardt & Kriegel, 2005; Shervashidze et al., 2011; Grauman & Darrell, 2007). However, similar to logistic regression, we found that those kernel logistic regression-based PGD scores yielded looser bounds in practice, and elected to proceed with TabPFN.

**Estimation Procedure.** With a discriminator selected, we first apply the descriptor $d : \mathcal{X} \to \mathbb{R}^n$ to the graphs in the `fit` set to create vectorial features. We then train the binary classifier on these features using TabPFN. We apply the descriptor to the `test` set and use the trained classifier to evaluate the data log-likelihood, providing an approximate lower bound of the JS divergence. Finally, we take the square root to estimate the JS *distance*.

## 4.2 Descriptor Selection for the Tightest Bound

A single graph descriptor captures only one specific aspect of graph structure. To obtain a more comprehensive evaluation, we consider a collection of $K$ distinct descriptors $\{d_1, \ldots, d_K\}$. The goal is to identify the single descriptor that most effectively distinguishes between the reference and generated graphs, as this descriptor will yield the tightest lower bound on the true JS distance. This descriptor selection process must be performed carefully to avoid data leakage from the `test` set,

which would invalidate our final estimate. We therefore perform selection using only the `fit` data via cross-validation.

**Cross-Validation on the Fit Set.** For each descriptor $d_k : \mathcal{X} \to \mathbb{R}^n$, we estimate its ability to separate the distributions by performing 4-fold stratified cross-validation on the $(P_{\text{ref}}^{\text{fit}}, Q_{\text{gen}}^{\text{fit}})$ data. In each fold, three-quarters of the data are used for training a discriminator, and the remaining quarter is used for validation. The average validation score across the four folds provides a robust estimate of the lower bound achievable by that descriptor.

**The PolyGraph Discrepancy.** After performing cross-validation for all $K$ descriptors, we select the descriptor $d^\star$ that yielded the highest average score. This is the descriptor that is empirically the most informative. Finally, we train a new discriminator for $d^\star$ on the *entire* `fit` set and evaluate it on the held-out `test` set. The resulting score is the `PolyGraph Discrepancy` (PGD). This procedure ensures that the descriptor selection and final evaluation are performed on separate data, yielding a principled and tight estimate of the divergence between the graph distributions.

## 5 EXPERIMENTS

We empirically validate PGD through a series of experiments designed to test its robustness, sensitivity, and practical utility against standard MMD-based metrics for evaluating graph generative models. Our investigation consists of four stages:

- First, Section 5.1 shows that MMD evaluations suffer from *substantial bias and variance on current datasets*, motivating the use of larger datasets, unbiased estimators, and subsampling to assess estimate stability.
- In Section 5.2, we show that *PGD correlates well with controlled perturbations* applied to synthetic datasets, showing the power of JSD to distinguish samples from different distributions.
- In a realistic use case for a state-of-the-art diffusion model (Section 5.3), we show that *PGD reliably tracks training progress and performance gains* when increasing the number of denoising steps. Our results indicate that PGD captures model quality more reliably than MMD metrics.
- Finally, in Section 5.4 we leverage PGD to conduct a *comprehensive benchmark* of several representative GGMs.

Unless otherwise stated, all PGD scores are based on the Jensen-Shannon (JS) distance estimated using TabPFN as the discriminator. Following previous works (You et al., 2018; Liao et al., 2019; Thompson et al., 2022), we use degree histograms (abbreviated as Degree/Deg. in our tables and figures), clustering coefficient histograms (Clustering/Clust.), the Laplacian spectrum (Spectral/Spec.), orbit counts (Orbit), and GIN embeddings (GIN) as descriptors. For molecular graphs, we use domain-specific descriptors based on topological properties, physico-chemical parameters, and learned representations. We refer to Appendix C for further details.

### 5.1 HIGH BIAS AND VARIANCE PLAGUE MMD-BASED GGM BENCHMARKS

The evaluation of GGMs is predominantly conducted on synthetic, procedurally generated datasets, including lobster graphs, stochastic block models (SBMs), and planar graphs, which permit the generation of arbitrarily large numbers of samples. Krimmel et al. (2026) first raised the issue that MMD values computed on such datasets can exhibit considerable variance, thereby casting doubt on the robustness of model rankings derived from these metrics. In order to more rigorously characterize this phenomenon, we exploited the procedural nature of these datasets to systematically vary the subsample sizes used in MMD. The MMD shown here is obtained with the radial basis function (RBF) kernel; more details are given in Appendix G.

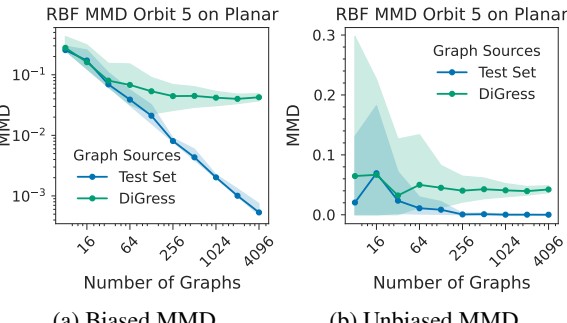

(a) Biased MMD.    (b) Unbiased MMD.

Figure 2: Examples of MMD estimates that suffer from high bias (left) and variance (right).

In the regime of commonly used synthetic graph benchmarks (between 20 and 40 test graphs, cf. Appendix P), bias dominates the MMD values (Figure 2a, in log scale for clarity). Even when using the unbiased MMD estimator[1], the variance across subsamples remains large enough to make model comparisons at these sample sizes unreliable (Figure 2b). Figure 2 illustrates these issues for DiGress-generated samples for planar graphs described with orbit counts, but extensive experiments in Appendix G show that they persist across all combinations of models, descriptors, and datasets.

This finding yields three actionable insights. First, prefer *unbiased MMD* estimates, as bias depends heavily on sample size. Second, akin to Krimmel et al. (2026), use *larger sample sizes* to reduce estimator variance; we propose SBM-L, Planar-L, and Lobster-L for this purpose (with more details in Appendix M)[2]. Third, report the *variance* of MMD across subsamples to quantify the stability of the estimates. To assess the effect of dataset size on PGD, we conducted analogous experiments in Appendix L, which show that its mean and variance stabilize beyond subsample sizes of about 256. This is particularly relevant because TabPFN's discriminative power may depend on sample size.

## 5.2 PolyGraph Discrepancy Tracks Synthetic Data Perturbations

To validate PGD as a reliable metric, we verify its ability to correlate with the magnitude of perturbations applied to graph datasets, a standard procedure for evaluating graph metrics (O'Bray et al., 2022; Thompson et al., 2022). Our experiments demonstrate that PGD effectively tracks these changes, performing on par with MMDs.

**Experimental Setup.** We conduct our experiments on five datasets: Protein contact graphs (Dobson & Doig, 2003), ego nets extracted from Citeseer (Sen et al., 2008), and three procedural datasets (Planar, SBM, Lobster). Each procedural dataset contains 4096 samples, while the proteins dataset contains 918 samples, and the ego dataset contains 757 samples. Dataset details are in Appendix P.

To simulate data corruption, we apply five distinct perturbation types, four of which are adapted from previous studies (O'Bray et al., 2022; Thompson et al., 2022). Each perturbation modifies the graph structure (or dataset) in a controlled manner. Edge deletion/addition removes or adds a specified number of edges selected at random. Edge rewiring replaces one of the incident vertices of some edges with a randomly selected vertex. Mixing operates on the dataset level by replacing a fraction of the graphs within a dataset with new samples from an Erdős–Rényi model. Finally, we propose a novel perturbation type which we term "edge swapping". Edge swapping selects pairs of edges and swaps two of their incident vertices. This transformation preserves the vertex degrees, making it a more challenging perturbation for some metrics to detect.

**Perturbation Experiments.** Our core experiment involves splitting each dataset into two equal subsets: one serves as a fixed reference distribution, and the other is subjected to the perturbations. We then measure the distance between the reference and the perturbed subset using PGD and MMD metrics. In contrast to MMD, whose boundedness requires normalized kernels that are seldom used in practice, PGD is naturally bounded in $[0, 1]$. This means it can saturate, or reach its maximum value, when perturbations are too large and the distributions become non-overlapping. To account for this, we first determine the perturbation magnitude at which PGD saturates (specifically, exceeds 0.95). We then apply perturbations only within this non-saturating range and compute the Spearman correlation between the metric scores and perturbation magnitudes. We visualize these correlation coefficients in Fig. 3, where each data point represents a combination of dataset, perturbation type, and metric. Our results show that PGD consistently exhibits a strong *rank* correlation with perturbation magnitude, comparable to that of MMD metrics. We note that while the degree-based and GIN-based MMD metrics struggle to detect the edge-swapping perturbation, PGD remains robust by leveraging multiple descriptors that compensate for compromised ones.

We provide more details in Appendix H, where we illustrate the behavior of PGD as a function of perturbation magnitude for all combinations of datasets and perturbations. From that analysis, we conclude that no single descriptor dominates the others across all combinations of datasets and

---

[1]Our MMD estimates are not unbiased, as we take the maximum MMD value over a set of kernel bandwidths, but we do use the unbiased MMD estimate without diagonals, see Eq. 3 in Gretton et al. (2012).

[2]AutoGraph reaches similar validity, uniqueness and novelty (VUN) scores with markedly lower loss in SBM-L than on the original SBM dataset (see Appendix N), showing reduced overfitting, which is underexplored in GGMs (Vignac et al., 2023).

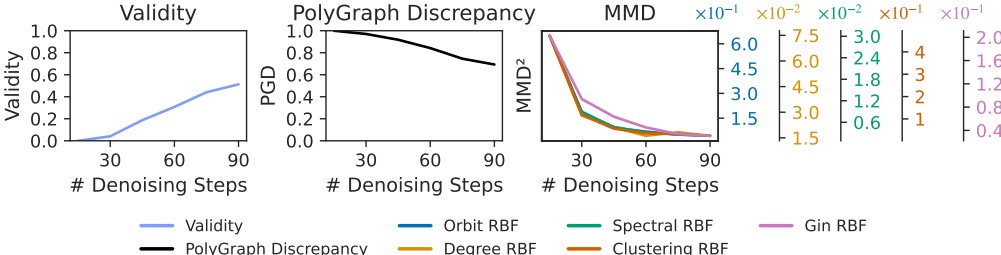

Figure 3: Spearman correlation of MMDs and PGD with magnitude of perturbation.

Figure 4: Trajectory of validity, PGD, and MMDs when increasing the number of denoising steps in DiGress on PLANAR-L.

perturbation types, underlining the necessity of considering a diverse set of graph descriptors. We present additional experiments for a PGD estimating the Total Variation distance in Appendix F.2. Appendix J provides similar results for a PGD variant using logistic regression instead of TabPFN.

### 5.3 POLYGRAPH DISCREPANCY CORRELATES WITH MODEL QUALITY

To demonstrate practical utility, we evaluate PGD on DiGress (Vignac et al., 2023), a state-of-the-art GGM, using denoising iterations and training epochs as proxies for model quality. PGD strongly correlates with both, capturing model improvement more faithfully than MMD metrics while maintaining a strong linear correlation with the percentage of valid graphs generated. All metrics were computed comparing 2048 reference graphs against 2048 generated graphs.

**Denoising Iterations.** We first analyze the impact of the number of denoising steps on sample quality. Six DiGress models are trained on the large procedural planar dataset using a range of 15 to 90 denoising steps. As shown in Fig. 4, increasing the number of steps generally improves model performance across all metrics. We find that PGD has a much stronger *linear* relationship with validity than MMD metrics, as shown by the *Pearson* correlation coefficients in Table 2. This tight relationship is especially encouraging as validity, alongside uniqueness and novelty, is often considered a gold standard metric for assessing model quality. Yet, validity is not always defined. Uniqueness and novelty can be provided jointly with PGD to offer complementary insights.

**Training Iterations.** Similarly, we assess the ability of MMD and PGD to track model quality throughout the training process on LOBSTER-L, PLANAR-L, and SBM-L. The central hypothesis is that reliable metrics should improve monotonically with training duration. We note that this relationship is non-linear, hence the use of Spearman's correlation coefficient. As illustrated for the SBM-L dataset in Fig. 5, PGD and validity align with this hypothesis, whereas MMD metrics exhibit erratic behavior. Analogous results for PLANAR-L and LOBSTER-L are provided in Appendix I. Spearman's rank correlation in Table 3 confirms this quantitatively across all datasets: both PGD and validity are strongly correlated with training duration, while MMD metrics show weak or even negative correlations. The Pearson correlations in Table 2 further show that *PGD maintains its strong linear correlation with validity* during training, a property not consistently shared by MMD metrics.

### 5.4 BENCHMARKING REPRESENTATIVE MODELS

We next present concrete PGD values and their associated subscores on a set of well-established models spanning distinct generative paradigms, including autoregressive architectures such as GRAN

Table 2: Negative Pearson correlation (↑) of validity with other distance-based metrics. Denoising refers to the experiments in which we vary the number of denoising iterations. Training refers to the experiments in which we monitor performance metrics during the training of DiGress models. Values are multiplied by 100 for readability.

|  | Dataset | PGD | Orb. RBF | Deg. RBF | Eig. RBF | Clust. RBF | GIN RBF |
|---|---|---|---|---|---|---|---|
| Denoising | PLANAR-L | **99.29** | 73.50 | 70.87 | 73.41 | 71.36 | 82.52 |
| Training | PLANAR-L | **97.65** | 84.32 | 59.11 | 76.47 | 88.31 | 68.11 |
|  | SBM-L | **92.42** | 52.80 | 5.14 | 23.32 | 85.27 | 6.32 |
|  | LOBSTER-L | 86.54 | -34.65 | -35.17 | -22.95 | **87.05** | -30.55 |

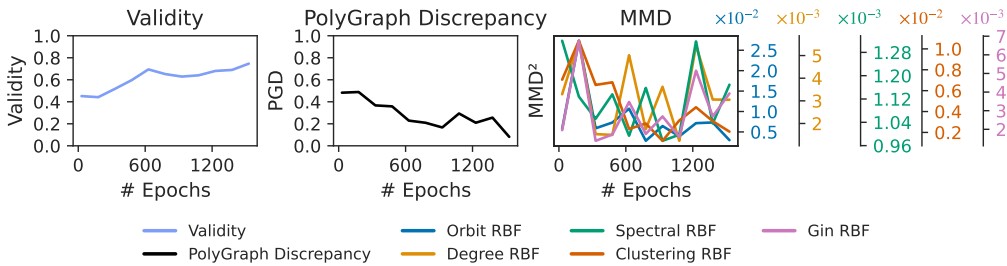

Figure 5: Trajectory of validity, PGD, and MMD metrics during training of DiGress on SBM-L.

(Liao et al., 2019) and AutoGraph (Chen et al., 2025) and diffusion models such as ESGG (Bergmeister et al., 2024) and DiGRESS (Vignac et al., 2023). We benchmark them on our proposed datasets, SBM-L, LOBSTER-L, and PLANAR-L (with 4096 samples each, see Appendix M) as well as the Proteins dataset with 92 test samples (Dobson & Doig, 2003). Additionally, we present PGD benchmarks of AutoGraph and DiGress on the molecular datasets GuacaMol (Brown et al., 2019) and MOSES (Polykovskiy et al., 2020) using 10,000 generated samples for benchmarking. For these datasets, we propose domain-specific descriptors which we describe in Appendix C.2. Appendix K contains further benchmarking methodological details.

As shown in Table 4, AutoGraph and DiGress achieve the best overall PGD scores across most datasets. PGD generally aligns with VUN or validity rankings, though some exceptions exist—ESGG ranks highest in VUN on PLANAR-L but performs worse in PGD. The Proteins dataset yields the highest scores, suggesting greater modeling difficulty; its higher standard deviations reflect the smaller test set (92 samples vs. 4096 for procedural datasets). Max-reduction proves helpful in edge cases like LOBSTER-L, where clustering coefficients are uniformly zero, preventing a single uninformative subscore from masking other structural flaws. When interpreting the final PGD score, note it can differ from individual subscores since they use different datasets. Subscores are averaged over cross-validation splits on the *training set* to select the most informative descriptor, while the final PGD is computed on the *test set*, potentially yielding different results. Appendix K compares MMD and PGD values using Gaussian TV pseudo-kernels (Table 11) and optimized RBF kernels (Tables 12 and 13). Overall, PGD yields more interpretable model rankings than MMDs.

We also consider a feature concatenation variant of PGD as an alternative to max-reduction, where we concatenate all descriptors and train a single discriminator on the combined features (Appendix Q). While this yields tighter bounds (higher JSD estimates), it prevents identifying the most informative descriptor; therefore, we recommend max-reduction in practice.

## 6 CONCLUSION

We introduce PGD, a classifier-based evaluation that yields unit-scale metrics by training a discriminator on standard graph descriptors and selecting the most informative one. Instantiated with TabPFN to estimate the JS distance, PGD is fast and tuning-free. Across perturbation and model-quality studies, PGD increases monotonically with synthetic noise and correlates strongly–and often linearly–with validity and training progress. It also produces robust rankings with descriptor-specific

Table 3: Sign-adjusted Spearman correlation (↑) of validity, PGD, and MMDs with the number of training iterations for DiGress.

| Dataset | Val. | PGD | Orb. RBF | Deg. RBF | Eig. RBF | Clust. RBF | GIN RBF |
|---|---|---|---|---|---|---|---|
| PLANAR-L | **96.36** | 93.64 | 75.45 | 68.18 | 50.00 | 86.36 | 77.27 |
| SBM-L | **82.73** | 77.27 | 23.64 | 10.91 | 10.91 | 68.18 | -24.55 |
| LOBSTER-L | **85.47** | 83.33 | -7.35 | -13.24 | 6.86 | 68.14 | 1.23 |

Table 4: Mean PGD ± standard deviation across synthetic and real-world graphs. AutoGraph* denotes a model pretrained on the PubChem dataset. More details can be found in the original paper (Chen et al., 2025). Values are multiplied by 100 for readability. Subscores are computed on the training set to select the best descriptor, and the final PGD refers to the score computed on the test set with the best descriptor. All results have been obtained with TabPFN 2.5.

| Dataset | Model | | | PGD subscores | | | | | |
|---|---|---|---|---|---|---|---|---|---|
| | | VUN (↑) | PGD (↓) | Clust. (↓) | Deg. (↓) | GIN (↓) | Orb5. (↓) | Orb4. (↓) | Eig. (↓) |
| PLANAR-L | AutoGraph | 85.7 | 34.1 ± 1.7 | **10.2** ± 2.7 | **8.7** ± 1.9 | **9.8** ± 1.6 | **34.1** ± 1.7 | **31.6** ± 0.9 | **28.7** ± 1.8 |
| | DIGRESS | 79.6 | 46.7 ± 1.7 | 28.3 ± 2.3 | 25.4 ± 2.0 | 29.8 ± 2.0 | 46.7 ± 1.7 | 43.0 ± 1.3 | 39.9 ± 1.9 |
| | GRAN | 3.1 | 99.5 ± 0.5 | 99.3 ± 0.4 | 98.1 ± 0.8 | 97.9 ± 0.7 | 99.4 ± 0.9 | 98.9 ± 0.9 | 99.0 ± 0.9 |
| | ESGG | **93.9** | 47.7 ± 1.1 | 16.4 ± 3.0 | 26.3 ± 1.2 | 33.1 ± 2.2 | 47.7 ± 1.1 | 45.3 ± 1.6 | 30.9 ± 1.8 |
| LOBSTER-L | AutoGraph | 82.6 | 16.6 ± 1.4 | 3.7 ± 2.5 | 11.0 ± 1.5 | 13.4 ± 1.4 | 16.6 ± 1.4 | 14.8 ± 1.8 | 12.4 ± 1.6 |
| | DIGRESS | **91.8** | **4.0** ± 3.1 | 2.3 ± 0.6 | **1.4** ± 1.7 | **3.5** ± 1.1 | **5.8** ± 2.2 | **5.9** ± 1.8 | **1.0** ± 1.6 |
| | GRAN | 42.9 | 86.0 ± 0.7 | 21.3 ± 0.6 | 78.2 ± 0.6 | 83.0 ± 0.5 | 85.9 ± 0.7 | 85.6 ± 0.7 | 71.0 ± 1.2 |
| | ESGG | 70.9 | 70.6 ± 0.6 | **0.0** ± 0.0 | 64.0 ± 0.9 | 67.4 ± 0.9 | 70.6 ± 0.6 | 66.0 ± 0.7 | 56.6 ± 1.4 |
| SBM-L | AutoGraph | **87.1** | **6.8** ± 1.2 | **1.3** ± 1.4 | **5.6** ± 1.6 | **6.8** ± 1.1 | **3.3** ± 2.3 | **3.9** ± 2.8 | **3.2** ± 1.9 |
| | DIGRESS | 74.1 | 16.6 ± 1.9 | 7.0 ± 2.4 | 10.4 ± 1.8 | 14.3 ± 1.7 | 16.6 ± 1.9 | 14.7 ± 1.7 | 7.8 ± 3.0 |
| | GRAN | 22.3 | 69.1 ± 1.1 | 50.4 ± 1.7 | 58.3 ± 1.8 | 69.3 ± 1.2 | 67.4 ± 1.4 | 64.2 ± 1.2 | 60.6 ± 1.7 |
| | ESGG | 11.0 | 99.3 ± 0.3 | 97.1 ± 0.3 | 97.4 ± 0.6 | 98.6 ± 0.3 | 95.6 ± 0.3 | 90.2 ± 0.6 | 99.3 ± 0.3 |
| Proteins | AutoGraph | - | **57.3** ± 11.5 | 39.8 ± 14.7 | 26.7 ± 7.1 | 44.3 ± 7.9 | **62.1** ± 5.1 | 48.1 ± 4.9 | 50.0 ± 10.5 |
| | DIGRESS | - | 86.7 ± 2.4 | **29.0** ± 11.7 | **17.2** ± 11.1 | **13.6** ± 8.5 | 86.7 ± 2.4 | 57.3 ± 4.6 | 23.3 ± 10.0 |
| | GRAN | - | 88.9 ± 5.5 | 83.0 ± 4.3 | 68.1 ± 7.4 | 68.6 ± 7.1 | 90.7 ± 3.2 | 86.7 ± 3.6 | 64.8 ± 6.3 |
| | ESGG | - | 78.8 ± 5.0 | 58.4 ± 8.5 | 55.2 ± 6.8 | 59.3 ± 5.2 | 79.4 ± 5.3 | 72.0 ± 6.7 | 29.3 ± 12.1 |

| Dataset | Model | | | PGD subscores | | | | |
|---|---|---|---|---|---|---|---|---|
| | | Valid (↑) | PGD (↓) | Topo (↓) | Morgan (↓) | ChemNet (↓) | MolCLR (↓) | Lipinski (↓) |
| GUACAMOL | AutoGraph | 91.6 | 27.3 ± 0.5 | 10.0 ± 1.0 | 18.2 ± 0.5 | 27.3 ± 0.5 | 17.4 ± 0.5 | 22.8 ± 0.4 |
| | AutoGraph* | **95.9** | **11.5** ± 0.5 | **6.3** ± 0.4 | **5.6** ± 0.6 | **7.8** ± 0.3 | **3.6** ± 0.6 | **11.5** ± 0.5 |
| | DIGRESS | 85.2 | 37.4 ± 0.7 | 22.1 ± 0.5 | 25.3 ± 0.5 | 37.4 ± 0.7 | 24.8 ± 0.5 | 35.2 ± 0.5 |
| MOSES | AutoGraph | **87.4** | **33.5** ± 0.6 | **24.5** ± 0.5 | **24.8** ± 0.8 | **33.0** ± 0.7 | **26.1** ± 0.4 | **33.5** ± 0.6 |
| | DIGRESS | 85.7 | 37.1 ± 0.3 | 28.7 ± 0.5 | 29.5 ± 0.7 | 36.3 ± 0.5 | 30.2 ± 0.6 | 37.1 ± 0.3 |

subscores. To standardize GGM evaluation and model selection, we release the `PolyGraph` library, PGD, and the larger datasets, which we show are necessary to avoid high bias and variance observed in evaluation metrics. We discuss potential limitations in Appendix A. We hope that our work catalyzes progress in graph generation and, more broadly, enables effective evaluations of generative models where multiple combinations of possibly complementary descriptors are required.

## ETHICS STATEMENT

This work develops evaluation methods for graph generative models and does not involve human subjects, animals, or personal data. We foresee no harm to individuals, groups, or the environment.

## REPRODUCIBILITY STATEMENT

To ensure the reproducibility of all results and plots of this work, we provide the `PolyGraph` library at ⓞ BorgwardtLab/polygraph-benchmark, implementing the `PolyGraph Discrepancy`, MMD metrics, and datasets discussed here, with tests ensuring consistency with the MMD implementations of Liao et al. (2019); Thompson et al. (2022). Details on the PGD estimation procedure, graph descriptors, and procedural dataset generation are in Appendices B, C and M. All generated data, including model checkpoints and computed metrics, will be stored in a long-term private archive with a minimum guaranteed access period of ten years.

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

# Appendix

## CONTENTS

## A  LIMITATIONS

Here, we touch upon some of the limitations of this work.

**Descriptor dependence and information loss.**  PGD operates on hand-crafted descriptors rather than raw graphs. It therefore yields a lower bound of the divergence between *descriptor distributions*, which itself is a lower bound of the divergence between the *graph distributions*. If the divergence between descriptor distributions does not tightly approximate the divergence between graph distributions, the PGD is also inherently a loose bound on the divergence between graph distributions. This highlights the importance of considering expressive descriptors.

The final max-reduction can also under-utilize complementary signals across descriptors. This could be addressed by combining features prior to TabPFN fitting, and using TabPFN extensions for automatic feature selection[3].

**Sample-size dependence**  On the one hand, PGD requires several hundred samples to get an accurate metric value, as indicated in Appendix L, which requires some computational burden, especially for difficult-to-compute descriptors. On the other hand, the sample size used in our formulation of PGD is constrained by TabPFN's recommended 10k training limit, though this restriction is only an implementation detail. This might be problematic in practice if a large number of samples is required to obtain a tight bound. We recommend that users assess the variance of PGD carefully when considering new descriptors, graph types, and discriminators. The TabPFN extensions package also implements some approaches to extend the training size via subsampling and ensembling[4].

**Feature dimensionality.**  While MMD can operate on high-dimensional graph descriptors, the classifier used in PGD may be less effective on very high-dimensional features. The graph descriptors proposed in previous works (cf. Appendix C.1) are low-dimensional and well-suited for TabPFN v2.5, which does not impose feature dimensionality limits. In the context of evaluating molecule generative models, we employ random projections to map 512-dimensional graph representations to a more compact feature space (cf. Appendix C.2). A more sophisticated feature selection process may yield tighter bounds on the JS distance. We leave the exploration of optimal feature selection to future work.

**Scopes of graph types, datasets, and models.**  Our experiments focus on common procedural datasets (with specific parameters), proteins, and molecules. We do not evaluate directed, temporal, or heterogeneous graphs, and leave this to future work. While we benchmark four different GGMs, covering autoregressive and denoising diffusion paradigms, we hope that future works adopt the PGD framework to extend benchmarks to a wider variety of methods.

**Application to other domains.**  We focus on applying PGD to generative graph evaluation, where the need for rigorous assessment is particularly acute. Nonetheless, the same approach could extend to other domains, though we leave this unexplored. One promising direction is improving InceptionV3-style scoring: our multi-descriptor strategy could mitigate the sensitivity of FID to network initialization by max-reducing across multiple InceptionV3 initializations, which was shown to be problematic by Barratt & Sharma (2018).

## B  PGD PSEUDOCODE

We provide pseudocode for the computation of PGD in Algorithm 1. We note that the procedure `estimate_divergence` corresponds to the algorithm we describe in Section 4.1 while `polygraphscore` implements the combination of descriptors we outline in Section 4.2.

---

[3] https://github.com/PriorLabs/tabpfn-extensions
[4] see Footnote 3

---

**Algorithm 1** PGD computation

---

1: **procedure** estimate_divergence(train, val, mode)
2:     clf ← fit_tabpfn(train)
3:     preds ← clf.predict(val.x)
4:     **if** mode = "jsd" **then**
5:         metric ← $\sqrt{\max(\texttt{log\_likelihood}(\text{preds}, \text{val.y}), 0)}$
6:     **else**
7:         $\gamma$ ← max_info_threshold(clf.predict(train.x), train.y)
8:         metric ← informedness(preds, val.y, $\gamma$)
9:     **return** metric
10:
11: **procedure** train_test_divergence(reference, generated, descriptor, mode, $k$)
12:     ref_train, ref_test ← reference[0 :: 2], reference[1 :: 2]          ▷ Split reference graphs
13:     gen_train, gen_test ← generated[0 :: 2], generated[1 :: 2]          ▷ Split generated graphs
14:     $(X, Y)$ ← (descriptor(ref_train ‖ gen_train), $[0 \ldots 0, 1 \ldots 1]$)
15:     folds ← stratified_folds($X, Y, k$)
16:     cv_metric ← 0
17:     **for** train, val ∈ folds **do**
18:         cv_metric ← cv_metric + estimate_divergence(train, val, mode)
19:     cv_metric ← cv_metric$/k$
20:     $(X_{\text{test}}, Y_{\text{test}})$ ← (descriptor(ref_test ‖ gen_test), $[0 \ldots 0, 1 \ldots 1]$)
21:     test_metric ← estimate_divergence($(X, Y), (X_{\text{test}}, Y_{\text{test}})$, mode)
22:     **return** cv_metric, test_metric
23:
24: **procedure** polygraphdiscrepancy(ref, gen, mode)
25:     all_descriptors ← [orbit4, orbit5, deg, clust, spec, gin]
26:     all_metrics ← hash_map()
27:     **for** $d$ ∈ all_descriptors **do**
28:         all_metrics[$d$] ← train_test_divergence(ref, gen, $d$, mode, $k = 4$)
29:     best_desc ← $\arg\max_d$ all_metrics[$d$].cv_metric
30:     **return** all_metrics[best_desc].test_metric

---

# C GRAPH DESCRIPTORS

In this section, we discuss the vectorial graph descriptions used in our work. In Appendix C.1, we provide details on the descriptors we apply to the synthetic datasets (PLANAR-L, SBM-L, LOBSTER-L) and the Proteins dataset. These descriptors are, for the most part, identical to established descriptors introduced for MMD evaluations (You et al., 2018; Liao et al., 2019; Thompson et al., 2022). In Appendix C.2, we introduce novel descriptors for evaluating generative models for molecules.

We recommend that practitioners use domain-specific and expressive descriptors whenever possible, similar to our procedure for molecules in Appendix C.2. As discussed previously, one should aim to maximize the PGD metric when engineering graph descriptors.

## C.1 GENERIC DESCRIPTORS

We use graph descriptors that have previously been proposed for evaluations via Maximum Mean Discrepancy. Histograms of clustering coefficients and node degrees, as well as 4-node orbit counts, have been proposed by You et al. (2018). These descriptors were extended by Liao et al. (2019) via the spectrum of the graph Laplacian. Finally, Thompson et al. (2022) proposed to featurize graphs via randomly initialized GIN models. We extend these descriptors with 5-node orbit counts, computed with the ORCA algorithm (Hočevar, 2025). In our model benchmarks, we find that 5-node orbit counts oftentimes yield the highest PGD, hence representing a strong descriptor (cf. Table 4). However, we find in the perturbation experiments (cf. Appendix H) that no single descriptor consistently dominates the others. This demonstrates the importance of considering a wide variety of graph featurizers. We summarize our descriptors in Table 5.

Table 5: Generic graph descriptors.

| Descriptor | Meaning | Reference |
|---|---|---|
| Clust. | Histogram of clustering coefficients, discretized to 100 bins in $[0, 1]$ | You et al. (2018) |
| Deg. | Histogram of node degrees | You et al. (2018) |
| GIN | Activations of a randomly initialized GIN graph neural network | Thompson et al. (2022) |
| Eig. | Histogram of Laplacian spectrum, discretized to 200 bins in $[-10^{-5}, 2]$ | Liao et al. (2019) |
| Orb. 4 | 4-node orbit counts | You et al. (2018); Hočevar (2025) |
| Orb. 5 | 5-node orbit counts | Hočevar (2025) |

## C.2 MOLECULE-SPECIFIC DESCRIPTORS

We propose several novel descriptors for evaluating generative models for molecules via the PolyGraphScore framework. Some of these descriptors are established in chemoinformatics and are computed via RDKit (RDKit, 2024). Namely, topological quantities (`rdkit.Chem.GraphDescriptors`), physico-chemical parameters (`rdkit.Chem.Lipinski`) and classical Morgan molecule fingerprints (`rdkit.Chem.AllChem.GetMorganGenerator`). Additionally, we use learned representations extracted either from a SMILES-based LSTM model (Mayr et al., 2018) (termed ChemNet), or from the contrastively trained MolCLR graph neural network (Wang et al., 2022). The SMILES-based model has previously been used to formulate the Fréchet ChemNet distance (Preuer et al., 2018). To obtain more compact features, we map the learned representations into a 128-dimensional space via sparse random projections with a fixed random seed.

These descriptors can only be computed for molecular graphs which can be converted into `rdkit.Chem.rdchem.Mol` objects, i.e., for graphs which are chemically valid. Hence, we must

filter generated graphs before computing a PGD score. A similar approach has been taken in the Fréchet ChemNet distance.

We summarize these descriptors in more detail in Table 6.

Table 6: Descriptors used for molecular graphs.

| Descriptor | Meaning | Features | Reference |
|---|---|---|---|
| Morgan | 128-D Morgan count fingerprint | Substructure hash counts | RDKit (2024) |
| ChemNet | 128-D projection of ChemNet embedding of canonical SMILES string | Latent | Mayr et al. (2018) |
| MolCLR | 128-D projection of MolCLR embedding of molecule graph | Latent | Wang et al. (2022) |
| Topo | Topological/topochemical descriptors based on the bond structure | 1. AvgIpc
2. BertzCT
3. BalabanJ
4. HallKierAlpha
5. Kappa1
6. Kappa2
7. Kappa3
8. Chi0
9. Chi0n
10. Chi0v
11. Chi1
12. Chi1n
13. Chi1v
14. Chi2n
15. Chi2v
16. Chi3n
17. Chi3v
18. Chi4n
19. Chi4v | RDKit (2024) |
| Lipinski | Structural and physico-chemical parameters | 1. HeavyAtomCount
2. NHOHCount
3. NOCount
4. NumHAcceptors
5. NumHDonors
6. NumHeteroatoms
7. NumRotatableBonds
8. RingCount
9. NumAliphaticCarbocycles
10. NumAliphaticHeterocycles
11. NumAliphaticRings
12. NumAromaticCarbocycles
13. NumAromaticHeterocycles
14. NumAromaticRings
15. NumHeterocycles
16. NumSaturatedCarbocycles
17. NumSaturatedHeterocycles
18. NumSaturatedRings
19. NumAmideBonds
20. NumAtomStereoCenters
21. NumUnspecifiedAtomStereoCenters
22. NumBridgeheadAtoms
23. NumSpiroAtoms
24. FractionCSP3
25. Phi | RDKit (2024) |

## D  MMD as Linear Classification Risk

In this section, we expand on the discussion in Section 3.1 and derive how MMD may be seen as the optimal risk for distinguishing between $P$ and $Q$ of a binary classifier in the reproducing kernel Hilbert space $\mathcal{H}$.

Using the notation $\mathbb{E}_x[k(x, \cdot)]$ for the Riesz representative of the (under mild conditions) bounded linear form $f \mapsto \mathbb{E}_x[\langle f, k(x, \cdot)\rangle]$, one may show:

$$
\begin{aligned}
\mathrm{MMD}(P, Q, k) &= \|\mathbb{E}_{x\sim P}[k(x, \cdot)] - \mathbb{E}_{y\sim Q}[k(y, \cdot)]\|_{\mathcal{H}} \\
&= \sup_{\|D\|_{\mathcal{H}}\le 1} \langle D, \mathbb{E}_{x\sim P}[k(x, \cdot)] - \mathbb{E}_{y\sim Q}[k(y, \cdot)]\rangle \\
&= \sup_{\|D\|_{\mathcal{H}}\le 1} \langle D, \mathbb{E}_{x\sim P}[k(x, \cdot)]\rangle - \langle D, \mathbb{E}_{y\sim Q}[k(y, \cdot)]\rangle \\
&= \sup_{\|D\|_{\mathcal{H}}\le 1} \mathbb{E}_{x\sim P}[D(x)] - \mathbb{E}_{y\sim Q}[D(y)].
\end{aligned}
\tag{3}
$$

We use the Cauchy-Schwarz inequality in the second equality, the linearity of the inner product in the third equality, and the definition of the Riesz representative in the last equality.

This framing reveals that MMD is precisely the optimal linear classification risk achievable by a discriminator $D$ in the unit ball of the function space induced by the kernel.

## E  Background on $f$-Divergences and Total Variation Distance

Let $P$ and $Q$ be probability measures on $\mathcal{X}$ that are assumed to be absolutely continuous with respect to a base measure $\mu$, having densities $p$ and $q$. For now, also assume $P$ to be absolutely continuous w.r.t. $Q$. For a convex, lower-semicontinuous function $f : \mathbb{R}_+ \to \mathbb{R}$ satisfying $f(0) = 1$, the $f$-divergence of $P$ from $Q$ is defined as:

$$
D_f(P \| Q) := \int_{\mathcal{X}} q(x) f\left(\frac{p(x)}{q(x)}\right) d\mu
\tag{4}
$$

As shown by Nguyen et al. (2010), $f$-divergence can be estimated via a variational objective similar to that of MMD. Using the Fenchel conjugate $f^*(v) := \sup_{u\in\mathbb{R}_+} uv - f(u)$, the $f$-divergence is lower-bounded by:

$$
D_f(P \| Q) \ge \sup_{D\in\mathcal{F}} \mathbb{E}_{x\sim P}[D(x)] - \mathbb{E}_{y\sim Q}[f^*(D(x))],
\tag{5}
$$

for any family $\mathcal{F}$ of measurable functions $D : \mathcal{X} \to \mathbb{R}$. The bound is tight if and only if the functional class $\mathcal{F}$ is sufficiently expressive to contain a subderivative of $f$ at the density ratio $p(x)/q(x)$. Such a function then achieves the supremum. The variational formulation of the Jensen-Shannon divergence in Eqn. (2) is a special case of Eqn. (5)

**Total Variation Distance.**  The total variation (TV) distance corresponds to $f(x) = \frac{1}{2}|1 - x|$. One may easily verify that the integral in Eqn. (4) evaluates to half of the $L^1$ distance between $p$ and $q$. As we show in Appendix F.1, its variational objective in Eqn. (5) can be reduced to:

$$
\sup_{\substack{D:\mathcal{X}\to[0,1] \\ \gamma\in[0,1]}} \mathbb{E}_{x\sim P}[[D(x) > \gamma]] - \mathbb{E}_{x\sim Q}[[D(x) > \gamma]],
\tag{6}
$$

where we use the Iverson bracket $[D(x) > \gamma]$ to denote the binarization of $D$ at the threshold $\gamma$. This objective is also known as the Informedness (or Youden's J statistic) of the discriminator $D$. It has a clear geometric interpretation as the maximum vertical distance between the ROC curve of $D$ and the chance diagonal, with a fixed scale of $[0, 1]$.

## F  PGD-TV: Estimating Total Variation Distances

In this section, we propose an alternative variant of the PGD, using variational estimates of the total variation (TV) distance in place of the Jensen-Shannon distance. We term this variant PGD-TV.

We recall from Appendix E that the variational objective for the TV distance is given by the informedness of a dichotomized classifier. We provide a proof of this fact in Appendix F.1. When computing PGD-TV, the choice of binarization threshold is considered part of the fitting process of the classifier. Hence, we choose $\gamma$ to maximize the vertical distance of the ROC on the *fit* set. We refer to Appendix B for pseudocode. In Appendices F.2 and F.3, we present an empirical investigation of PGD-TV, analogous to the experiments presented in Sections 5.2 and 5.3. Finally, we discuss the advantages of PGD over PGD-TV in Appendix F.4

### F.1 VARIATIONAL FORMULATION OF TV DISTANCE

One may easily verify that for $f(u) = \frac{1}{2}|1 - u|$, we have the following Fenchel conjugate:

$$f^*(v) = \sup_{u \in \mathbb{R}_+} uv - \frac{1}{2}|1 - u| = \begin{cases} -\frac{1}{2} & \text{if} \quad v < -\frac{1}{2} \\ v & \text{if} \quad v \in [-\frac{1}{2}, \frac{1}{2}] \\ \infty & \text{if} \quad v > \frac{1}{2} \end{cases} \tag{7}$$

We recall the variational lower bound:

$$D_{TV}(P \| Q) \geq \sup_{D \in \mathcal{F}} \mathbb{E}_{x \sim P}[D(x)] - \mathbb{E}_{y \sim Q}[f^*(D(x))] \tag{8}$$

Without weakening the lower bound, we may restrict ourselves to families of functions which are upper-bounded by $\frac{1}{2}$ almost everywhere w.r.t. $Q$. Indeed, discriminators $D$ that do not satisfy this have a variational bound of $-\infty$. Since we are assuming $P \ll Q$, the discriminators are then also upper-bounded almost everywhere w.r.t. $P$. Hence, w.l.o.g., we may assume that they are upper-bounded by $\frac{1}{2}$ everywhere. Under these assumptions, we obtain the simpler formulation:

$$D_{TV}(P \| Q) \geq \sup_{D \in \mathcal{F}} \mathbb{E}_{x \sim P}[D(x)] - \mathbb{E}_{y \sim Q}\left[\max\left(D(x), -\frac{1}{2}\right)\right]$$
$$= \sup_{D \in \mathcal{F}} \int_{\mathcal{X}} D(x)p(x) - \max\left(D(x), -\frac{1}{2}\right)q(x)d\mu \tag{9}$$

Under the constraint that $D(x) \leq \frac{1}{2}$, we may maximize the expression above in a pointwise fashion by:

$$D(x) = \begin{cases} \frac{1}{2} & \text{if} \quad p(x) > q(x) \\ -\frac{1}{2} & \text{if} \quad p(x) \leq q(x) \end{cases} \tag{10}$$

We note that this is consistent with the finding of Nguyen et al. (2010) that $D(x)$ should attain a subderivative of $f$ at the point $\frac{p(x)}{q(x)}$. Therefore, without weakening the lower bound, we may write:

$$D_{TV}(P \| Q) \geq \sup_{D:\mathcal{X} \to \{-\frac{1}{2}, \frac{1}{2}\}} \mathbb{E}_{x \sim P}[D(x)] - \mathbb{E}_{x \sim Q}\left[\max\left(D(x), -\frac{1}{2}\right)\right]$$
$$= \sup_{D:\mathcal{X} \to \{-\frac{1}{2}, \frac{1}{2}\}} \mathbb{E}_{x \sim P}[D(x)] - \mathbb{E}_{x \sim Q}[D(x)]$$
$$= \sup_{D:\mathcal{X} \to \{0, 1\}} \mathbb{E}_{x \sim P}[D(x)] - \mathbb{E}_{x \sim Q}[D(x)] \tag{11}$$
$$= \sup_{\substack{D:\mathcal{X} \to [0,1] \\ \gamma \in [0,1]}} \mathbb{E}_{x \sim P}[[D(x) > \gamma]] - \mathbb{E}_{x \sim Q}[[D(x) > \gamma]]$$

The first equality is derived from the observation that $D(x) \geq -\frac{1}{2}$ always holds and the maximum is therefore redundant. The second equality is obtained by noting that the expression is invariant under the addition of constants to $D$ (in this case, we add $\frac{1}{2}$).

Without relying on the results of Nguyen et al. (2010), we now show that this bound is tight, even when $P \not\ll Q$. To work in this more general setting, we redefine the total variation distance as half the $L^1$ distance of $p$ and $q$:

$$D_{TV}(P \| Q) := \frac{1}{2}\|p - q\|_{L^1(\mathcal{X}, \mu)} = \frac{1}{2}\int_{\mathcal{X}}|p(x) - q(x)|d\mu \tag{12}$$

One may verify that this matches our original definition when $P \ll Q$. For any measurable set $A \subset \mathcal{X}$, we note that:

$$1 = \int_A p(x)d\mu + \int_{A^C} p(x)d\mu = \int_A q(x)d\mu + \int_{A^C} q(x)d\mu \qquad (13)$$

Hence, rearranging, we obtain:

$$\int_A p(x) - q(x)d\mu = \int_{A^C} q(x) - p(x)d\mu \qquad (14)$$

Defining $A := \{x \in \mathcal{X} \ : \ p(x) \geq q(x)\}$ and applying this identity, we get:

$$\frac{1}{2}\int_{\mathcal{X}} |p(x) - q(x)|d\mu = \frac{1}{2}\int_A p(x) - q(x)d\mu + \frac{1}{2}\int_{A^C} q(x) - p(x)d\mu$$
$$= \int_A p(x) - q(x)d\mu \qquad (15)$$

Since $A$ is exactly the set on which $p(x) - q(x)$ is non-negative, it is also clear that for any other $B \subset \mathcal{X}$, we have:

$$\frac{1}{2}\int_{\mathcal{X}} |p(x) - q(x)|d\mu \geq \int_B p(x) - q(x)d\mu \qquad (16)$$

Thus, we may write:

$$D_{TV}(P \parallel Q) = \sup_{B \subset \mathcal{X}} \int_B p(x) - q(x)d\mu$$
$$= \sup_{D:\mathcal{X}\to\{0,1\}} \int_{\mathcal{X}} D(x)(p(x) - q(x))d\mu \qquad (17)$$
$$= \sup_{D:\mathcal{X}\to\{0,1\}} \mathbb{E}_{x\sim P}[D(x)] - \mathbb{E}_{x\sim Q}[D(x)]$$

This is exactly the variational lower bound which we have derived above. Hence, we have shown it to be tight, even in the setting where $P \not\ll Q$.

### F.2 PGD-TV Tracks Synthetic Data Perturbations

We now present perturbation experiments for the PGD-TV variant that are analogous to those shown in Section 5.2.

We plot a summary of the Spearman correlation of the metrics with perturbation magnitude in Fig. 6. Compared to Fig. 3, we find that PGD-TV exhibits slightly lower correlations. Figs. 7 and 8 show the response of PGD-TV to perturbation over the entire and cropped magnitude range, respectively. For a more detailed explanation of this type of plot, we refer to Appendix H. From the plots we conclude that PGD-TV qualitatively exhibits the expected behavior of increasing with perturbation magnitude and eventually saturating. However, in some cases (e.g., edge addition on proteins), the PGD-TV flattens out, leading to lower correlations.

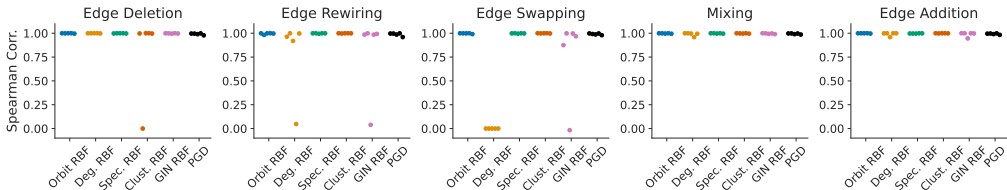

Figure 6: Spearman correlation of MMD metrics and PGD-TV with the magnitude of perturbation of datasets.

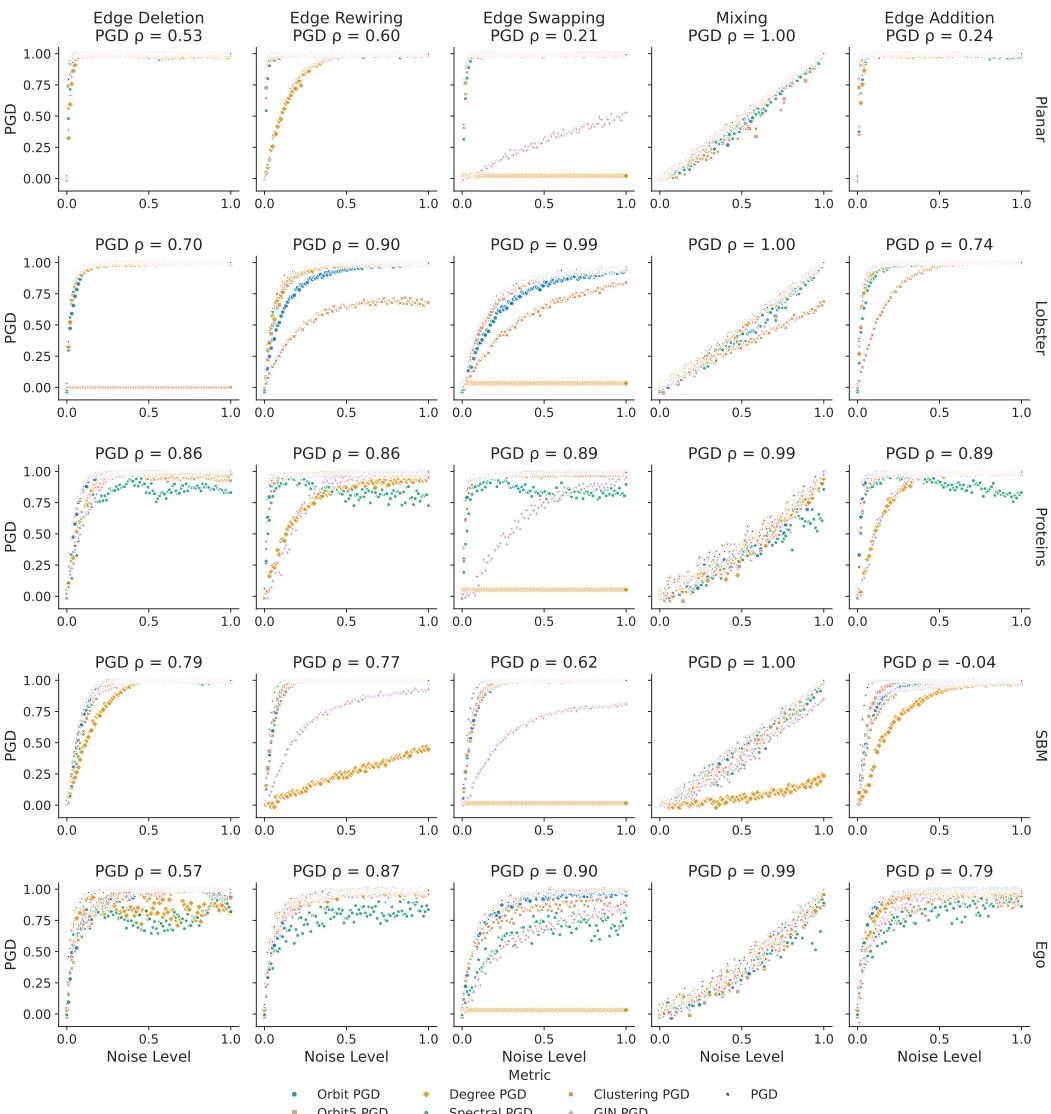

Figure 7: Behavior of descriptor-specific and aggregated PGD-TV as data distributions are perturbed. The perturbation type varies across rows while dataset varies across columns. The Spearman correlation of the aggregate PGD and the perturbation level is denoted by $\rho$.

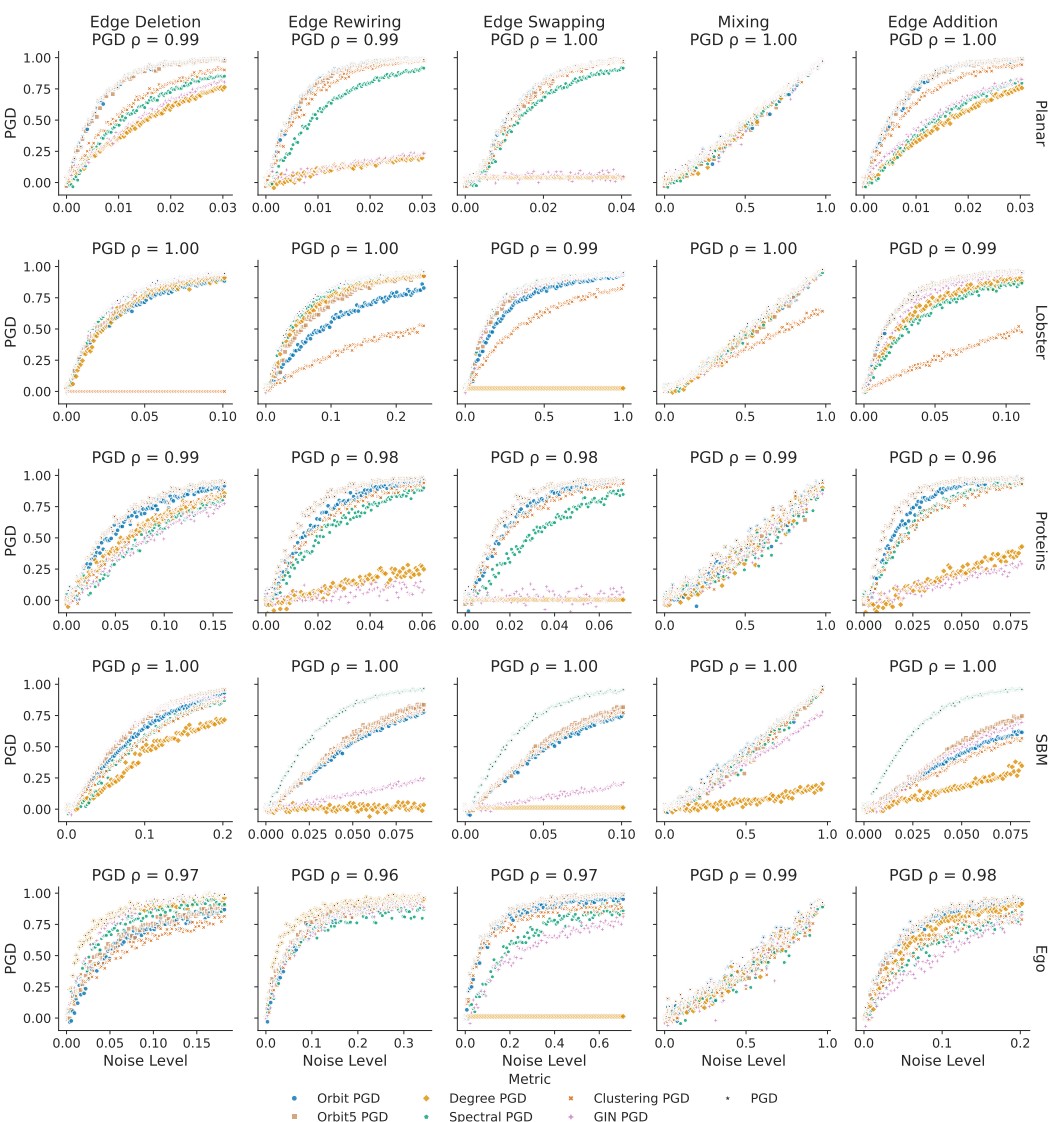

Figure 8: Behavior of descriptor-specific and aggregated PGD-TV as data distributions are perturbed. The perturbation type varies across rows, while the dataset varies across columns. The Spearman correlation of the aggregate PGD and the perturbation level is denoted by $\rho$.

## F.3 PGD-TV CORRELATES WITH MODEL QUALITY

Analogous to Section 5.3, we now investigate how the PGD-TV variant correlates with proxy variables of model quality. In Fig. 9, we illustrate how PGD-TV behaves as the number of denoising steps in DiGress is varied. As in Fig. 4, we find that PGD-TV correlates with validity in a highly linear fashion.

As in Section 5.3, we compute Pearson correlation coefficients between PGD-TV and validity. When varying the number of denoising steps, we find that PGD-TV exhibits a more linear relationship with validity than any of the MMD metrics.

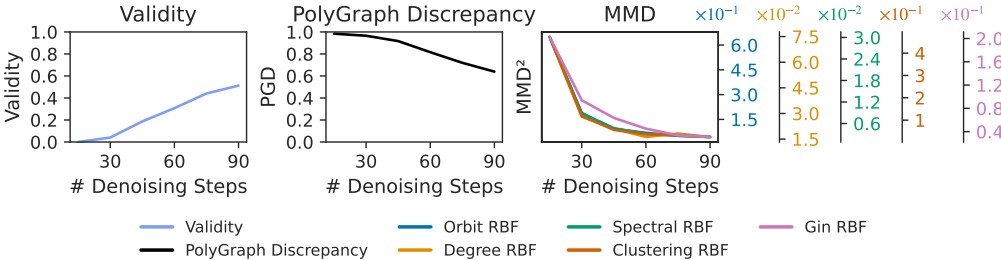

Figure 9: Behavior of validity, PGD-TV, and MMDs as the number of denoising steps in DiGress is varied on PLANAR-L.

We examine the behavior of PGD-TV throughout training in Fig. 9. Qualitatively, a clear positive relationship emerges between training duration and PGD-TV. This trend is confirmed quantitatively in Table 8, where Spearman correlation coefficients show that most MMD metrics often exhibit weak or negative correlations, while PGD-TV consistently correlates positively with training duration. However, this correlation is weaker than that of PGD-JS (see Table 3) and the clustering-based MMD metric. A similar pattern appears in Table 7 (bottom three rows): PGD-TV correlates reliably with validity, whereas most MMD metrics show inconsistent behavior. Nevertheless, in two out of three cases, the clustering-based MMD metric achieves a stronger correlation with validity than PGD-TV.

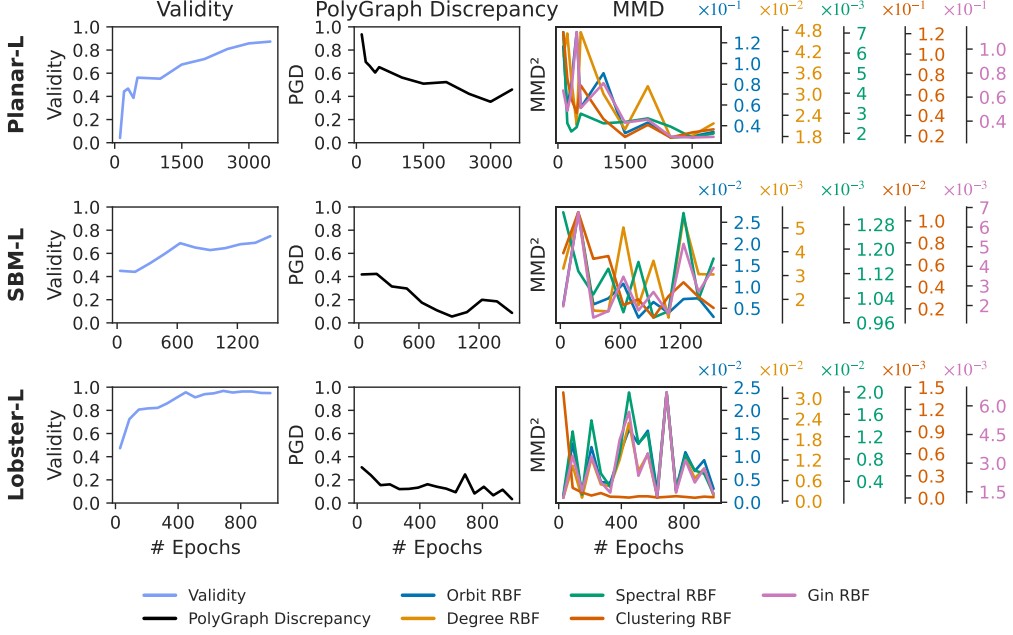

Figure 10: Behavior of validity, PGD-TV, and MMD metrics throughout training of DiGress on procedural graph datasets.

Table 7: Negative Pearson correlation (↑) of validity with other performance metrics. Denoising refers to the experiments in which we vary the number of denoising iterations. Training refers to the experiments in which we monitor performance metrics during the training of DiGress models.

| | Dataset | TV-PGD | Orb. RBF | Deg. RBF | Eig. RBF | Clust. RBF | GIN RBF |
|---|---|---|---|---|---|---|---|
| Denoising | Planar-L | **98.56** | 73.50 | 70.87 | 73.41 | 71.36 | 82.52 |
| | Planar-L | **95.23** | 84.32 | 59.11 | 76.47 | 88.31 | 68.11 |
| Training | SBM-L | **85.56** | 53.22 | 6.59 | 23.49 | 85.07 | 6.85 |
| | Lobster-L | 70.49 | -34.65 | -35.17 | -22.95 | **87.05** | -30.55 |

Table 8: Sign-adjusted Spearman correlation (↑) of validity, PGD-TV, and MMDs with number of training iterations of DiGress.

| Dataset | Val. | PGD | Orb. RBF | Deg. RBF | Eig. RBF | Clust. RBF | GIN RBF |
|---|---|---|---|---|---|---|---|
| Planar-L | **96.36** | 95.45 | 75.45 | 68.18 | 50.00 | 86.36 | 77.27 |
| SBM-L | **87.27** | 72.73 | 23.64 | 10.91 | 10.91 | 68.18 | -24.55 |
| Lobster-L | **85.47** | 66.58 | -7.35 | -13.24 | 6.86 | 68.14 | 1.23 |

### F.4 Comparison of PGD and PGD-TV

Overall, the experiments in Appendices F.2 and F.3 have demonstrated that PGD-TV is a viable alternative to the PGD metric we presented in the main paper, correlating to a high degree with synthetic data perturbations and proxy variables of model quality. Nevertheless, we found that PGD exhibits stronger correlations and appears like a more robust choice.

While we have no definite explanation for these observations, we hypothesize that the choice of binarization threshold in PGD-TV may introduce some noise into the estimate. Additionally, maximum likelihood classifiers (like logistic regression) inherently maximize the log-likelihood objective of the JS divergence. Bayesian inference (approximated by TabPFN) may be expected to behave similarly in the large sample size limit (van der Vaart, 1998). However, neither maximum likelihood estimation nor Bayesian inference directly optimizes the variational objective of the TV distance, i.e., informedness. This can lead to a misalignment when estimating the PGD-TV, potentially resulting in looser variational bounds.

For these reasons, we recommend using the PGD variant presented in the main paper, estimating lower bounds on the Jensen-Shannon distance.

## G Supplemental for: High Bias and Variance Plague MMD-based GGM Benchmarks

Here, we show that the conclusions of Section 5.1 expand to all combinations of models, descriptors, and datasets, and provide additional experimental details. All MMD estimates provided here and in Figs. 2a and 2b are RBF MMDs, as proposed by Thompson et al. (2022). The kernel is selected by taking the maximum over the bandwidths $\{\sigma_i\}_{i=1}^{1}0 = \{0.01, 0.1, 0.25, 0.5, 0.75, 1.0, 2.5, 5.0, 7.5, 10.0\}$.

Specifically, we subsampled 8 to 4096 graphs 100 times with replacement from a total of 8192 samples for the reference and generated graphs. We subsequently computed the median, $5^{th}$ and $95^{th}$ quantiles to estimate the variation of MMD. We computed such experiments for all model-generated samples we considered (ESGG, AutoGraph, DiGress and GRAN) and considered all descriptors (degree histogram, clustering histogram, orbit count for graphlet sizes 4 and 5, and the graph Laplacian eigenvalues) and all procedural datasets (SBM, Lobster and Planar).

Based on those findings, we introduce Planar-L, SBM-L, and Lobster-L, larger versions of the previously used datasets. Details for these new datasets are presented in Appendix M

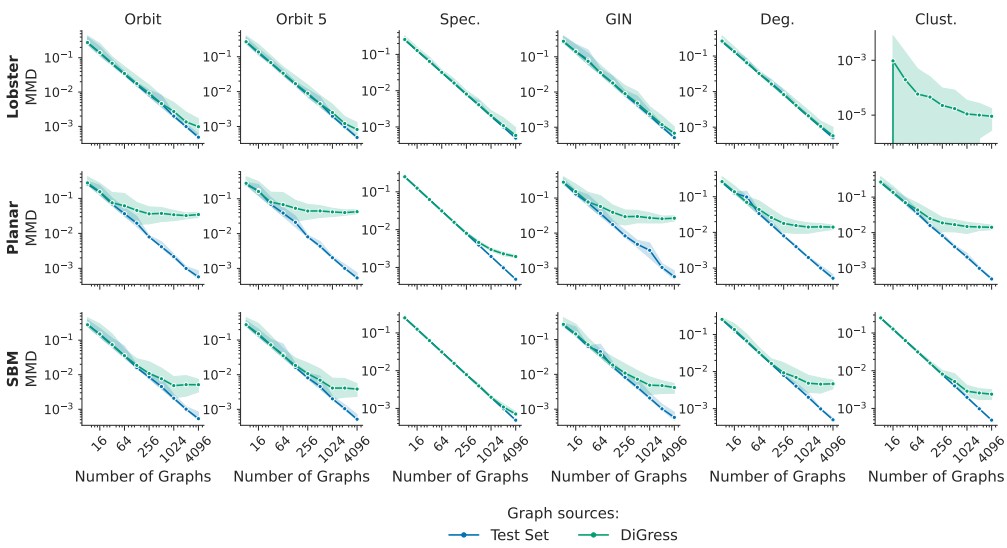

Figure 11: Behavior of biased MMD estimates as the number of samples is varied for DiGress.

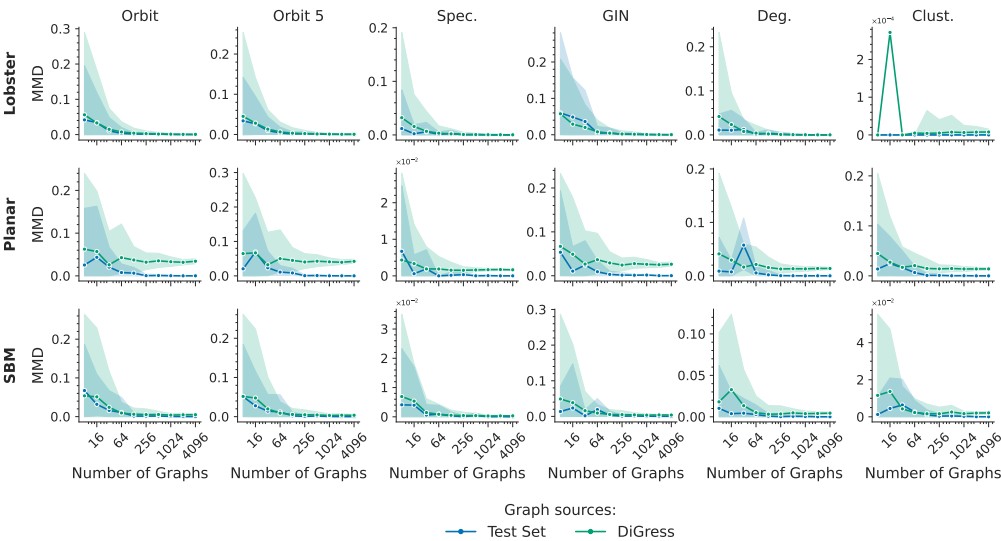

Figure 12: Behavior of unbiased MMD estimates as the number of samples is varied for DiGress.

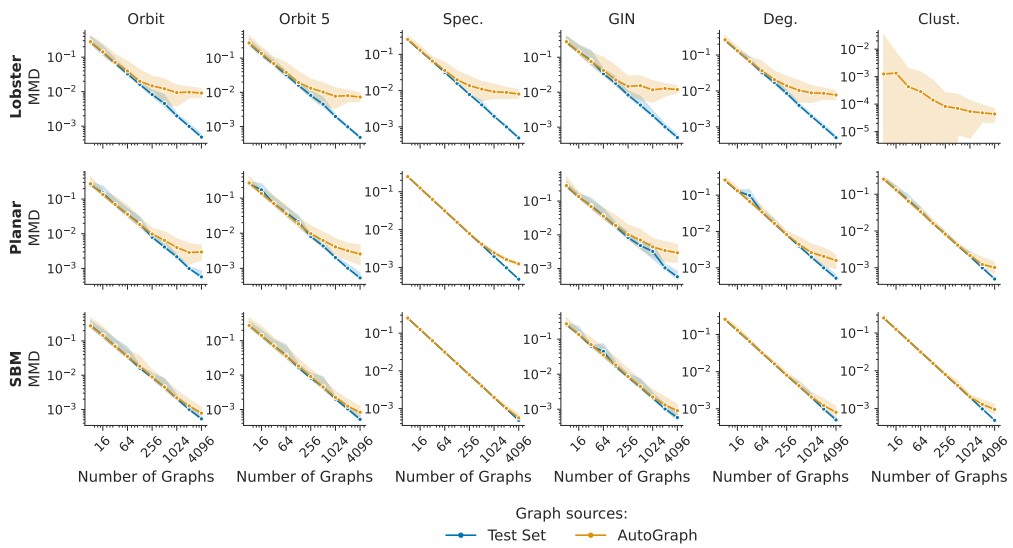

Figure 13: Behavior of biased MMD estimates as the number of samples is varied for AutoGraph.

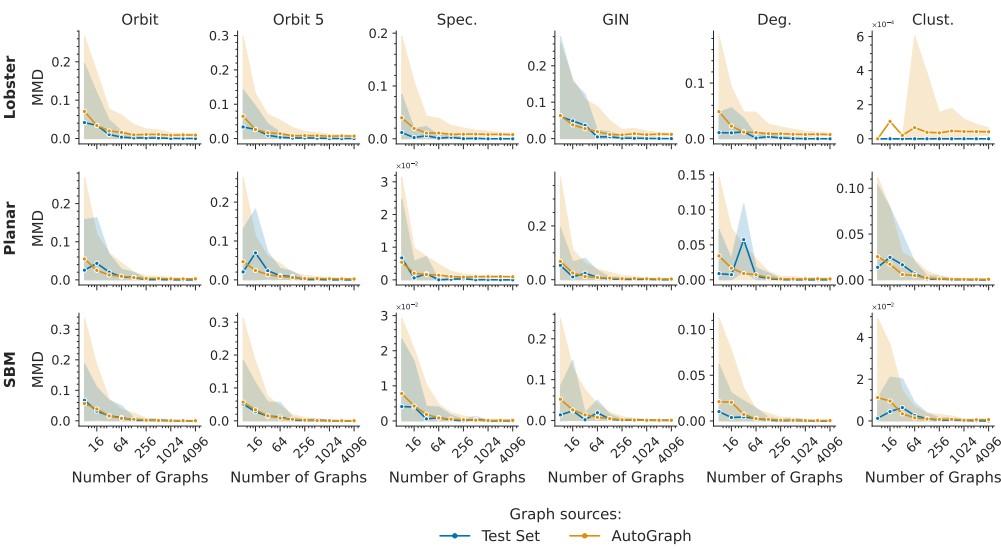

Figure 14: Behavior of unbiased MMD estimates as the number of samples is varied for AutoGraph.

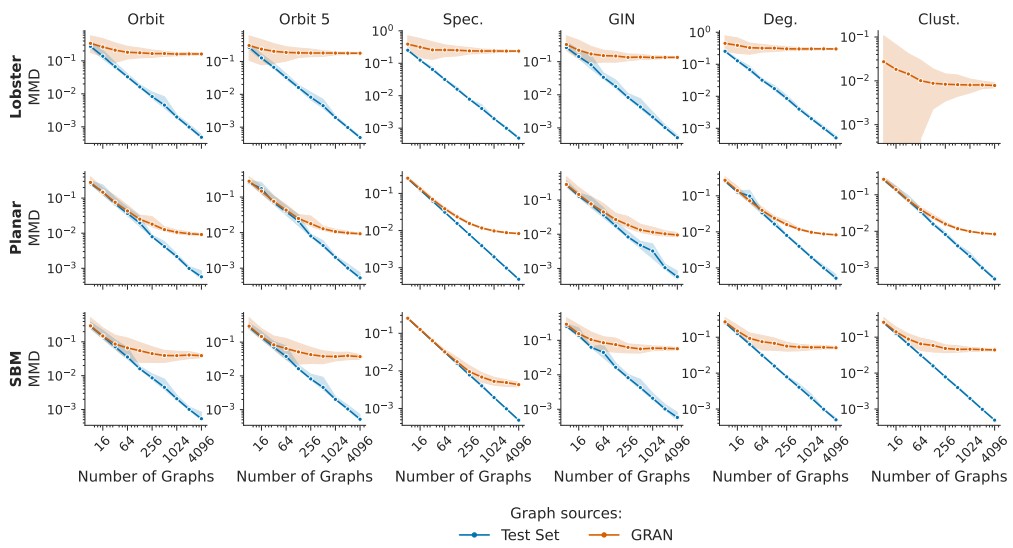

Figure 15: Behavior of biased MMD estimates as the number of samples is varied for GRAN.

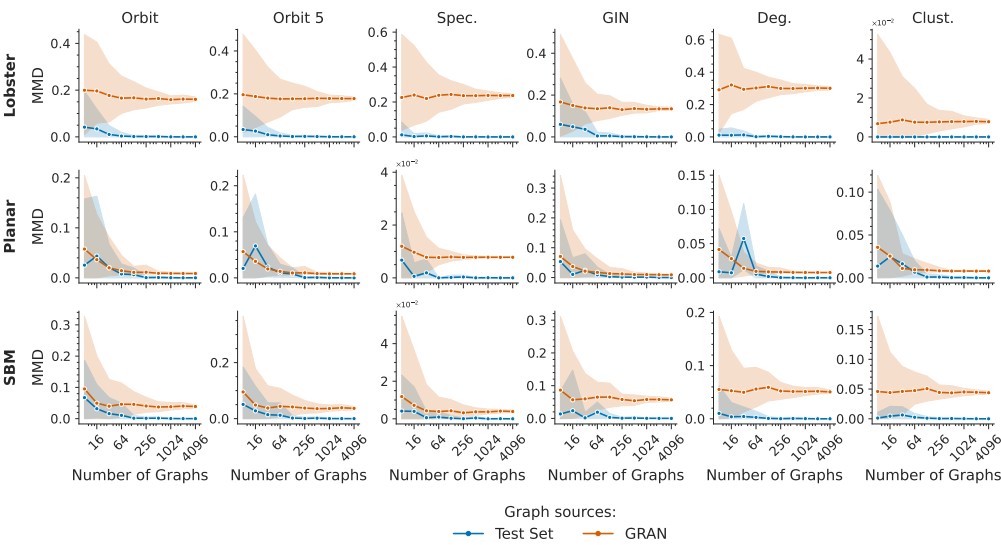

Figure 16: Behavior of unbiased MMD estimates as the number of samples is varied for GRAN.

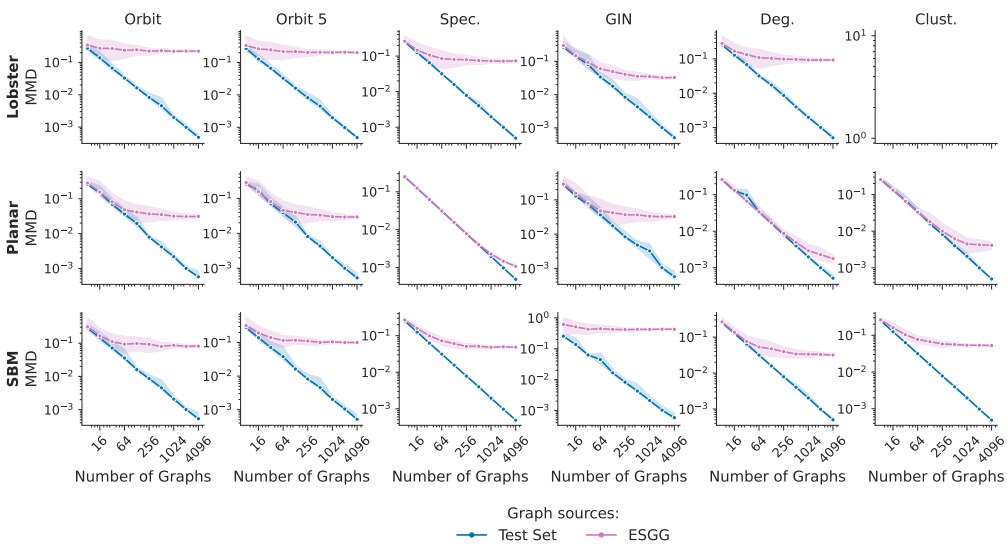

Figure 17: Behavior of biased MMD estimates as the number of samples is varied for ESGG.

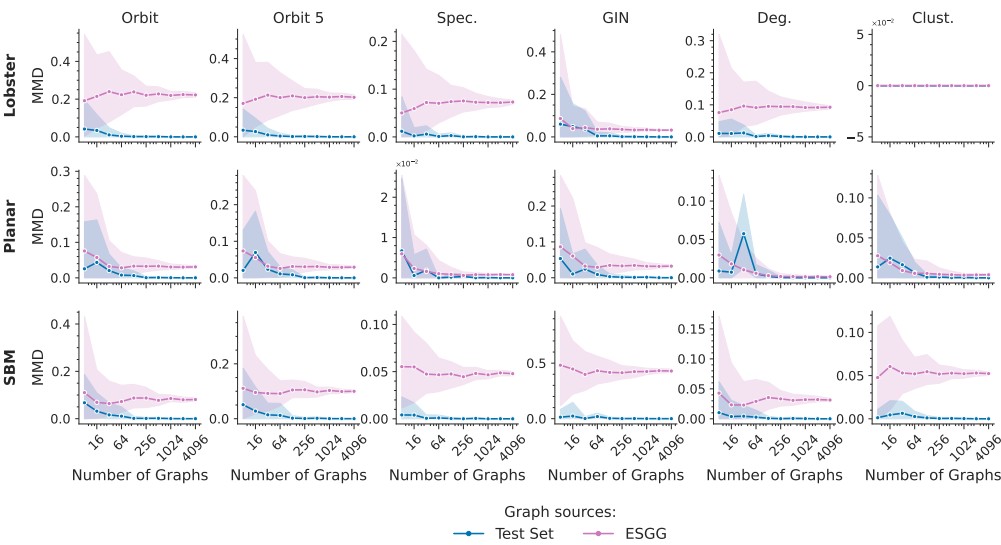

Figure 18: Behavior of unbiased MMD estimates as the number of samples is varied for ESGG.

# H SUPPLEMENTAL FOR: PGD TRACKS SYNTHETIC DATA PERTURBATIONS

In this section, we provide further details for the experiments presented in Section 5.2. In particular, we illustrate in more detail how PGD responds to perturbations and present results for the TV variant.

In Fig. 19, we illustrate how PGD (descriptor-specific scores and the summary PGD) responds to various perturbations on different datasets. In this figure, we illustrate the response over the whole range of magnitudes [0, 1]. As anticipated, the PGD saturates quickly as the support of the perturbed distribution becomes disjoint from the support of the true data distribution. We note that the PGD consistently responds in a monotonic fashion to the magnitude of perturbation.

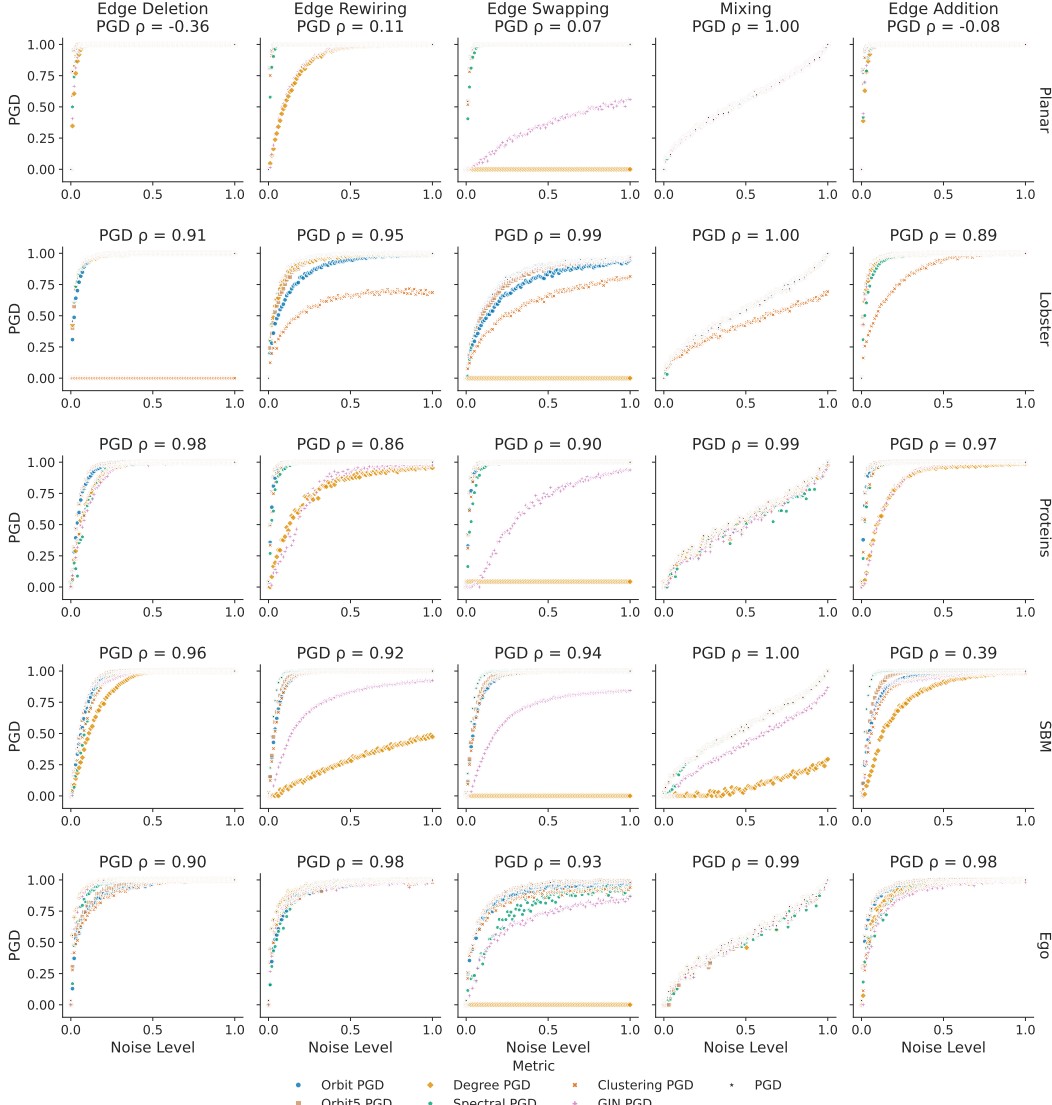

Figure 19: Behavior of descriptor-specific and aggregated PGD (JS) as data distributions are perturbed.

Based on the data from Fig. 19, we select a threshold for each combination of perturbation type and dataset at which the summary PGD saturates above 0.95. We illustrate the behavior of PGD-JS on these cropped ranges in Fig. 20.

We find that there is no single descriptor that consistently provides the tightest PGD estimate. This highlights the importance of evaluating many different descriptors when computing a PGD.

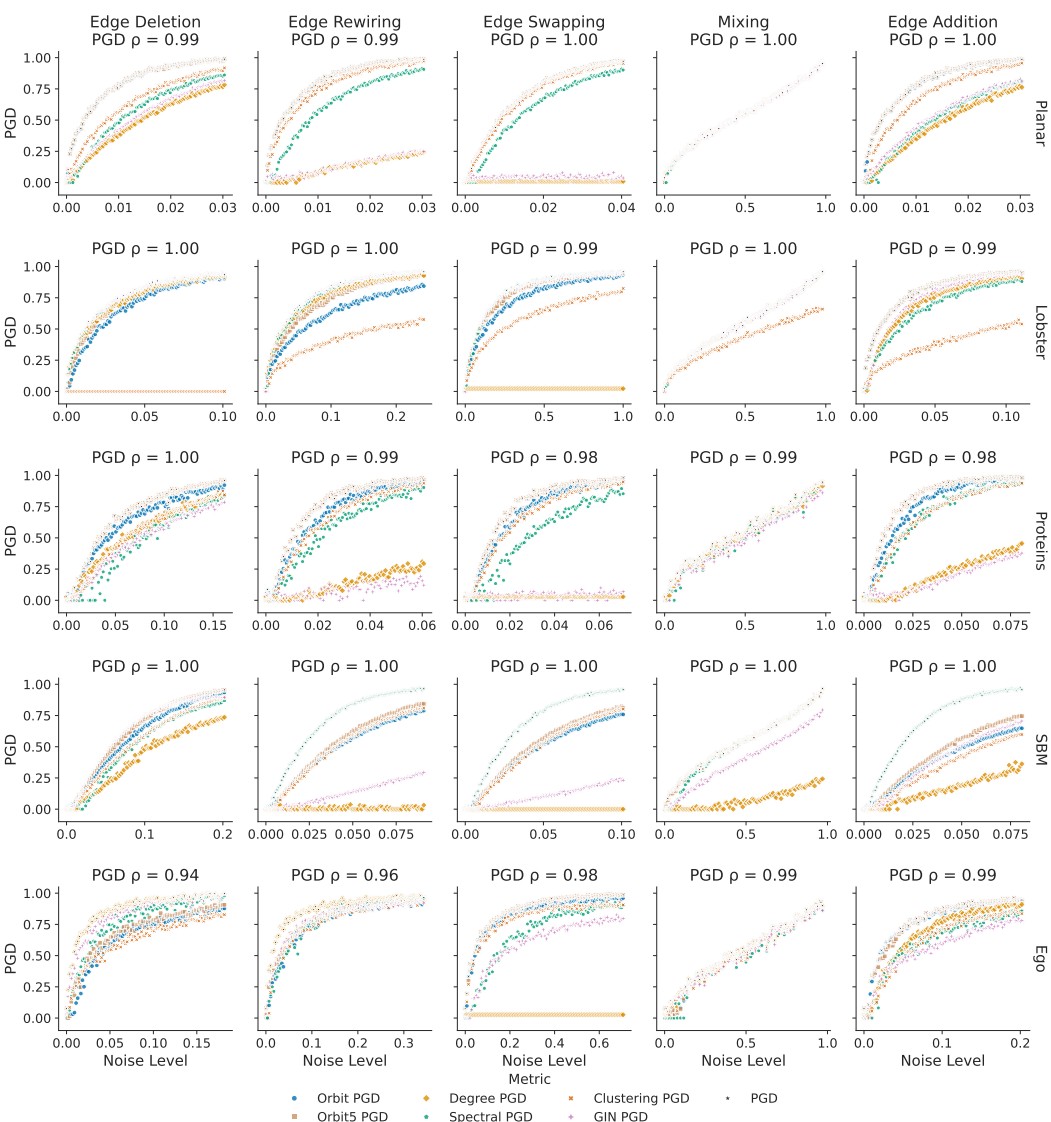

Figure 20: Behavior of descriptor-specific and aggregated PGD (JS) as data distributions are perturbed. The perturbation type varies across rows, while the dataset varies across columns. The Spearman correlation of the aggregate PGD and the perturbation level is denoted by $\rho$.

# I SUPPLEMENTAL FOR: PGD CORRELATES WITH MODEL QUALITY

In this section, we provide further details for the experiments presented in Section 5.3.

In Table 9 we provide the exact MMD metrics attained by DiGress as the number of denoising iterations is varied. Analogously, we provide the values of the PGD and descriptor-specific subscores in Table 10. We find that orbit counts appear to be the most discriminative descriptors, as they lead to the highest PGD values.

Table 9: Behavior of RBF-based MMD metrics as the number of denoising steps in DiGress is varied. A separate model is trained for each row for 5k epochs on PLANAR-L.

| Steps | Orb. RBF | Deg. RBF | Eig. RBF | Clust. RBF | GIN RBF |
|---|---|---|---|---|---|
| 15 | 0.6489 | 0.0748 | 0.0301 | 0.4722 | 0.2027 |
| 30 | 0.1892 | 0.0281 | 0.0088 | 0.1181 | 0.0939 |
| 45 | 0.0925 | 0.0212 | 0.0047 | 0.0566 | 0.0636 |
| 60 | 0.0686 | 0.0162 | 0.0031 | 0.0363 | 0.0451 |
| 75 | 0.0511 | 0.0182 | 0.0026 | 0.0331 | 0.0338 |
| 90 | 0.0435 | 0.0162 | 0.0022 | 0.0246 | 0.0308 |

Table 10: Behavior of PGD as the number of denoising steps in DiGress is varied. A separate model is trained for each row for 5k epochs on PLANAR-L.

| Steps | VUN | PGD | Orb. | Orb5. | Deg. | Eig. | Clust. | GIN |
|---|---|---|---|---|---|---|---|---|
| 15 | 0.00 | 100.00 | 100.00 | 100.00 | 69.92 | 98.87 | 99.97 | 79.19 |
| 30 | 4.05 | 97.04 | 96.90 | 97.04 | 45.37 | 79.25 | 90.65 | 56.17 |
| 45 | 18.70 | 91.81 | 90.07 | 91.81 | 33.88 | 63.69 | 77.58 | 43.13 |
| 60 | 30.76 | 84.25 | 81.78 | 84.25 | 28.60 | 49.20 | 70.06 | 37.42 |
| 75 | 44.09 | 74.62 | 72.74 | 74.62 | 32.15 | 43.23 | 59.21 | 37.58 |
| 90 | 51.27 | 69.30 | 66.47 | 69.30 | 27.72 | 38.26 | 51.35 | 33.51 |

In Fig. 21, we supplement the experiments presented previously in Fig. 5 with the corresponding results on PLANAR-L and LOBSTER-L.

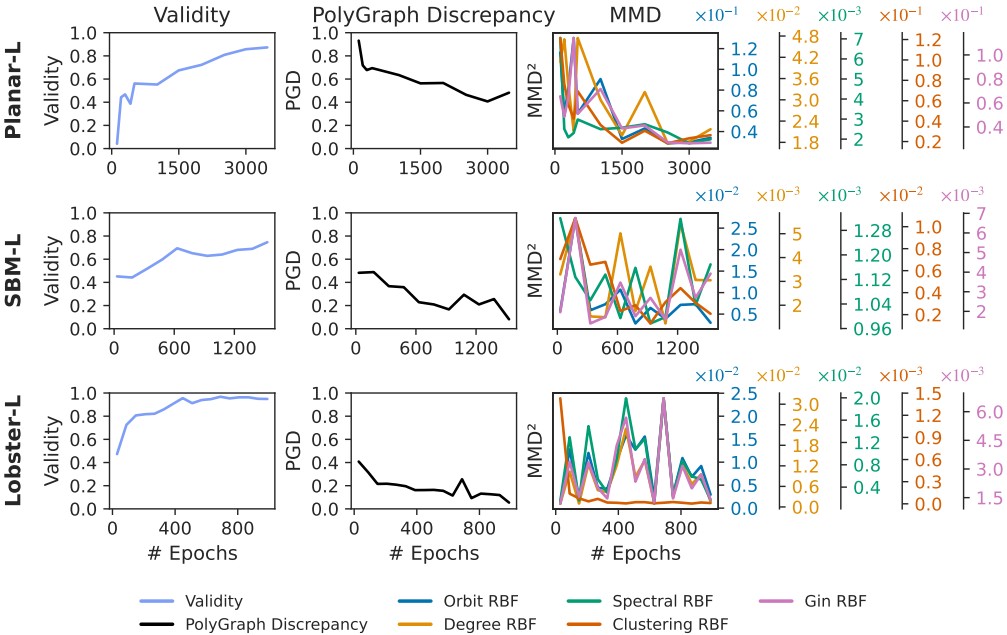

Figure 21: Behavior of validity, PGD, and MMD metrics throughout training of DiGress on procedural datasets.

## J    ABLATION: TABPFN VS LOGISTIC REGRESSION

In this section, we study logistic regression as an alternative to TabPFN as a discriminator. To this end, we repeat the perturbation experiments from Section 5.2 and Appendix H with logistic regression as a discriminator. We refer to the PGD variant using logistic regression LR PGD.

In Fig. 22 we plot the response of PGD and LR PGD to synthetic perturbations. We find that TabPFN consistently produces PGD estimates that are at least as high as those obtained by logistic regression. In some cases, TabPFN clearly outperforms logistic regression. This may be attributed to the fact that TabPFN can model non-linear decision boundaries and is thus more powerful than logistic regression. We also qualitatively observe that logistic regression leads to a noisier response to the variation of perturbation magnitude.

Hence, since TabPFN simultaneously produces tighter bounds and less noisy estimates, we prefer it to logistic regression.

## K    SUPPLEMENTAL FOR: BENCHMARKING REPRESENTATIVE MODELS

In this Appendix, we do a thorough benchmark of PGD and MMD on LOBSTER-L, PLANAR-L, SBM-L and Proteins. To obtain the standard deviations for PGD scores and MMD values in Tables 4 and 11 to 13, we subsample half of the dataset *without replacement* (2048 samples for procedural datasets, and 92 samples for proteins) 10 times. In all those tables, means and standard deviations are scaled by a factor of 100 for legibility purposes. The time taken to compute each of those metrics is reported in Table 15. Timing experiments were run on a compute node equipped with two AMD EPYC 9534 CPUs (using 10 vCPUs in total), an NVIDIA H100 GPU with 80 GB memory (CUDA 12.2, driver 535.230.02), and 128 GB system RAM. Reference values (i.e. the score obtained by computing the metric between the train and test set) for all metrics discussed are in Tables 16 and 17. We note that the PGD discrepancy between the train and test set of MOSES is relatively high, as the test set consists of a separate scaffold split. Importantly, the PGD between the train and test set is very close to 0 (save for MOSES due to changes in the underlying distribution), further showing the absolute nature of PGD, making it much easier to interpret compared to MMD.

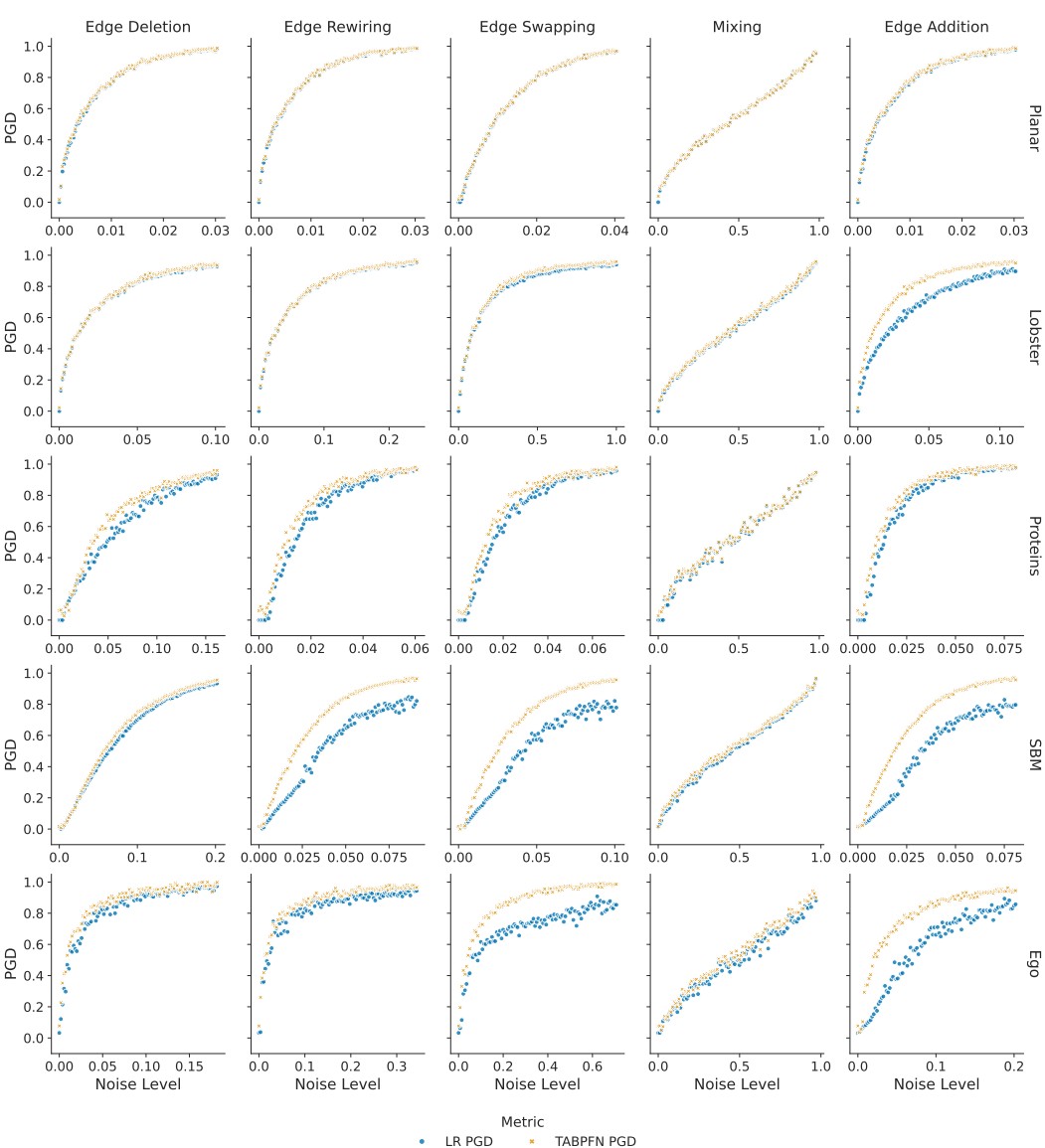

Figure 22: Comparing the behavior of the aggregated PGD (JS) computed via logistic regression (LR PGD) to the aggregated PGD computed via a TabPFN classifier (PGD).

Table 11: Comparison of VUN and PGD with biased Gaussian TV-based MMD formulations from Liao et al. (2019). We computed the standard deviation from 10 subsamples of size 2048 except for Proteins, where the subsample size is 92 (50% of the size of the test set). All MMD hyperparameter choices are specified in table 14.

| Dataset | Model | VUN (↑) | PGD (↓) | GTV MMD$^2$ Deg. (↓) | GTV MMD$^2$ Clust. (↓) | GTV MMD$^2$ Orb. (↓) | GTV MMD$^2$ Eig. (↓) |
|---|---|---|---|---|---|---|---|
| PLANAR-L | AutoGraph | - | **33.451** ±1.312 | 6.541e-05 ±2.412e-05 | **2.239e-03** ±4.579e-04 | **1.219e-04** ±5.647e-05 | 9.894e-04 ±7.280e-05 |
| | DiGress | - | 45.391 ±1.780 | 5.521e-04 ±4.867e-05 | 1.504e-02 ±1.855e-03 | 3.781e-03 ±3.749e-04 | 1.365e-03 ±1.123e-04 |
| | GRAN | - | 99.257 ±1.056 | 7.046e-05 ±2.937e-05 | 4.876e-03 ±1.208e-03 | 5.324e-04 ±1.826e-04 | 1.332e-03 ±1.296e-04 |
| | ESGG | - | 45.012 ±1.388 | **1.896e-05** ±1.094e-05 | 5.787e-03 ±9.100e-04 | 1.333e-03 ±3.332e-04 | **8.016e-04** ±8.585e-05 |
| LOBSTER-L | AutoGraph | - | 16.445 ±1.765 | 3.400e-04 ±9.501e-05 | 4.859e-06 ±3.267e-06 | 5.235e-03 ±1.459e-03 | 9.435e-04 ±1.735e-04 |
| | DiGress | - | **3.924** ±3.248 | **5.168e-05** ±1.793e-05 | 1.525e-06 ±9.071e-07 | **3.385e-04** ±1.601e-04 | **2.171e-04** ±5.957e-05 |
| | GRAN | - | | 1.179e-02 ±5.092e-04 | 3.278e-03 ±4.650e-04 | 1.905e-01 ±6.329e-02 | 2.331e-02 ±6.922e-04 |
| | ESGG | - | 69.881 ±0.554 | 6.423e-03 ±4.194e-04 | **0.000e+00** ±0.000e+00 | 6.474e-02 ±3.729e-03 | 1.064e-02 ±6.960e-04 |
| SBM-L | AutoGraph | - | - | **1.396e-04** ±7.417e-05 | **2.000e-03** ±3.215e-05 | **1.893e-03** ±4.269e-04 | **2.500e-04** ±3.213e-05 |
| | DiGress | - | **16.698** ±1.724 | 7.302e-04 ±2.200e-04 | 2.052e-03 ±2.706e-05 | 3.105e-03 ±4.784e-04 | 3.204e-04 ±4.209e-05 |
| | GRAN | - | - | 9.831e-03 ±4.322e-04 | 3.996e-03 ±1.570e-04 | 1.392e-02 ±1.225e-04 | 1.254e-03 ±7.861e-05 |
| | ESGG | - | 99.377 ±0.215 | 3.442e-03 ±3.866e-04 | 6.816e-03 ±2.112e-04 | 4.541e-02 ±2.479e-03 | 2.765e-02 ±4.144e-04 |
| Proteins | AutoGraph | - | **55.308** ±12.023 | 3.936e-03 ±1.829e-03 | 5.877e-02 ±5.158e-04 | 2.460e-02 ±5.919e-03 | 4.824e-03 ±9.309e-04 |
| | DiGress | - | 86.781 ±2.902 | **6.364e-04** ±5.158e-04 | **5.096e-02** ±6.025e-03 | 1.610e-02 ±7.680e-03 | **2.807e-03** ±4.655e-04 |
| | GRAN | - | 86.062 ±7.232 | 3.526e-02 ±9.218e-03 | 1.425e-01 ±1.437e-02 | 3.239e-01 ±6.636e-02 | 1.199e-02 ±1.829e-03 |
| | ESGG | - | 77.764 ±5.149 | 2.509e-03 ±1.015e-03 | 8.186e-02 ±1.219e-02 | **1.382e-02** ±9.628e-03 | 3.543e-03 ±3.264e-04 |

Table 12: Unbiased RBF kernel-based MMD estimates. We computed the standard deviation from 10 subsamples of size 2048 except for Proteins, where the subsample size is 92 (50% of the size of the test set). All MMD hyperparameter choices are specified in table 14.

| Dataset | Model | UMVE Deg. (↓) | UMVE Clust. (↓) | UMVE Orb. (↓) | UMVE Eig. (↓) |
|---|---|---|---|---|---|
| PLANAR-L | AutoGraph | **1.582e-03** ±7.653e-04 | **5.774e-04** ±3.849e-04 | **2.539e-03** ±1.327e-03 | 1.086e-03 ±8.244e-05 |
| | DiGress | 1.426e-02 ±1.602e-03 | 1.418e-02 ±2.011e-03 | 3.373e-02 ±5.418e-03 | 1.668e-03 ±1.676e-04 |
| | GRAN | 7.801e-03 ±8.410e-05 | 7.889e-03 ±1.033e-04 | 8.947e-03 ±7.599e-04 | 7.807e-03 ±9.350e-05 |
| | ESGG | 1.607e-03 ±7.006e-04 | 4.901e-03 ±1.110e-03 | 3.055e-02 ±4.147e-03 | **8.361e-04** ±1.417e-04 |
| LOBSTER-L | AutoGraph | 6.481e-03 ±2.049e-03 | 4.949e-05 ±3.900e-05 | 7.372e-03 ±2.547e-03 | 7.146e-03 ±1.919e-03 |
| | DiGress | **5.082e-04** ±4.830e-04 | **5.340e-06** ±3.793e-06 | **7.115e-04** ±6.269e-04 | **2.744e-04** ±4.076e-04 |
| | GRAN | 3.032e-01 ±9.305e-03 | 7.566e-03 ±1.163e-03 | 1.608e-01 ±7.665e-03 | 2.356e-01 ±5.756e-03 |
| | ESGG | 8.858e-02 ±4.995e-03 | **0.000e+00** ±0.000e+00 | 2.242e-01 ±1.451e-02 | 6.965e-02 ±5.450e-03 |
| SBM-L | AutoGraph | **5.239e-04** ±4.515e-04 | **4.474e-04** ±2.889e-04 | **6.822e-04** ±8.766e-04 | **1.865e-04** ±8.581e-05 |
| | DiGress | 3.763e-03 ±1.100e-03 | 1.911e-03 ±4.720e-04 | 4.264e-03 ±1.646e-03 | 2.368e-04 ±6.208e-05 |
| | GRAN | 5.071e-02 ±2.771e-03 | 4.454e-02 ±1.381e-03 | 3.575e-02 ±5.071e-03 | 4.043e-03 ±5.350e-04 |
| | ESGG | 3.196e-02 ±2.302e-03 | 5.450e-02 ±2.256e-03 | 8.004e-02 ±8.053e-03 | 4.765e-02 ±8.582e-04 |
| Proteins | AutoGraph | 1.888e-02 ±6.463e-03 | 3.906e-02 ±1.792e-02 | **1.135e-02** ±3.868e-03 | 1.544e-02 ±5.011e-03 |
| | DiGress | **1.718e-02** ±9.130e-03 | **1.740e-02** ±8.327e-03 | 3.312e-02 ±2.999e-02 | **1.601e-03** ±1.711e-03 |
| | GRAN | 3.440e-01 ±8.525e-02 | 2.642e-01 ±4.053e-02 | 3.983e-01 ±8.373e-02 | 5.004e-02 ±1.001e-02 |
| | ESGG | 8.358e-02 ±2.763e-02 | 8.125e-02 ±2.759e-02 | 4.607e-02 ±2.787e-02 | 4.234e-03 ±1.443e-03 |

Table 13: Biased RBF kernel-based MMD estimates. We computed the standard deviation from 10 subsamples of size 2048 except for Proteins, where the subsample size is 92 (50% of the size of the test set). All MMD hyperparameter choices are specified in table 14.

| Dataset | Model | RBF Deg. (↓) | RBF Clust. (↓) | RBF Orb. (↓) | RBF Eig. (↓) |
|---|---|---|---|---|---|
| PLANAR-L | AutoGraph | **2.943e-03** ±7.042e-04 | **2.139e-03** ±2.931e-04 | **3.915e-03** ±1.179e-03 | 2.509e-03 ±6.142e-05 |
| | DiGress | 1.528e-02 ±1.632e-03 | 1.520e-02 ±2.009e-03 | 3.475e-02 ±5.528e-03 | 2.946e-03 ±1.258e-04 |
| | GRAN | 9.743e-03 ±8.840e-05 | 9.833e-03 ±1.032e-04 | 1.079e-02 ±6.316e-04 | 9.752e-03 ±9.332e-05 |
| | ESGG | 3.152e-03 ±6.404e-04 | 5.962e-03 ±1.108e-03 | 3.160e-02 ±4.137e-03 | **2.316e-03** ±1.045e-04 |
| LOBSTER-L | AutoGraph | 7.492e-03 ±2.000e-03 | 6.176e-05 ±4.349e-05 | 8.102e-03 ±2.507e-03 | 8.110e-03 ±1.915e-03 |
| | DiGress | **2.121e-03** ±2.611e-04 | 1.042e-05 ±5.455e-06 | **2.302e-03** ±5.577e-04 | **2.030e-03** ±2.359e-04 |
| | GRAN | 3.043e-01 ±9.296e-03 | 7.728e-03 ±1.174e-03 | 1.619e-01 ±7.680e-03 | 2.368e-01 ±5.751e-03 |
| | ESGG | 8.945e-02 ±5.056e-03 | **0.000e+00** ±0.000e+00 | 2.249e-01 ±1.453e-02 | 7.052e-02 ±5.446e-03 |
| SBM-L | AutoGraph | **2.174e-03** ±2.651e-04 | **2.114e-03** ±1.887e-04 | **2.345e-03** ±6.156e-04 | 1.979e-03 ±2.048e-05 |
| | DiGress | 4.814e-03 ±1.055e-03 | 3.091e-03 ±4.050e-04 | 5.155e-03 ±1.641e-03 | **1.959e-03** ±6.636e-06 |
| | GRAN | 5.168e-02 ±2.770e-03 | 4.556e-02 ±1.382e-03 | 3.685e-02 ±5.084e-03 | 5.610e-03 ±5.368e-04 |
| | ESGG | 3.347e-02 ±2.302e-03 | 5.608e-02 ±2.254e-03 | 8.110e-02 ±8.034e-03 | 4.926e-02 ±8.573e-04 |
| Proteins | AutoGraph | **5.178e-02** ±7.611e-03 | 6.716e-02 ±1.571e-02 | **2.369e-02** ±4.492e-03 | 5.063e-02 ±4.687e-03 |
| | DiGress | 5.236e-02 ±7.499e-03 | **5.146e-02** ±6.231e-03 | 6.808e-02 ±2.714e-02 | **4.360e-02** ±2.098e-04 |
| | GRAN | 3.652e-01 ±8.576e-02 | 2.913e-01 ±3.995e-02 | 4.156e-01 ±8.411e-02 | 7.697e-02 ±1.011e-02 |
| | ESGG | 1.130e-01 ±2.740e-02 | 1.098e-01 ±2.652e-02 | 8.222e-02 ±2.511e-02 | 4.360e-02 ±1.778e-04 |

Table 14: Mapping of display columns in results tables to MMD configurations. For all RBF MMDs, the final MMD was computed as the maximum value over the following bandwidths $\{\sigma_i\}_{i=1}^6 = \{0.1, 0.5, 1.0, 2.0, 5.0, 10.0\}$ as per Thompson et al. (2022). For the descriptor parameters, we used 100,000 for the width of the sparse degree histogram, 100 bins for the clustering histogram, and 4 for the orbit count. RBF: radial basis function; GTV: Gaussian total variation distance; UMVE: unbiased minimum variance estimator, see Gretton et al. (2012).

| Name | Variant | Kernel | | Descriptor |
| | | Name | Parameter | |
|---|---|---|---|---|
| GTV MMD$^2$ Deg. | | | 1.0 | Degree |
| GTV MMD$^2$ Clust. | Biased | GTV | 0.1 | Clustering |
| GTV MMD$^2$ Orb. | | | 30 | Orbit |
| GTV MMD$^2$ Eig. | | | 1.0 | Eigenvalues |
| RBF MMD$^2$ Deg. | | | | Degree |
| RBF MMD$^2$ Clust. | UMVE | RBF | $\{\sigma_i\}_{i=1}^6$ | Clustering |
| RBF MMD$^2$ Orb. | | | | Orbit |
| RBF MMD$^2$ Eig. | | | | Eigenvalues |
| RBF MMD$^2$ Deg. | | | | Degree |
| RBF MMD$^2$ Clust. | Biased | RBF | $\{\sigma_i\}_{i=1}^6$ | Clustering |
| RBF MMD$^2$ Orb. | | | | Orbit |
| RBF MMD$^2$ Eig. | | | | Eigenvalues |

Table 15: Compute time (s) per metric across datasets. Standard deviations are obtained from the metrics computed on different model samples. Caching of intermediate or reused MMD values in `PolyGraph` help make MMD computations substantially faster. Int. indicates whether the metric yields an interval through subsampling. VUN scores were parallelized across 10 CPUs.

| Metric | Int. | PLANAR-L | LOBSTER-L | SBM-L | Proteins | Overall |
|---|---|---|---|---|---|---|
| VUN | ✗ | 425.60 ± 17.72 | 253.32 ± 8.95 | 1181.26 ± 101.98 | - | 620.06 ± 37.98 |
| PGD | ✗ | 73.64 ± 3.01 | 338.82 ± 190.27 | 125.02 ± 17.77 | 140.35 ± 73.67 | 169.46 ± 52.81 |
| PGD | ✓ | 192.13 ± 14.27 | 696.11 ± 367.93 | 280.98 ± 47.45 | 223.67 ± 111.39 | 348.22 ± 118.35 |
| RBF MMD$^2$ Deg. | ✓ | 12.61 ± 0.33 | 12.38 ± 0.14 | 12.72 ± 0.27 | 3.68 ± 0.59 | 10.35 ± 0.23 |
| Biased RBF MMD$^2$ Deg. | ✓ | 10.50 ± 0.21 | 10.32 ± 0.24 | 10.46 ± 0.21 | 1.49 ± 0.65 | 8.19 ± 0.23 |
| GTV MMD$^2$ Deg. | ✓ | 7.74 ± 1.22 | 8.04 ± 0.21 | 8.46 ± 2.54 | 3.26 ± 0.40 | 6.88 ± 0.78 |
| GTV MMD$^2$ Deg. | ✗ | 3.53 ± 0.25 | 3.54 ± 0.32 | 3.83 ± 0.39 | 3.51 ± 0.70 | 3.60 ± 0.21 |
| RBF MMD$^2$ Clust. | ✓ | 16.23 ± 0.46 | 13.69 ± 0.38 | 22.63 ± 1.61 | 16.48 ± 8.16 | 17.26 ± 1.86 |
| Biased RBF MMD$^2$ Clust. | ✓ | 16.60 ± 0.50 | 14.00 ± 1.22 | 25.60 ± 1.86 | 16.73 ± 8.25 | 18.23 ± 2.26 |
| GTV MMD$^2$ Clust. | ✓ | 11.80 ± 1.17 | 10.24 ± 0.13 | 16.75 ± 2.00 | 14.16 ± 8.20 | 13.24 ± 1.93 |
| GTV MMD$^2$ Clust. | ✗ | 7.63 ± 0.06 | 5.54 ± 0.13 | 12.90 ± 2.09 | 14.27 ± 8.35 | 10.08 ± 2.03 |
| RBF MMD$^2$ Orb. | ✓ | 11.87 ± 0.20 | 11.84 ± 0.32 | 14.58 ± 0.62 | 4.84 ± 2.64 | 10.78 ± 0.66 |
| Biased RBF MMD$^2$ Orb. | ✓ | 11.82 ± 0.07 | 11.95 ± 0.36 | 14.64 ± 0.52 | 4.75 ± 2.69 | 10.79 ± 0.70 |
| GTV MMD$^2$ Orb. | ✓ | 5.75 ± 1.08 | 5.85 ± 0.08 | 6.71 ± 1.31 | 3.73 ± 2.13 | 5.51 ± 0.56 |
| GTV MMD$^2$ Orb. | ✗ | 1.64 ± 0.02 | 1.22 ± 0.02 | 2.73 ± 0.41 | 3.71 ± 2.12 | 2.32 ± 0.50 |
| RBF MMD$^2$ Eig. | ✓ | 21.56 ± 0.83 | 19.13 ± 0.71 | 25.83 ± 1.47 | 31.99 ± 16.42 | 24.63 ± 4.14 |
| Biased RBF MMD$^2$ Eig. | ✓ | 25.16 ± 6.52 | 18.75 ± 0.47 | 25.86 ± 1.84 | 33.11 ± 16.31 | 25.72 ± 2.80 |
| GTV MMD$^2$ Eig. | ✓ | 17.85 ± 1.18 | 17.55 ± 0.24 | 20.77 ± 1.83 | 29.67 ± 17.44 | 21.46 ± 4.21 |
| GTV MMD$^2$ Eig. | ✗ | 13.80 ± 0.09 | 12.92 ± 0.16 | 16.88 ± 1.56 | 32.26 ± 19.52 | 18.97 ± 4.82 |

Table 16: Reference values between the test and training set for various metrics.

| Metric | Planar-L | Lobster-L | SBM-L | Proteins |
|---|---|---|---|---|
| **PGD** ($\downarrow$) | $0.2 \pm 0.6$ | $0.0 \pm 0.0$ | $0.4 \pm 0.8$ | $1.6 \pm 2.7$ |
| **Clust.** ($\downarrow$) | $0.5 \pm 0.9$ | $0.0 \pm 0.0$ | $0.7 \pm 1.5$ | $3.4 \pm 4.9$ |
| **Deg.** ($\downarrow$) | $0.3 \pm 0.7$ | $0.4 \pm 0.6$ | $1.2 \pm 1.0$ | $2.4 \pm 3.2$ |
| **GIN** ($\downarrow$) | $0.2 \pm 0.6$ | $0.2 \pm 0.3$ | $0.9 \pm 0.9$ | $3.6 \pm 3.8$ |
| **Orb5.** ($\downarrow$) | $0.3 \pm 0.6$ | $0.2 \pm 0.5$ | $0.4 \pm 0.9$ | $2.2 \pm 4.5$ |
| **Orb4.** ($\downarrow$) | $0.5 \pm 0.8$ | $0.2 \pm 0.4$ | $1.0 \pm 1.2$ | $2.0 \pm 2.6$ |
| **Eig.** ($\downarrow$) | $0.4 \pm 0.7$ | $1.0 \pm 1.4$ | $0.3 \pm 0.8$ | $1.0 \pm 3.1$ |
| **GTV MMD$^2$ Clust.** ($\downarrow$) | 2.91e-04 | 0.00e+00 | 4.87e-04 | 0.0068 |
| **GTV MMD$^2$ Clust.** ($\downarrow$) | 5.87e-04 $\pm$ 1.3e-04 | 0.00e+00 $\pm$ 0.0e+00 | 9.69e-04 $\pm$ 9.4e-06 | 0.0104 $\pm$ 9.4e-04 |
| **RBF MMD$^2$ Clust.** ($\downarrow$) | 3.44e-05 $\pm$ 5.1e-05 | 0.00e+00 $\pm$ 0.0e+00 | 1.62e-06 $\pm$ 3.7e-06 | 0.0014 $\pm$ 0.0016 |
| **RBF MMD$^2$ Clust.** ($\downarrow$) | 5.34e-04 $\pm$ 1.5e-04 | 0.00e+00 $\pm$ 0.0e+00 | 6.10e-04 $\pm$ 2.6e-05 | 0.0077 $\pm$ 0.0020 |
| **GTV MMD$^2$ Deg.** ($\downarrow$) | 1.51e-05 | 1.79e-05 | 1.69e-05 | 3.16e-04 |
| **GTV MMD$^2$ Deg.** ($\downarrow$) | 2.14e-05 $\pm$ 1.1e-05 | 3.06e-05 $\pm$ 1.3e-05 | 3.86e-05 $\pm$ 2.4e-05 | 5.67e-04 $\pm$ 4.6e-04 |
| **RBF MMD$^2$ Deg.** ($\downarrow$) | 1.69e-04 $\pm$ 1.7e-04 | 1.19e-04 $\pm$ 1.2e-04 | 1.48e-04 $\pm$ 1.2e-04 | 0.0052 $\pm$ 0.0038 |
| **RBF MMD$^2$ Deg.** ($\downarrow$) | 6.38e-04 $\pm$ 2.7e-04 | 6.03e-04 $\pm$ 2.0e-04 | 8.54e-04 $\pm$ 1.3e-04 | 0.0117 $\pm$ 0.0039 |
| **GTV MMD$^2$ Orb.** ($\downarrow$) | 3.43e-06 | 1.36e-05 | 3.26e-04 | 0.0032 |
| **GTV MMD$^2$ Orb.** ($\downarrow$) | 2.18e-05 $\pm$ 2.1e-05 | 5.79e-05 $\pm$ 2.8e-05 | 8.79e-04 $\pm$ 2.1e-04 | 0.0065 $\pm$ 0.0042 |
| **RBF MMD$^2$ Orb.** ($\downarrow$) | 1.05e-04 $\pm$ 9.8e-05 | 3.41e-04 $\pm$ 2.8e-04 | 2.98e-05 $\pm$ 3.7e-05 | 0.0044 $\pm$ 0.0055 |
| **RBF MMD$^2$ Orb.** ($\downarrow$) | 0.0010 $\pm$ 3.3e-05 | 0.0012 $\pm$ 2.3e-04 | 9.99e-04 $\pm$ 3.4e-05 | 0.0132 $\pm$ 0.0038 |
| **GTV MMD$^2$ Eig.** ($\downarrow$) | 7.39e-05 | 5.12e-05 | 4.93e-05 | 4.85e-04 |
| **GTV MMD$^2$ Eig.** ($\downarrow$) | 1.27e-04 $\pm$ 2.5e-05 | 1.10e-04 $\pm$ 2.6e-05 | 9.75e-05 $\pm$ 1.9e-05 | 6.97e-04 $\pm$ 1.1e-04 |
| **RBF MMD$^2$ Eig.** ($\downarrow$) | 1.69e-05 $\pm$ 2.9e-05 | 2.78e-05 $\pm$ 4.0e-05 | 5.21e-06 $\pm$ 9.7e-06 | 1.41e-04 $\pm$ 2.1e-04 |
| **RBF MMD$^2$ Eig.** ($\downarrow$) | 5.80e-04 $\pm$ 5.0e-05 | 6.43e-04 $\pm$ 1.0e-04 | 4.02e-04 $\pm$ 3.1e-05 | 0.0024 $\pm$ 2.9e-04 |

Table 17: Reference PGD metrics between the molecule test and training sets. Note that MOSES uses a scaffold split, resulting in a high discrepancy between the train and test set.

| Dataset | PGD subscores | | | | | |
|---|---|---|---|---|---|---|
| | **PGD** ($\downarrow$) | **Topo** ($\downarrow$) | **Morgan** ($\downarrow$) | **ChemNet** ($\downarrow$) | **MolCLR** ($\downarrow$) | **Lipinski** ($\downarrow$) |
| Guacamol | $0.2 \pm 0.4$ | $0.2 \pm 0.4$ | $0.3 \pm 0.5$ | $0.3 \pm 0.6$ | $0.1 \pm 0.2$ | $0.0 \pm 0.0$ |
| Moses | $21.0 \pm 0.6$ | $21.0 \pm 0.6$ | $17.8 \pm 0.7$ | $16.0 \pm 1.2$ | $18.0 \pm 0.8$ | $20.7 \pm 0.7$ |

## L    STABILITY OF PGD UNDER VARYING SAMPLE SIZES.

Figs. 23 to 26 show the relationship between the PGD score and the number of samples. The PGD score of the reference graphs with respect to another set of reference graphs issued from the same distribution is given as a comparison. For all experiments, we show the mean as well as the 5th and 95th quantile to give an estimate of the variance of PGD at different sample sizes.

For most models, some separation from the test set occurs above 256 samples, with PGD scores, and especially the upper bound is mostly stable beyond this range. This both showcases the stability of the metric, the number of samples required to get a reliable PGD estimate, as well as the overall PGD ranges for the various models we considered for this study.

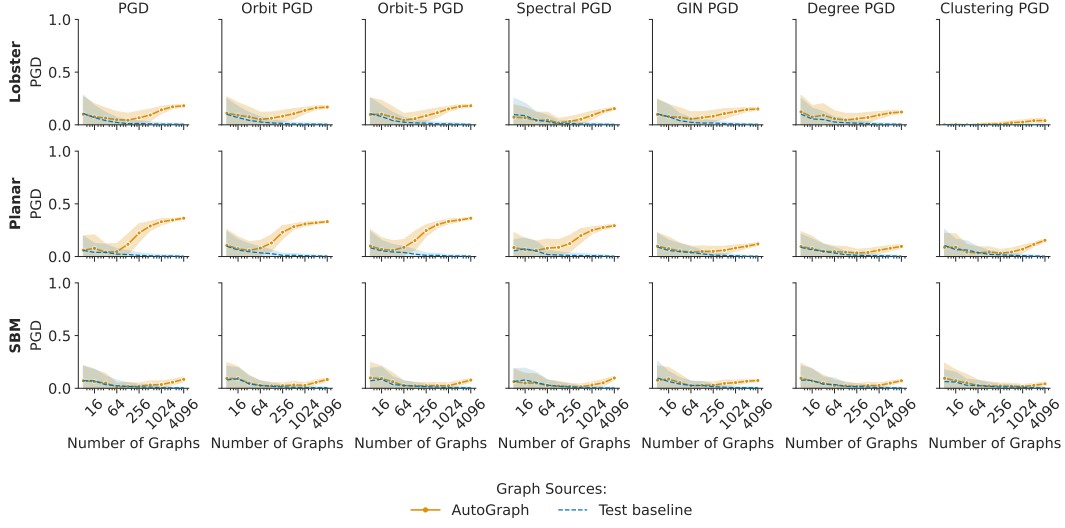

Figure 23: PGD obtained from varying sample sizes generated by AutoGraph.

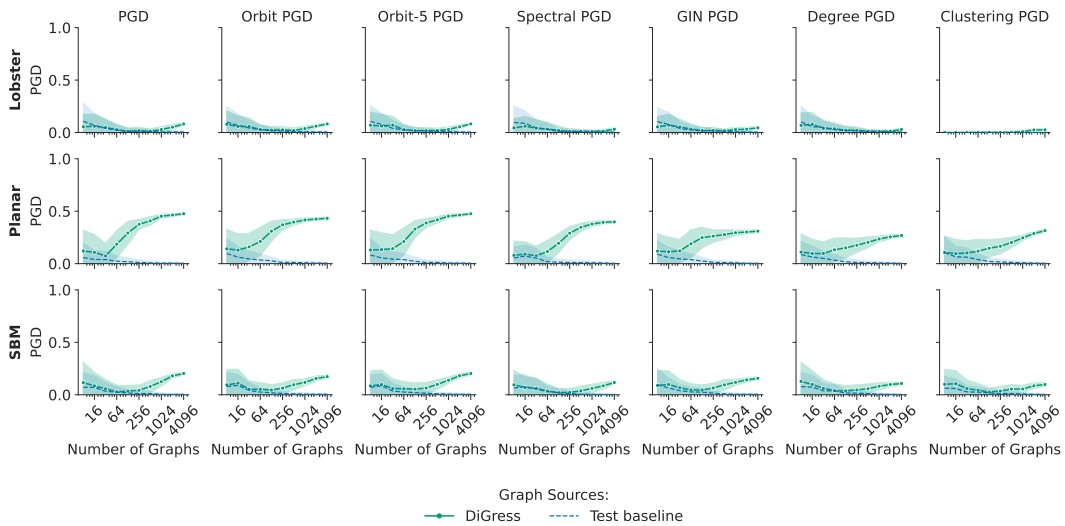

Figure 24: PGD obtained from varying sample sizes generated by DiGress.

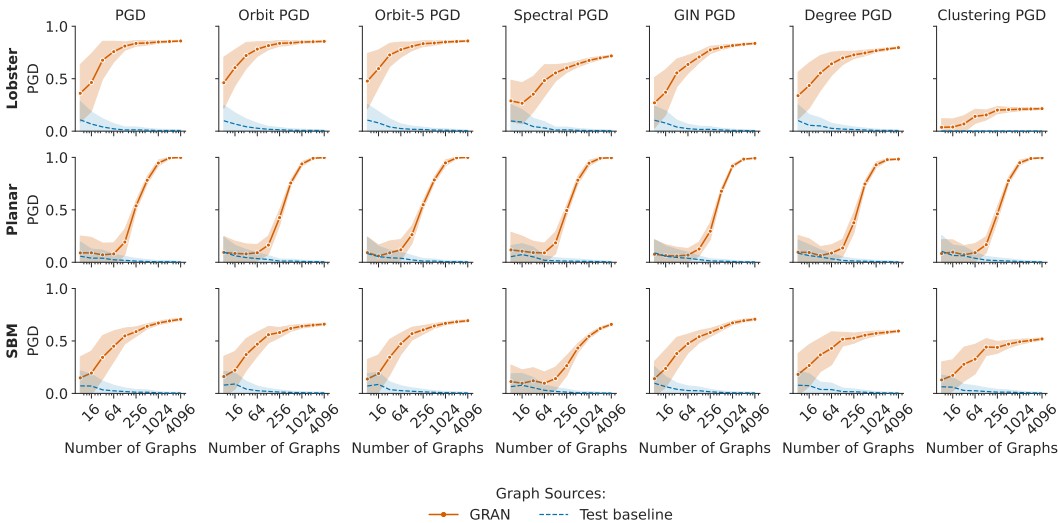

Figure 25: PGD obtained from varying sample sizes generated by GRAN.

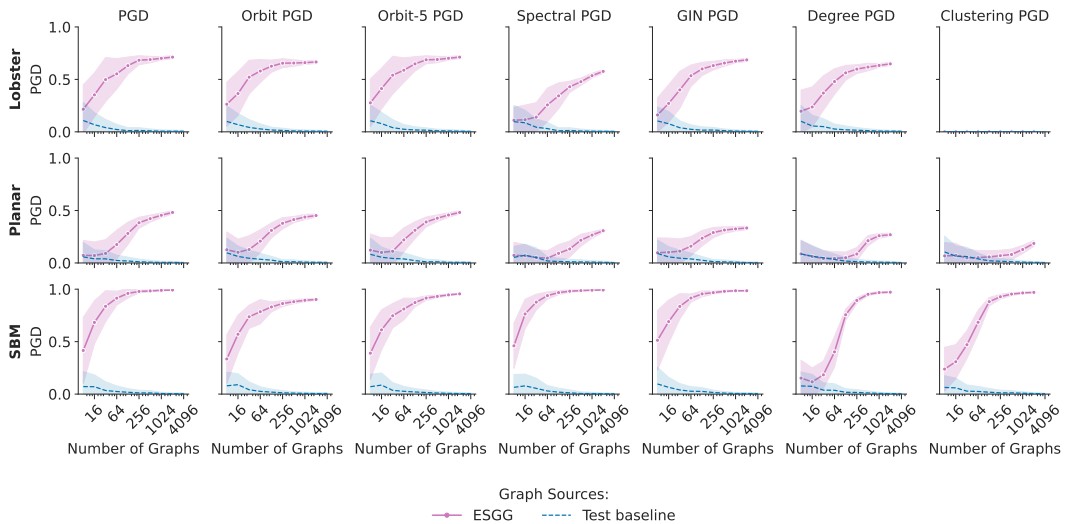

Figure 26: PGD obtained from varying sample sizes generated by ESGG.

## M    LARGER PROCEDURAL REFERENCE DATASETS FOR BETTER GGM BENCHMARKING

Following our findings of Section 5.1 and Appendix G, we introduce larger procedurally-generated datasets for planar, lobster and SBM graphs, which we term PLANAR-L, LOBSTER-L and SBM-L. LOBSTER-L is a set of tree-shaped lobster graphs generated using `nx.random_lobster`, controlled by expected node count (80) and attachment probabilities to the backbone and its neighbors (set to 0.7 for both). PLANAR-L is a set of connected planar graphs generated by uniformly sampling 64 node positions in the unit square and forming the Delaunay triangulation via `scipy.spatial.Delaunay`, yielding planar edge sets from triangle simplices. SBM-L is a set of stochastic block model graphs with the number of communities sampled uniformly from 2 to 5 and nodes per community from 20 to 40, where edges are drawn with intra-community probability 0.3 and inter-community probability 0.005. SBM-L, PLANAR-L, and LOBSTER-L datasets follow `networkx`'s BSD-3 license.

Table 18: Dataset sizes (number of graphs) per split.

| Dataset | Train | Val | Test |
|---|---|---|---|
| SBM-L | 8192 | 4096 | 4096 |
| PLANAR-L | 8192 | 4096 | 4096 |
| LOBSTER-L | 8192 | 4096 | 4096 |

## N    INFLUENCE OF TRAINING SET SIZE ON AUTOGRAPH

As shown in Fig. 27, AutoGraph converges to similar VUN values across datasets, yet the loss is substantially lower for SBM-L after training than for SBM-S. This finding indicates that models may overfit on the existing small procedural datasets, further drawing into question the validity of previously reported evaluation results (Vignac et al., 2023).

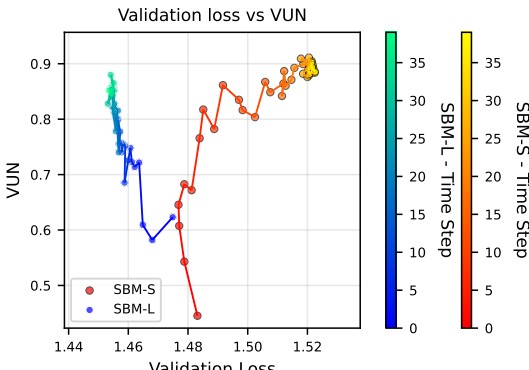

Figure 27: VUN vs. Loss for AutoGraph over the course of a training run.

## O    COMMON SHORTFALLS OF EXISTING SOLUTIONS

To address the lack of inherent scale in MMD, some have proposed normalizing the MMD between generated and test graphs by the MMD between train and test graphs (Martinkus et al., 2022). However, this approach has several shortcomings:

**Limited theoretical justification**  MMD was originally introduced as a kernel two-sample test. Its manipulation beyond direct use as a performance metric or for $p$-value computation remains poorly understood.

**Lack of composability**  The MMD ratio does not enable combining information across multiple descriptors.

**Sample-size sensitivity** As shown in Appendix G, MMD strongly depends on sample size. Dividing MMDs computed on different sample sizes produces ratios with unclear or unreliable interpretation.

## P    DATASET DETAILS

Here, we provide details about the datasets used in this study. Licenses for those datasets are summarized in Table 19. Table 20 shows the dataset statistics of the Citeseer dataset (Sen et al., 2008). The statistics for the small procedural datasets are presented in Table 21 (Planar), Table 22 (SBM), and Table 23 (Lobster).

Table 19: License and author information of the datasets used in our experiments.

| Dataset | Author | License |
|---|---|---|
| Citeseer | (Sen et al., 2008) | CC BY-NC-SA 3.0 |
| Procedural (Planar, SBM, Lobster) | (Martinkus et al., 2022; Hagberg et al., 2008) | BSD-3 |
| Proteins | (Dobson & Doig, 2003) | CC0 1.0 Universal |

Table 20: Ego dataset statistics (extracted from Citeseer).

| Metric | Train | Val | Test |
|---|---|---|---|
| Number of Graphs | 454 | 151 | 152 |
| Minimum number of Nodes | 50 | 50 | 50 |
| Maximum number of Nodes | 399 | 333 | 364 |
| Average number of Nodes | 141.72 | 139.29 | 158.08 |
| Minimum number of Edges | 64 | 56 | 63 |
| Maximum number of Edges | 1066 | 898 | 1004 |
| Average number of Edges | 325.16 | 321.87 | 369.30 |
| Edge/Node Ratio | 2.29 | 2.31 | 2.34 |

Table 21: Dataset statistics for the Planar dataset (train, validation, and test splits).

| Metric | Train | Validation | Test |
|---|---|---|---|
| Number of Graphs | 128 | 32 | 40 |
| Minimum Number of Nodes | 64 | 64 | 64 |
| Maximum Number of Nodes | 64 | 64 | 64 |
| Average Number of Nodes | 64.00 | 64.00 | 64.00 |
| Minimum Number of Edges | 173 | 174 | 174 |
| Maximum Number of Edges | 181 | 181 | 181 |
| Average Number of Edges | 177.83 | 177.75 | 177.93 |
| Edge-to-Node Ratio | 2.78 | 2.78 | 2.78 |

## Q    FEATURE CONCATENATION AS AN ALTERNATIVE TO MAX-REDUCTION

An alternative to taking the maximum JSD across descriptors consists of obtaining an overall PGD score by concatenating all vectors arising from the different descriptors, and training a discriminator atop these concatenated features. This makes attributing potentially high values to specific descriptors impossible, but in practice still results in a tighter bound (i.e., higher scores) as can be seen in Table 24.

Table 22: Dataset statistics for the SBM dataset (train, validation, and test splits).

| Metric | Train | Validation | Test |
|---|---|---|---|
| Number of Graphs | 128 | 32 | 40 |
| Minimum Number of Nodes | 44 | 49 | 54 |
| Maximum Number of Nodes | 187 | 162 | 174 |
| Average Number of Nodes | 105.99 | 91.28 | 107.85 |
| Minimum Number of Edges | 129 | 183 | 210 |
| Maximum Number of Edges | 1129 | 857 | 972 |
| Average Number of Edges | 512.51 | 425.19 | 521.88 |
| Edge-to-Node Ratio | 4.84 | 4.66 | 4.84 |

Table 23: Dataset statistics for the Lobster dataset (train, validation, and test splits).

| Metric | Train | Validation | Test |
|---|---|---|---|
| Number of Graphs | 60 | 20 | 20 |
| Minimum Number of Nodes | 10 | 11 | 14 |
| Maximum Number of Nodes | 98 | 98 | 84 |
| Average Number of Nodes | 53.67 | 56.30 | 50.80 |
| Minimum Number of Edges | 9 | 10 | 13 |
| Maximum Number of Edges | 97 | 97 | 83 |
| Average Number of Edges | 52.67 | 55.30 | 49.80 |
| Edge-to-Node Ratio | 0.98 | 0.98 | 0.98 |

## R    KERNEL LOGISTIC REGRESSION WITH GRAPH KERNELS

One can adopt the kernel logistic regression classifier and use graph kernels directly to evaluate GGMs, effectively showing that any (graph) kernel also suitable for MMD can also be used in PGD. We showcase this with the Weisfeiler-Lehman (Shervashidze et al., 2011), shortest path (Borgwardt & Kriegel, 2005), and PyramidMatch (Grauman & Darrell, 2007) graph kernels in Table 25. However, they almost always show looser bounds compared to the standard PGD formulation, so we do not favor such kernels.

## S    USE OF LARGE LANGUAGE MODELS

The authors used large language models in the following ways:

**Intelligent tab completion** During software development, tools for intelligent line-wise tab completion were used.

**Preparation of visualizations** LLMs were partly used to generate code for figure layouts. The correctness of all code and data was checked manually. The data shown in the figures was generated by manually written code.

**Information retrieval** LLMs were queried for related work, but produced no relevant results. All related work presented in the manuscript was manually retrieved, save for Endres & Schindelin (2003), which was manually checked to contain the required proof.

**Polishing of manuscript** LLMs were occasionally used to refine or rephrase individual sentences.

Table 24: Comparison of VUN, max-reduced PGD (the default we also use in Table 4) and PGD with concatenated descriptors. PGD-Concat. is obtained by concatenating all descriptor features and training a single discriminator on the combined representation. The final score is obtained similarly to PGD.

| Dataset | Model | VUN (↑) | PGD (↓) | PGD-Concat. (↓) |
|---|---|---|---|---|
| PLANAR-L | AutoGraph | 85.7 | **34.1** ± 1.7 | **46.2** ± 2.1 |
| | DIGRESS | 79.6 | 46.7 ± 1.7 | 55.7 ± 1.9 |
| | GRAN | 3.1 | 99.5 ± 0.5 | 99.3 ± 1.1 |
| | ESGG | **93.9** | 47.7 ± 1.1 | 53.7 ± 1.6 |
| LOBSTER-L | AutoGraph | 82.6 | 16.6 ± 1.4 | **34.7** ± 1.4 |
| | DIGRESS | **91.8** | **4.0** ± 3.1 | 47.6 ± 1.0 |
| | GRAN | 42.9 | 86.0 ± 0.7 | 86.3 ± 0.5 |
| | ESGG | 70.9 | 70.6 ± 0.6 | 71.0 ± 1.1 |
| SBM-L | AutoGraph | **87.1** | **7.1** ± 1.4 | **29.1** ± 1.6 |
| | DIGRESS | 74.1 | 16.6 ± 1.9 | 33.2 ± 2.5 |
| | GRAN | 22.3 | 69.1 ± 1.1 | 78.1 ± 1.1 |
| | ESGG | 11.0 | 99.3 ± 0.3 | 98.2 ± 0.5 |
| Proteins | AutoGraph | - | **57.3** ± 11.5 | **81.6** ± 5.7 |
| | DIGRESS | - | 86.7 ± 2.4 | 97.7 ± 0.9 |
| | GRAN | - | 88.9 ± 5.5 | 98.7 ± 0.8 |
| | ESGG | - | 78.8 ± 5.0 | 98.8 ± 0.3 |

Table 25: Comparison of PGD (as shown in Table 4) with a PGD variant with a graph kernel logistic regression (GKLR) model as the classifier. The kernels used here are the PyramidMatch (PM) kernel, the shortest-path (SP) kernel, and the Weisfeiler-Lehman (WL) kernel.

| Dataset | Model | | | Subscores | | |
|---|---|---|---|---|---|---|
| | | PGD (↓) | PGD-GKLR (↓) | PM (↓) | SP (↓) | WL (↓) |
| PLANAR-L | AutoGraph | **33.5** ± 1.3 | **6.6** ± 1.1 | 5.8 ± 0.7 | 5.3 ± 0.8 | 7.3 ± 1.0 |
| | DIGRESS | 45.4 ± 1.8 | 22.0 ± 1.3 | 19.1 ± 0.9 | 22.7 ± 0.8 | 20.4 ± 1.0 |
| | GRAN | 99.3 ± 1.1 | 43.1 ± 0.4 | 8.1 ± 0.7 | 5.5 ± 1.6 | 43.1 ± 0.4 |
| | ESGG | 45.0 ± 1.4 | 14.4 ± 1.0 | **2.7** ± 2.3 | 12.8 ± 0.7 | 14.6 ± 0.8 |
| LOBSTER-L | AutoGraph | 16.4 ± 1.7 | 8.8 ± 2.1 | 9.5 ± 1.1 | 6.2 ± 2.7 | 8.3 ± 1.8 |
| | DIGRESS | **3.9** ± 3.2 | **2.0** ± 2.3 | **2.8** ± 2.3 | **1.5** ± 2.2 | **0.3** ± 1.0 |
| | GRAN | 85.7 ± 0.7 | 72.5 ± 0.8 | 52.4 ± 0.7 | 57.5 ± 1.3 | 72.5 ± 0.8 |
| | ESGG | 69.9 ± 0.6 | 56.1 ± 0.6 | 42.0 ± 0.6 | 41.8 ± 1.0 | 56.1 ± 0.6 |
| SBM-L | AutoGraph | **5.8** ± 1.2 | **2.3** ± 2.6 | **1.5** ± 1.6 | **3.0** ± 2.4 | **0.5** ± 1.3 |
| | DIGRESS | 16.7 ± 1.7 | 8.7 ± 2.5 | 8.0 ± 1.8 | 4.4 ± 2.6 | 9.1 ± 2.4 |
| | GRAN | 68.8 ± 1.1 | 46.7 ± 1.2 | 45.9 ± 1.2 | 32.4 ± 1.1 | 46.7 ± 1.2 |
| | ESGG | 99.4 ± 0.2 | 93.5 ± 0.3 | 23.8 ± 1.8 | 93.5 ± 0.3 | 42.6 ± 1.1 |
| Proteins | AutoGraph | **55.3** ± 12.0 | 38.2 ± 3.4 | 13.2 ± 5.2 | 38.2 ± 3.4 | 14.1 ± 5.7 |
| | DIGRESS | 86.8 ± 2.9 | 38.6 ± 2.4 | **1.2** ± 1.9 | 38.6 ± 2.4 | **3.5** ± 3.9 |
| | GRAN | 86.1 ± 7.2 | 53.6 ± 3.1 | 48.5 ± 2.9 | 40.5 ± 3.5 | 53.6 ± 3.1 |
| | ESGG | 77.8 ± 5.1 | **27.7** ± 4.9 | 11.8 ± 2.4 | **29.5** ± 3.1 | 16.1 ± 2.7 |

