# OpenReview forum: "PolyGraph Discrepancy: a classifier-based metric for graph generation"
_ICLR.cc/2026/Conference — ICLR 2026 Poster_

### Official Review · Reviewer_VKfC · 2025-11-01

**Soundness:** 3
**Presentation:** 3
**Contribution:** 2
**Rating:** 4
**Confidence:** 4

**Summary:**

This work introduces PGS: a method for evaluating graph generative models. The idea behind the paper is to use the JS divergence to measure the similarity between the true graph distribution (reference)  and that of the generated distribution. Similar to the literature employing MMD instead of using the graph distribution it uses the distribution of certain descriptors derived from statistics of the graph (like degrees, orbit counts, etc.) The benefit of using JSD over MMD is that the former is bounded. JSD has been used in GANs as the objective function. Hence, this work approximates JSD using a binary classifier that tries to distinguish the original graph from the generated ones. To take into account multiple predictors it takes the maximum divergence among all descriptors as the divergence between the reference and generated graphs.

**Strengths:**

* The proposed method for evaluating GGMs addresses some of the limitations of MMD: Critically, its unboundedness and the lack of a way to combine predictors
* It introduces a new axis of evaluating how suitable a given measure (or metric) is: How predictable it is with respect to the training steps of well-known models. Intuitively, the distance should decrease as the training steps increase.

**Weaknesses:**

* On the other hand, it follows the existing paradigm of MMD -- using a set of predictors to approximate the graph distance with the distance of predictors, without offering a new angle that would revitalize the literature in GGM evaluation.
* The proposed way to aggregate multiple predictors (by taking the maximum distance) is potentially susceptible to outliers & does not take complementary signals from predictors into account.
* The scale issue is remediated by using a bounded in [0,1] distance. However, certain predictors  (or classifiers applied to those predictors) might be more sensitive to perturbations than others, hence the max aggregation scheme could favor them.

**Questions:**

Q1: Have the authors considered other aggregations than max, or was it some certain predictor that shows higher sensitivity to perturbations and thus most frequently dominates?
Q2: How fast is the training of TabPFN?
Q3: Taking into account that for real graphs (as opposed to synethetic ones) the number of reference graphs could be limited, do the authors see a limitation in a method that also requires to split the dataset into train/test sets and hence a small number of graphs could be available for either training or testing? How many samples does usually TabPFN require?

---

> ### Author Response · Authors · 2025-11-21
> **Response to Reviewer VKfC**
>
> We thank you for your thoughtful and constructive feedback. Please find below our response to each of your points.
>
> > On the parallels with the MMD paradigm and the lack of a new angle that would revitalize the literature in GGM evaluation
>
> While PGS and MMD share the same goal, namely distinguishing between graph distributions, the methodological foundations are fundamentally different, and this difference has substantial practical consequences.
>
> **MMD is a kernel-based distance with no probabilistic interpretation.** It provides a value whose magnitude depends on the kernel choice, preventing meaningful comparison across descriptors. In contrast, PGS provides a variational lower bound on a well-defined information-theoretic quantity: the Jensen-Shannon divergence. This is not merely a different computational approach to the same problem; it establishes a principled connection between classifier performance and distributional distance that MMD lacks entirely.
>
> This theoretical grounding yields three concrete advantages that directly address long-standing pain points in GGM evaluation:
> 1. **Absolute interpretability**: A PGS of 0.05 means something on its own: the distributions are nearly indistinguishable to an optimal classifier. An MMD of 0.05 means nothing without context.
> 2. **Cross-descriptor comparability**: Because each PGS subscore bounds the same quantity (JSD), we can rigorously compare and aggregate scores across descriptors via max-reduction. MMD offers no such guarantee -- comparing MMD values across different kernels is impossible.
> 3. **Diagnostic power**: The descriptor achieving the maximum PGS identifies which structural aspect distinguishes the distributions, directly informing model development. MMD provides no such actionable feedback.
>
> Beyond the metric itself, our work also makes two additional contributions to revitalize the field:
> (i) we systematically expose how bias and variance in existing benchmarks have led to unreliable model comparisons (Section 5.1), providing concrete remedies
> (ii) we release PolyGraph, an extensible library with tested implementations that lowers the barrier for rigorous evaluation.
>
> We believe replacing an ad-hoc distance measure with a theoretically grounded, interpretable, and comparable metric constitutes a meaningful methodological advance.
>
> Regarding **how PGS addresses inherent limitation of MMDs**, we kindly refer the reviewer to our general comment "Addressing the concern on the motivation and expressivity of our proposed metric". We also made changes to the manuscript to clarify this motivation.
>
> > The proposed way to aggregate multiple predictors [...] is potentially susceptible to outliers & does not take complementary signals from predictors into account.
>
> **Outliers have minimal impact in our setting because we are comparing distributions of graphs**, and given sufficient sample sizes, the influence of such outliers is minimal. We detailed how many samples are required to reach such low variance estimates in Section 5.1 and Appendix L. Regarding the complementarity of graph descriptors, we address it in our general comment, but in short: there is an inherent tradeoff between the interpretability of the score and the tightness of the bound. We now explore this tradeoff in Table 24 of Appendix Q.
>
> > Certain descriptors might be more sensitive to perturbations than others, hence the max aggregation scheme could favor them.
>
> We agree that certain descriptors may be more sensitive to perturbations and consequently yield higher PGS subscores than others. These descriptors then dominate the max-aggregation. This is the intended behavior:
>
> We recall that PGS is designed to estimate a tight lower bound on the intrinsic Jensen Shannon distance between two graph distributions. Each PGS sub-score associated to a descriptor provides a lower bound on this distance. Descriptors that are especially expressive (e.g., sensitive to perturbations) will produce the most discriminative features for TabPFN and hence the tightest bound on the JS distance (that is, the highest PGS subscores). The max-aggregation scheme *automatically selects these expressive descriptors while discarding less informative ones*.
>
> The max-reduction technique hence allows PGS to be invariant to the addition of uninformative features while automatically selecting the most expressive ones. This is certainly a desirable property of evaluation metrics and one that is not satisfied by MMD.
>
> We highlight again that max-aggregation provides a principled way to select the most discriminative descriptors. **Only through max-aggregation can PGS be rigorously interpreted as a maximally tight bound on the Jensen Shannon distance between graph distributions** given a set of descriptors. We also refer to our general comment which further discusses the relationship between the expressivity of the descriptors and the tightness of the bound.

---

> > ### Author Response · Authors · 2025-11-21
> > **Response to Reviewer VKfC - continued**
> >
> > > Have the authors considered other aggregations than max[-reduction?]
> >
> > No, as max-aggregation is the only valid option to obtain a maximally tight bound given the descriptors. To consider potential descriptor complementarity, we did run an experiment concatenating features. Those results are presented in Table 4 in Appendix Q. We also discuss the tradeoffs associated with this alternative aggregation strategy in our general comment.
> >
> > > How fast is the training of TabPFN?
> >
> > Table 15 in Appendix K referenced in section 5.4 contains all relevant timings of PGS (including the TabPFN training), VUN and MMD. Note that these timings include the training of multiple TabPFN models required for cross-validation. In practice, computing the equivalent MMDs for a typical dataset takes 1-2 minutes while for PGS it takes 2-5 minutes. For this paper, we computed thousands of different versions of PGS, a testament to its ease of use and light computational burden.
> >
> > > Do the authors see a limitation in a method that also requires to split the dataset into train/test sets and hence a small number of graphs could be available for either training or testing?
> >
> > As long as the sample size yields confidence intervals obtained through subsampling are reported, we see no issue. On the real graph datasets reported in this paper, splitting is not an issue and confidence intervals are low. However, we do encourage that practitioners do compute those variance estimates to ensure fair model comparisons.
> >
> > > How many samples does usually TabPFN require?
> >
> > The number of samples required to get reasonable PGS estimates vary depending on the dataset. Figures 23-26 in Appendix L answer exactly this question. As a general rule of thumb, 256 samples are required. But in general, as with any metric, the more samples are considered, the more reliable the estimate. Again, we encourage practitioners to evaluate the variance of the metric when developing models to assess the reliability of the estimate, and consider more samples if required.

---

> > > ### Author Response · Authors · 2025-11-28
> > > **Kind reminder**
> > >
> > > Dear reviewer VKfC,
> > >
> > > We would like to remind you that we provided a detailed response to your review and would sincerely appreciate a response as the discussion period is ending very soon. If you have any other specific concerns or questions, we are happy to answer them. Thank you again for your time.
> > >
> > > Kind regards,
> > >
> > > The authors.

---

### Official Review · Reviewer_3AL7 · 2025-11-01

**Soundness:** 3
**Presentation:** 3
**Contribution:** 3
**Rating:** 8
**Confidence:** 3

**Summary:**

This paper introduces PGS, an alternative to MMD for measuring the dissimilarity between sets of graphs. The introduced measure is based on the Jensen-Shannon divergence. The authors make use of the property that this divergence is bounded by the best possible binary classifier between these distributions. To obtain an estimate of this bound, they train classifiers based on graph representations and estimate the JS divergence.

The authors perform several experiments that highlight defects of MMD and which demonstrate that their JS divergence overcomes these defects.

**Strengths:**

The information-theoretical setup of the JS divergence is very elegant.

While the idea is simple, executing it properly requires some amount of engineering. From what I can see, the authors did a good job at this.

Their approach finds the graph descriptor that is best at discriminating the two graph samples (among some selection of descriptors). It is nice that this gives insight into the aspect in which the samples differ.

The paper is well written and I enjoyed reading it.

**Weaknesses:**

From the paper, it is not entirely clear to me how efficiently PGS can be computed. I'm assuming that it is considerably heavier than computing MMDs based on the same descriptors. In line 231, it says that it is based on TabPFN, which is "fast", though it does not elaborate on this claim.

There's several popular and interesting graph kernels that are missing from the comparison. Like the Weisfeiler-Lehman (WL) kernel, the PyramidMatch (PM, [1]) kernel and the Shortest-Paths (SP) kernel. The SP kernel, in particular, seems to fit the descriptor framework well and it was shown to be very powerful in distinguishing graphs from different distributions [2].

The experiments are somewhat hard to follow and some of the figures are somewhat messy. For example, Figure 3 could be replaced by box-plots for clarity and Figure 5 is quite dense.

The authors mention that MMD lacks an intrinsic scale, but most kernels satisfy some Cauchy-Schwarz-like bound $k(x,y)\le\sqrt{k(x,x)k(y,y)}$, which can be used to normalize the kernels like $k(x,y)/\sqrt{k(x,x)k(y,y)}$. After doing so, the MMD is in the interval [0,2].

Table 1 describes "Multi-Descriptor Aggregation" as one of the advantages of PGS over MMD. I can't find where in the text this is explained. MMD can also be used to combine different descriptors. For example, we can concatenate the descriptors (before computing kernels) or take the mean of several kernels. I do agree that PGS utilizes multiple descriptors more efficiently than MMD, but I would call this "selection" rather than "aggregation".

In the introduction, the authors characterize MMD as a way to compare graph samples using descriptors. I always interpreted MMD as a way to aggregate graph kernels to a distance between sets of graphs. These kernels are usually based on comparing graph descriptors, but this is not a necessity. For example, for the PM or WL kernel, it is cumbersome to think of them in terms of descriptors.

In Appendix M, I see that you generate planar graphs from Delauney triangulations. Note that this does not lead to a uniform sample among planar graph. To sample uniformly from this sample, one could use Boltzmann samplers [3]. I found this implementation that seems to do this https://github.com/towink/boltzmann-planar-graph (disclaimer: I did not test this implementation). Moreover, it appears that in the SBM generation, the connection probabilities are kept constant while the community sizes and number of communities are varied. Perhaps it is better to change these probabilities with the sizes to ensure each vertex has a constant (expected) number of neighbors inside and outside its communities. With the current probabilities, the SBM with $2\times20$ nodes will only have have two inter-community edges in expectation.

[1] Borgwardt, Karsten M., and Hans-Peter Kriegel. "Shortest-path kernels on graphs." *Fifth IEEE international conference on data mining (ICDM'05)*. IEEE, 2005.

[2] Gösgens, Martijn, Alexey Tikhonov, and Liudmila Prokhorenkova. "Evaluating Graph Generative Models with Graph Kernels: What Structural Characteristics Are Captured?." *Transactions on Machine Learning Research*.

[3] Fusy, Éric. "Uniform random sampling of planar graphs in linear time." *Random Structures & Algorithms* 35.4 (2009): 464-522.

**Questions:**

Can you comment on the running time and scalability of PGS?

PGS uses descriptors as input rather than kernels. Could it be modified so that it works with graph kernels? I could imagine that some k-NN classifier could work, where we take nearest neighbors in the training set and use the validation set to estimate performance. That is, for a graph $G$ in the validation set, we find the $k$ graphs in the training set that best resemble it according to the kernel, and then estimate $D(G)$ as the fraction of these training graphs of each class.

Section 5.2 shows experiments with "Data perturbations" akin to those from O'Bray and Thompson. Instead of perturbing a given distribution, we could interpolate between pairs of graph distributions, as is done in [2]. In particular, it would be interesting to see how well PGS performs on interpolations that are quite subtle, like between geometric graphs with 1D and 2D latent geometries.

Table 2 uses Pearson correlation rather than Spearman correlation, which is used in Figure 3. Why are we interested in *linear* rather than *rank* correlation in the "Trianing Iterations" experiment?

If I understand Figure 2 correctly, we evaluate the MMD between two graph samples from the same distribution. This experiment is used to demonstrate the bias of MMD. However, since the true MMD should be zero and MMD is always nonnegative, it seems to me that any estimate with positive variance would be biased. Am I missing something here?

---

> ### Author Response · Authors · 2025-11-21
> **Response to Reviewer 3AL7**
>
> We thank you for your thoughtful and constructive feedback. Please find below our response to each of your points.
>
> > From the paper, it is not entirely clear to me how efficiently PGS can be computed.
>
> Detailed timings for PGS computation compared to MMD are provided in Table 15 in Appendix K referenced in section 5.4. We also make a reference to this table at l.231. While having a higher computational complexity than MMD (typically 2-5 minutes for PGS on 5 descriptors vs 1-2 minutes for MMD on those 5 descriptors), training TabPFN on a specific dataset is relatively fast and leverages hardware acceleration, keeping computation time low. In our perturbation experiments we have run thousands of PGS computations without any substantial computational burden, further supporting the relatively low compute burden of the metric. Typically, practitioners only evaluate their model a handful of times. Consequently, this overhead is not a substantial limitation.
>
> > Comparison to graph kernels
>
> The shortest-path, Weisfeiler-Lehmann and PyramidMatch kernels are not used in any of the MMD parametrizations to evaluate graph generative models. Yet, we would like to note that our framework can easily accomodate graph kernels by using a kernel logistic regression classifier. **We show examples with the shortest-path, Weisfeiler-Lehman and PyramidMatch graph kernels in a new experiment in Appendix R.** Overall, we saw looser bounds with graph kernel logistic regression, and in keeping with our findings that TabPFN has higher correlation coefficients with respect to perturbations, we ommit graph kernels in our final PGS formulation.
>
> > Clarity of Figures 3 and 5.
>
> In Figure 3, information (especially about outliers) would be lost if we moved from a swarm plot to a box plot. In Figure 5, we want to show all 5 MMDs. Due to space constraints, this inherently becomes crowded, but shows the noisiness and lack of scale of MMD.
>
> > MMD boundedness with a Cauchy-Schwarz-like bound
>
> In the context of graph generation, the literature adopted RBF and Gaussian TV (pseudo)-kernels. As a result, we have $k(x,x) = 1$ and $k(y,y) = 1$. Hence, the normalization would have no effect and the limitations we have described in our paper remain.
>
> Furthermore, the fact that MMDs are in the same interval $[0, 2]$ does not mean that they are directly comparable. In addition to comparability, we are not aware of a rigorous argument that the max-reduction we have proposed for PGS would be applicable to normalized MMD values.
>
> > Table 1 describes "Multi-Descriptor Aggregation" [...] MMD can also be used to combine different descriptors. For example, we can concatenate the descriptors (before computing kernels) or take the mean of several kernels. I would call this "selection" rather than "aggregation".
>
> The MMD metrics obtained from concatenating descriptors or taking the mean of kernels has, to the best of our knowledge, no rigorous interpretation. In contrast, the max-reduction we perform for PGS can be rigorously understood as tightening a lower bound on the Jensen Shannon distance.
>
> We agree that "selection" is a better term than "aggregation" and we updated Table 1 (l. 58-64) to refer to our max-reduction procedure as "descriptor selection", as opposed to "multi-descriptor aggregation".
>
> > In the introduction, the authors characterize MMD as a way to compare graph samples using descriptors. [...] kernels are usually based on comparing graph descriptors, but this is not a necessity.
>
> In the context of graph generation, all commonly used kernels are constructed via featurization with graph descriptors. Hence, in this specific context, this formulation is the most practical and maps to our implementation. While our method could be introduced in a broader context, we found positioning our work within the specific field of graph generation is more approachable and clear.
>
> > Planar and SBM graph construction
>
> We thank the reviewer for this careful analysis and the concrete suggestions, including the Boltzmann sampler reference for uniform planar graph generation.
> The reviewer is correct that Delaunay triangulations do not yield uniformly sampled planar graphs, and that our SBM parameterization produces sparse inter-community connectivity at larger scales. We adopted these constructions from prior work [1] to ensure comparability with existing benchmarks, but we acknowledge they represent specific instantiations of these graph families.
> Importantly, our core claims do not depend on uniformity of the planar distribution or specific SBM density regimes. PGS measures distributional divergence regardless of the reference distribution's construction. Nevertheless, we agree that exploring alternative constructions (e.g., Boltzmann samplers, degree-corrected SBMs) would strengthen future benchmarks.
> We have also clarified our discussion of dataset-related limitations to Appendix A and welcome suggested datasets in future versions of the `PolyGraph` library.

---

> > ### Author Response · Authors · 2025-11-21
> > **Response to Reviewer 3AL7 - continued**
> >
> > > Can you comment on the running time and scalability of PGS?
> >
> > We refer to our previous answer where we point to the empirical runtime of PGS.
> >
> > Asymptotically, the runtime complexity is quadratic in the number of graph samples, due to the self-attention mechanisms within TabPFN. This is the same asymptotic complexity as the standard MMD estimate.
> >
> > For more practical considerations we refer to Appendix A: TabPFN v2, which we use in our work, may be evaluated on up to 10 000 samples with up to 500 features. The newly released TabPFN v2.5 model supports up to 50 000 samples with up to 2000 features.
> >
> > > PGS uses descriptors as input rather than kernels. Could it be modified so that it works with graph kernels? I could imagine that some k-NN classifier could work, where we take nearest neighbors in the training set and use the validation set to estimate performance.
> >
> > One could compute a PGS with graph kernels. However, since a k-NN classifier does not optimize a data log-likelihood criterion, it does not explicitly tighten a lower bound on the Jensen Shannon divergence. Hence, the resulting PGS may be a looser and less principled bound. We can instead use kernel logistic regression, and use graph kernels with this classifier. **We have conducted those experiments and added them to Appendix R**. Overall, we saw looser bounds with graph kernel logistic regression, and in keeping with our findings that TabPFN has higher correlation coefficients with respect to perturbations (see Appendix J), we ommit graph kernels in our final PGS formulation.
> >
> > > Section 5.2 shows experiments with "Data perturbations" akin to those from O'Bray and Thompson. Instead of perturbing a given distribution, we could interpolate between pairs of graph distributions, as is done in [2].
> >
> > Our extensive experiments following O'Bray and Thompson have been established in the literature and reused by many others, e.g. [5, 6]. Our current perturbation regimes are also very similar to the proposed scheme, as our fully perturbed graph distribution can be viewed as a distribution in itself.
> >
> > > Table 2 uses Pearson correlation rather than Spearman correlation, which is used in Figure 3. Why are we interested in linear rather than rank correlation in the "Training Iterations" experiment?
> >
> > In Table 2, we explicitly want to show the correlation between validity and other evaluation metrics (MMD and PGS). We compute Pearson correlations to demonstrate that PGS exhibits a linear relationship *with validity* while MMD metrics do not. We note that *we perform these experiments for both the experiment on training iterations and denoising iterations*.
> >
> > In Table 3, we investigate how PGS, validity and MMD correlate with training duration. However, we typically expect the model performance to plateau towards the end of the training process. Hence, we don't expect a linear relationship and compute Spearman correlation coefficients. A similar relationship is expected between the amount of perturbation and the metrics, hence we also show the Spearman correlation coefficient in Figure 3.
> >
> > We clarified this in the manuscript.
> >
> > > If I understand Figure 2 correctly, we evaluate the MMD between two graph samples from the same distribution. This experiment is used to demonstrate the bias of MMD. However, since the true MMD should be zero and MMD is always nonnegative, it seems to me that any estimate with positive variance would be biased.
> >
> > While the MMD is always non-negative, the unbiased minimum variance MMD *estimator* can be negative (see eq. 3 on p.729 of [3]). It is true that it is necessary to have negative values to obtain an unbiased estimator despite its positive variance. We hope that this answers your question.
> >
> > Additionally, we want to use this opportunity to clarify our usage of the term "unbiased MMD" in the context of RBF metrics: we recall that these metrics are obtained by maximizing MMD estimates over RBF bandwidth parameters (see [2, 4]). The metrics we refer to as *unbiased RBF MMD* are obtained by maximizing unbiased MMD estimators over bandwidth parameters. However, we note that this maximum is not necessarily unbiased. We have updated our manuscript to clarify this nuance.
> >
> >
> > [1]: Martinkus et al. SPECTRE: spectral conditioning helps to overcome the expressivity limits of one-shot graph generators. ICML 2022.
> >
> > [2]: Fukumizu et al. Kernel Choice and Classifiability for RKHS Embeddings of Probability Distributions. NeurIPS 2009.
> >
> > [3]: Gretton et al. A Kernel Two-Sample Tests. JMLR 2012.
> >
> > [4]: Thompson et al. On Evaluation Metrics for Graph Generative Models. ICLR 2022.
> >
> > [5]: Chen et al. Efficient and Degree-Guided Graph Generation via Discrete Diffusion Modeling. ICML 2023.
> >
> > [6]: Huang et al. GraphGDP: Generative Diffusion Processes for Permutation Invariant Graph Generation. ICDM 2022.
> >
> > [7]: Vignac et al. DiGress: Discrete Denoising diffusion for graph generation. ICLR 2023.

---

> > > ### Comment · Reviewer_3AL7 · 2025-11-24
> > >
> > > I thank the authors for their thoughtful response. My rating remains unchanged.

---

### Official Review · Reviewer_bnvC · 2025-11-01

**Soundness:** 3
**Presentation:** 3
**Contribution:** 3
**Rating:** 6
**Confidence:** 3

**Summary:**

- The paper proposes PolyGraphScore (PGS), a new metric for evaluating graph generative models by measuring how well a classifier can distinguish real from generated graphs.
- PGS estimates the Jensen–Shannon divergence between the two distributions using probabilistic classification instead of kernel-based distances like MMD.
- It uses graph descriptors (e.g., degree, spectrum, GIN embeddings) as tabular features and applies a TabPFN classifier to compute an interpretable score between 0 and 1.
- Experiments show that PGS provides stable, bounded, and interpretable evaluations that correlate strongly with model quality and training progress, outperforming traditional MMD metrics.

**Strengths:**

- The paper introduces a principled, classifier-based metric that provides an interpretable and bounded measure of distance between real and generated graphs.
- The method is model-agnostic and descriptor-agnostic, allowing fair comparison across graph generative models without retraining or fine-tuning.
- It demonstrates strong empirical correlations with true model quality and training progress, showing that the score behaves predictably and robustly.

**Weaknesses:**

- The choice of TabPFN as the discriminator is only weakly justified, with limited empirical evidence beyond a single logistic-regression ablation.
- The method's dependence on precomputed graph descriptors introduces information loss and may underrepresent complex graph structures.
- The paper lacks error bars or confidence intervals, which limits the reader's ability to assess the statistical reliability and variability of the reported results.

**Questions:**

- Could you elaborate on why you chose the Jensen–Shannon divergence as the foundation for PolyGraphScore, instead of other f-divergences (e.g., KL, Hellinger)?
- Does the PolyGraphScore framework extend naturally to weighted, directed, or attributed graphs, or are there fundamental challenges?
- The paper uses TabPFN as the discriminator; could you provide more detailed empirical evidence comparing it to other probabilistic classifiers (e.g., random forest, XGBoost, small MLP)?
- Since TabPFN supports only limited feature dimensions, how would PGS scale to very high-dimensional descriptors (e.g., graph embeddings >1000 dimensions)?

**Details Of Ethics Concerns:**

There is no particular ethical concern.

---

> ### Author Response · Authors · 2025-11-21
> **Response to Reviewer bnvC**
>
> We thank you for your thoughtful and constructive feedback. Please find below our response to each of your points.
>
> > The choice of TabPFN as the discriminator is only weakly justified, with limited empirical evidence beyond a single logistic-regression ablation.
>
> We kindly refer the reviewer to our general comment "Addressing the concern on the choice of the discriminator".
>
> > The method's dependence on precomputed graph descriptors introduces information loss and may underrepresent complex graph structures.
>
> We want to first clarify that **using graph descriptors is standard practice for evaluating generative models** (see GraphRNN, GRAN, DiGress, etc.), and our work is grounded in this established setting. We kindly point the reviewer to our general comment about clarification on the motivation of our work.
>
> While it is true that the model does depend on fixed-length descriptors of graphs, the fact that they can be combined provably improves the expressive power of the metric (Section 4.2). As long as those descriptors are complementary and cover as many aspects of the graph's complexity as possible, the reliance on precomputed descriptors is not problematic. We note that existing MMD metrics suffer from the same limitation, yet descriptors cannot be combined in this case (see Section 3.1).
>
> >The paper lacks error bars or confidence intervals, which limits the reader's ability to assess the statistical reliability and variability of the reported results.
>
> We want to emphasize that **our paper extensively explores issues related to variance and discusses how it historically impeded effective generative modeling on graphs** (see section 5.1, 5.4, Appendices G, H, J, K, L, M). In this context we endeavored to report the full variability of MMD and different PGS formulations across different subsamples and datasets in all relevant figures and tables both in the main manuscript and in the appendix. That said, Table 2 and 3 do indeed lack variance estimates, but this is not feasible as it requires the training of ~10 different models at each parameter setting, resulting in over 70 graph generative models  that would need to be retrained (e.g., DiGress requires 48-95h to train for a single model). **Importantly, for all benchmarking experiments, standard deviations obtained using different subsamples from the same model are provided in Table 4.**
>
> > Could you elaborate on why you chose the Jensen–Shannon divergence as the foundation for PolyGraphScore, instead of other f-divergences (e.g., KL, Hellinger)?
>
> As discussed in Section 3.2, the Jensen–Shannon (JS) divergence arises naturally from the theoretical connection between _classifier log-likelihood_ and _distributional distance_. Specifically, the log-likelihood of any probabilistic classifier trained to discriminate between real and generated samples yields a valid lower bound on the JS divergence, and this bound becomes tighter as the classifier is optimized via maximum likelihood. This property makes JS uniquely compatible with our classifier-based formulation and provides a direct, principled route to constructing a metric.
>
> We also explored an alternative f-divergence, the **total variation (TV) distance**, in Appendix E, which is closely related to the JS divergence and provides additional perspective on our construction.
>
> Regarding other f-divergences:
>
> -   Hellinger divergence **does not admit a comparable connection to classifier log-likelihood,** meaning we cannot obtain a tight lower bound through optimization in the same way we can with JS.
> -   KL divergence is **asymmetric** and **unbounded**, both of which are undesirable for an evaluation metric intended for model comparison. Moreover, KL is ill-defined when the support of the two distributions do not overlap, an issue that routinely arises in generative modeling.
>
> > Does the PolyGraphScore framework extend naturally to weighted, directed, or attributed graphs, or are there fundamental challenges?
>
> **Yes, it does naturally extend to attributed graphs.** For example, we show in Table 4 that molecules, which have node and edge attributes, can be easily processed by using suitable descriptors. PGS also naturally extends to directed graphs if suitable descriptors exist.
>
> > The paper uses TabPFN as the discriminator; could you provide more detailed empirical evidence comparing it to other probabilistic classifiers (e.g., random forest, XGBoost, small MLP)?
>
> We kindly refer the reviewer to our general comment "Addressing the concern on the choice of the discriminator".

---

> > ### Author Response · Authors · 2025-11-21
> > **Response to Reviewer bnvC - continued**
> >
> > > Since TabPFN supports only limited feature dimensions, how would PGS scale to very high-dimensional descriptors (e.g., graph embeddings >1000 dimensions)?
> >
> > This is indeed a limitation we discuss in Appendix A. In Appendix C2, as an example, we apply random projections to obtain features that fit within the limits of TabPFN. In Appendix Q, we used PCA to project the concatenated, high dimensional feature to a 500-dimensional feature to compute the Jensen Shannon distance.
> >
> > However, we want to make clear that this is a mere implementation detail, and [recent updates to TabPFN](https://priorlabs.ai/technical-reports/tabpfn-2-5-model-report) indicate that the supported number of features has expanded to 2,000. We expect such model families to improve over time and mitigate this limitation.

---

> > > ### Comment · Reviewer_bnvC · 2025-11-25
> > >
> > > Thank you for the authors' clarification.
> > >
> > > I acknowledge I have read a rebuttal and other reviews.
> > >
> > > I would maintain my score.

---

> > > > ### Author Response · Authors · 2025-11-27
> > > >
> > > > Thank you for your response. Let us know if you have further questions.

---

### Official Review · Reviewer_VfPn · 2025-11-02

**Soundness:** 3
**Presentation:** 3
**Contribution:** 3
**Rating:** 6
**Confidence:** 4

**Summary:**

This paper proposes PolyGraphScore (PGS), a novel evaluation framework for graph generative models (GGMs) that addresses key weaknesses of the current standard Maximum Mean Discrepancy (MMD)-based metrics. Instead of computing kernel distances between graph descriptors, PGS trains a probabilistic discriminator (TabPFN) to distinguish real and generated graphs, using standard descriptors such as degree histograms, Laplacian spectra, or GIN embeddings. The classifier’s log-likelihood provides a variational lower bound on the Jensen–Shannon (JS) distance, yielding a unit-scaled score in [0, 1] that is interpretable and comparable across descriptors.
Comprehensive experiments demonstrate that PGS tracks synthetic perturbations, correlates strongly with model validity and training progress, and provides consistent model rankings across datasets and descriptors.

**Strengths:**

The paper is well written and easy to follow. The problem statement is clearly defined and well-motivated. The proposed evaluation framework is both theoretically sound and empirically well validated, demonstrating strong performance across diverse experiments.

**Weaknesses:**

1. Although TabPFN is relatively fast, training separate discriminators for each descriptor introduces additional computational overhead compared to a single MMD computation.

2. Regarding empirical validation, while the paper provides benchmarks on both real and synthetic datasets, expanding the experiments to include more specialized synthetic datasets could further demonstrate the generality of the proposed approach. Specifically, [1] notes that for Grid datasets — which consist of graphs with highly regular local structures — correlations between evaluation metrics are not close to 1, unlike for other datasets. Similarly, [2] reports different performance patterns of GGMs when generating grid-like structures. Comparing the proposed evaluation method with more expressive approaches, such as [1], on datasets where neither statistic-based MMD methods nor GNN-based methods clearly dominate, could further highlight the potential advantages of PGS.

3. The stability analysis of PGS under varying sample sizes (Figures 23–25) indicates that, similar to MMD-based approaches, PGS suffers from relatively high variance, particularly for graph sets containing fewer than 256 samples.


[1]. Shirzad, Hamed, Kaveh Hassani, and Danica J. Sutherland. "Evaluating graph generative models with contrastively learned features." NeurIPS 2022.

[2]. Zahirnia, Kiarash, et al. "Neural graph generation from graph statistics." NeurIPS 2023.

**Questions:**

How sensitive is PolyGraphScore (PGS) to the choice of graph descriptors? Have the authors explored combining multiple descriptors instead of selecting the best-performing one through max-reduction?

Could the authors clarify how PGS behaves when using different classifier architectures beyond TabPFN (e.g., GNNs), and whether this affects the tightness of the JS lower bound?

---

> ### Author Response · Authors · 2025-11-21
> **Response to Reviewer VfPn**
>
> We thank you for your thoughtful and constructive feedback. Please find our response to each of your points below.
>
> > Training separate discriminators for each descriptor introduces additional computational overhead compared to a single MMD computation.
>
> In practice, people typically compute multiple MMDs separately for each descriptor (e.g., GraphRNN, DiGress, etc.), which is similar to our setting.
> While computing each individual subscore is indeed more expensive than a single MMD computation, both remain practical--each completing in under a minute. Estimating PGS across five descriptors takes 2–5 minutes (due to cross-validation for descriptor selection), compared to 1–2 minutes for five separate MMD estimates; full results are in Table 15 (Appendix K). Our perturbation experiments, which required thousands of PGS computations, demonstrate this scalability. In practice, practitioners typically evaluate their models only a handful of times and already compute multiple MMDs separately for each descriptor, so we do not consider the modest overhead a substantial limitation given the advantages PGS provides.
>
> > Suggested experiments on additional synthetic datasets like grid graphs
>
> We investigated the Grid graph datasets in [1,3]. They were first introduced in the GraphRNN paper [2], and we found that the authors use, by default, grids with a minimum width and height of 20 and a maximum width and height of 30, generating 121 possible unique grids before randomly splitting them into train, validation and test sets. As we show in Section 5.1, having low sample sizes yields high-variance MMD estimates which make model comparisons statistically unreliable. Moreover, we note that our planar graph dataset also forms highly regular local structures while offering more diversity, which should be a more challenging dataset than a grid-only dataset.
>
>
> > Suggested experiment with more expressive approaches such as contrastively learned GNNs as suggested in [1].
>
> We refer the reviewer to our general comment about the relationship of our metrics to the expressivity of descriptors. We designed our library to be easily extensible to accommodate any customized graph descriptors. Additionally, we already apply such a contrastively learned descriptor (MolCLR, [4]) for our molecular benchmark, showing the flexibility of PGS and its various parametrizations.
>
> > Stability of PGS under varying sample sizes
>
> While PGS exhibits similar instability to MMD at very small sample sizes (until ~256, see Appendix L, Figures 23-26), we emphasize that this behavior reflects a **fundamental challenge of evaluating generative models with limited data. We view this result not as a weakness but as a contribution:** our work systematically identifies and characterizes this phenomenon for graph generative models, making its implications explicit. By quantifying the variance across subsamples, we hope to prevent misleading conclusions that may arise from under-sampled evaluations and **provide practical recommendations** to the community on how to assess and mitigate this issue in practice.
>
> > Q1. How sensitive is PolyGraphScore (PGS) to the choice of graph descriptors?
>
> PGS is sensitive to the choice of graph descriptors, because each descriptor contributes a lower bound on the true Jensen–Shannon distance between the underlying graph distributions. If an informative or highly expressive descriptor is omitted, the resulting bound may become looser, leading to a less faithful approximation of the true divergence. In practice, we recommend users to always use multiple descriptors if it is not trivial to prove the superiority of one to the others.
>
> > Q2. Have the authors explored combining multiple descriptors instead of selecting the best-performing one through max-reduction?
>
> Thank you for the suggestion. We conducted this experiment, and had to perform dimensionality reduction (with PCA) to ensure the features remained expressive while fitting in the TabPFN feature limit. Those results are in table 24 in the updated manuscript and in our general comment.
>
> While combining features through concatenation and dimensionality reduction does yield a tighter bound (higher scores are observed for all datasets and models), our max-reduction provides additional interpretability and we kindly refer the reviewer to our general comment (e.g. on the Lobster graph dataset, a non-zero clustering PGS subscore indicates the presence of triangles).
>
> > Q3. Could the authors clarify how PGS behaves when using different classifier architectures beyond TabPFN (e.g., GNNs), and whether this affects the tightness of the JS lower bound?
>
> We kindly refer the reviewer to our general comment "Addressing the concern on the choice of the discriminator". Note that we have a randomly initialized GNN descriptor [5] and another contrastively trained GNN (MolCLR) [4] for the molecular graph dataset.

---

> > ### Author Response · Authors · 2025-11-21
> > **References**
> >
> > [1]: Zahirnia, Kiarash, et al. "Neural graph generation from graph statistics." NeurIPS 2023.
> >
> > [2]: Jiaxuan You, Rex Ying, Xiang Ren, William L. Hamilton, Jure Leskovec. "GraphRNN: Generating Realistic Graphs with Deep Auto-regressive Models". ICML 2018.
> >
> > [3]: Shirzad, Hamed, Kaveh Hassani, and Danica J. Sutherland. "Evaluating graph generative models with contrastively learned features." NeurIPS 2022.
> >
> > [4]: Wang et al. Molecular contrastive learning of representations via graph neural networks. Nature Machine Intelligence. 2022.
> >
> > [5]: Thompson et al. On Evaluation Metrics for Graph Generative Models. ICLR 2022.

---

> > > ### Author Response · Authors · 2025-11-28
> > > **Kind reminder**
> > >
> > > Dear reviewer VfPn,
> > >
> > > We would like to remind you that we provided a detailed response to your review and would sincerely appreciate a response as the discussion period is ending very soon. If you have any other specific concerns or questions, we are happy to answer them. Thank you again for your time.
> > >
> > > Kind regards,
> > >
> > > The authors.

---

### Author Response · Authors · 2025-11-21
**General comment to all reviewers**

We thank all reviewers for their thoughtful and constructive feedback.

First, we appreciate the positive assessments highlighting key strengths of our work, including:

- A principled, theoretically grounded metric (bnvC and 3AL7)
- A clear analysis of inherent limitations of MMD (VKfC and VfPn)
- Empirical strength and correlation with model quality (measured with correlations to training and denoising steps) (bnvC and VKfC)
- Descriptor-wise interpretability and insight (3AL7 and VfPn)
- Clarity and readability (VfPn and 3AL7)

All changes we made to the manuscript are highlighted in blue. In summary, we made the following changes:

Methodological additions:
- Added Appendix Q: feature concatenation as alternative to max-reduction (referred to in Sec. 5)
- Added Appendix R: kernel logistic regression with graph kernels (referred to in Sec. 4.1)
- Noted that kernel logistic regression fits naturally into PGS framework (Sec. 4.1)

Clarifications:
- Emphasized that MMD's lack of inherent scale prevents assessing discriminative power (Sec. 2)
- Further clarified our discriminator choice (Sec. 4.1)
- Clarified use of Spearman's vs Pearson's correlations throughout experiments (Sec. 5.3)
- Specified procedural datasets use "specific parameters" (Appendix A)

---

> ### Author Response · Authors · 2025-11-21
> **Addressing the concern on the motivation and expressivity of our proposed metric (reviewer VKfC and bnvC)**
>
> We want to clarify that **using graph descriptors is standard practice for evaluating generative models**, and our work is grounded in this established setting. We note that these descriptors can be either feature vectors describing graph characteristics, such as orbit counts, spectral embeddings, or graph kernels. Traditionally, MMD has been used to construct metrics from these descriptors, yet it suffers from several inherent limitations (see more details in our Introduction and Table 1):
> 1. **Unboundedness**, making it difficult to understand the absolute goodness of fit of a generative model (e.g., it's hard to tell whether an MMD of 0.1 is good or not without comparison);
> 2. **No straightforward way of combining multiple descriptors**, since MMD is not comparable across descriptors, making model comparison difficult when using multiple descriptors;
> 3. **Limited interpretability**, as one needs to interpret each MMD in isolation for each descriptor, and can yield potentially contradictory model rankings.
>
> Our contribution is **not to introduce more expressive graph descriptors**, but to provide a principled alternative to MMD that directly **addresses the limitations of MMD we list above**.
> Our proposed PolyGraph Score (PGS) is bounded by design, and critically, provides a provable lower bound on the Jensen-Shannon distance (JSD) between graph distributions under the standard assumption that the chosen descriptors are sufficiently expressive. And the key property is that **more expressive descriptors yield tighter bounds**, a desirable and intuitive behavior: as the descriptors capture more aspects of the underlying graphs, the PGS better approximates the true JSD. This property makes PGS natively address the comparability and interpretability issues noted above.

---

> > ### Author Response · Authors · 2025-11-21
> > **Addressing the concern on the aggregation method (reviewer VKfC, VfPn and 3AL7)**
> >
> > Reviewers noted that one could concatenate descriptors and compute a single score to obtain an even tighter bound. While theoretically possible, this approach **eliminates descriptor-level interpretability**, which is essential for diagnosing failure modes and guiding generative model development.
> > In practice, knowing _which_ descriptor detects a distribution shift is far more actionable than receiving a single aggregated value.
> >
> > Our design, **a max-based reduction over descriptor-specific PGS subscores**, strikes a deliberate balance:
> >
> > - It preserves **interpretability**, by showing the contribution of each descriptor;
> > - It maintains a **meaningful theoretical connection to JSD**, as each subscore provides a valid lower bound;
> >
> > Therefore, our formulation represents an appropriate and practically useful middle ground between **tightness of the JSD approximation** and **descriptor-wise interpretability**, which multiple reviewers recognized as a key strength.
> >
> > To make our study more comprehensive, we provide the results for concatenating features below and in Appendix Q:
> >
> > **Table: VUN, max-reduced PGS and PGS-Concat. The PGS-Concat. metric is obtained by concatenating the descriptor vectors, then applying dimensionality reduction (with PCA) to the resulting representation to ensure it fits in TabPFN's recommended limits (for v2.0, it is 500), and finally computing a single PGS value from the reduced-dimensional space.**
> > | **Dataset** | **Model** | **VUN (↑)** | **PGS (↓)** | **PGS-Concat. (↓)** |
> > |-------------|-----------|-------------|-------------|---------------------|
> > | **Planar-L** | AutoGraph | *85.1* | **34.0 ± 1.8** | **44.8 ± 1.3** |
> > | | **DiGress** | 80.1 | 45.2 ± 1.8 | 55.3 ± 1.5 |
> > | | GRAN | 1.6 | 99.7 ± 0.2 | 99.4 ± 0.2 |
> > | | ESGG | **93.9** | *45.0 ± 1.4* | *52.4 ± 1.1* |
> > | **Lobster-L** | AutoGraph | *83.1* | *18.0 ± 1.6* | **29.0 ± 2.1** |
> > | | **DiGress** | **91.4** | **3.2 ± 2.6** | *43.2 ± 1.4* |
> > | | GRAN | 41.3 | 85.4 ± 0.5 | 86.4 ± 0.9 |
> > | | ESGG | 70.9 | 69.9 ± 0.6 | 69.9 ± 1.0 |
> > | **SBM-L** | AutoGraph | **85.6** | **5.6 ± 1.5** | **27.2 ± 3.0** |
> > | | **DiGress** | *72.8* | *17.4 ± 2.3* | *32.0 ± 2.0* |
> > | | GRAN | 21.4 | 69.1 ± 1.4 | 78.0 ± 0.8 |
> > | | ESGG | 10.6 | 99.4 ± 0.2 | 98.1 ± 0.4 |
> > | Proteins | AutoGraph | - | **67.7 ± 7.4** | **94.8 ± 2.6** |
> > | | **DiGress** | - | 88.1 ± 3.1 | 99.6 ± 0.3 |
> > | | GRAN | - | 89.7 ± 2.7 | 99.8 ± 0.1 |
> > | | ESGG | - | *79.2 ± 4.3* | *99.4 ± 0.3* |

---

> > > ### Author Response · Authors · 2025-11-21
> > > **Addressing the concern on the choice of the discriminator (reviewer VfPn, bnvC and 3AL7)**
> > >
> > > We choose TabPFN because it satisfies the specific requirements needed based on theoretical and practical considerations (see **Section 4.1**):
> > > - **Probabilistic** outputs to compute the JS distance
> > > - **High efficiency** across datasets and evaluation settings
> > > - **Hyperparameter-free** to ensure fair and reproducible comparisons.
> > >
> > > Classifiers that are sensitive to hyper-parameters would require substantial and cumbersome tuning by practitioners, preventing fair comparisons. This rules out tree-based models and neural networks (e.g., GNNs) trained with stochastic gradient descent. TabPFN is an adequate middle ground balancing practicality and expressivity.
> > >
> > > We compared TabPFN with another classifier that also meets those criteria, namely logistic regression (see Section J of the appendix). We also conducted a new experiment that includes **kernel logistic regression** directly on the graph kernels (WL, shortest path, and PyramidMatch) with the examples proposed by reviewer 3AL7 which can be found below and in Appendix R.
> > >
> > > **Table: PGS with a PGS variant with a graph kernel logistic regression (GKLR) model as the classifier. The kernels used here are the PyramidMatch (PM) kernel, shortest-path (SP) kernel, and Weisfeiler-Lehman (WL) kernel.**
> > >
> > > | **Dataset** | **Model** | **PGS (↓)** | **PGS-GKLR (↓)** | **Subscore - PM (↓)** | **Subscore - SP (↓)** | **Subscore - WL (↓)** |
> > > |-------------|-----------|-------------|------------------|------------|------------|------------|
> > > | **Planar-L** | AutoGraph | **34.0 ± 1.8** | **6.2 ± 2.1** | *5.3 ± 1.4* | **5.2 ± 0.9** | **6.7 ± 1.9** |
> > > | | **DiGress** | 45.2 ± 1.8 | 22.7 ± 0.9 | 19.3 ± 0.5 | 22.8 ± 0.6 | 20.5 ± 0.6 |
> > > | | GRAN | 99.7 ± 0.2 | 43.1 ± 0.3 | 8.8 ± 0.8 | *5.2 ± 2.4* | 43.1 ± 0.3 |
> > > | | ESGG | *45.0 ± 1.4* | *14.4 ± 1.0* | **2.7 ± 2.3** | 12.8 ± 0.7 | *14.6 ± 0.8* |
> > > | **Lobster-L** | AutoGraph | *18.0 ± 1.6* | *10.6 ± 1.2* | *10.3 ± 0.9* | *8.4 ± 1.4* | *10.5 ± 1.7* |
> > > | | **DiGress** | **3.2 ± 2.6** | **2.4 ± 2.5** | **2.6 ± 1.7** | **2.5 ± 2.2** | **2.2 ± 2.4** |
> > > | | GRAN | 85.4 ± 0.5 | 72.7 ± 0.8 | 52.3 ± 0.8 | 57.9 ± 1.2 | 72.7 ± 0.8 |
> > > | | ESGG | 69.9 ± 0.6 | 56.1 ± 0.6 | 42.0 ± 0.6 | 41.8 ± 1.0 | 56.1 ± 0.6 |
> > > | **SBM-L** | AutoGraph | **5.6 ± 1.5** | **5.7 ± 1.1** | **1.4 ± 1.5** | *5.7 ± 1.1* | **1.3 ± 2.0** |
> > > | | **DiGress** | *17.4 ± 2.3* | *8.8 ± 2.4* | *7.8 ± 2.4* | **4.0 ± 2.2** | *9.0 ± 2.5* |
> > > | | GRAN | 69.1 ± 1.4 | 47.4 ± 1.0 | 46.8 ± 1.0 | 32.7 ± 1.3 | 47.4 ± 1.0 |
> > > | | ESGG | 99.4 ± 0.2 | 93.5 ± 0.3 | 23.8 ± 1.8 | 93.5 ± 0.3 | 42.6 ± 1.1 |
> > > | Proteins | AutoGraph | **67.7 ± 7.4** | *39.2 ± 2.8* | *14.0 ± 2.5* | *39.2 ± 2.8* | *16.5 ± 2.1* |
> > > | | **DiGress** | 88.1 ± 3.1 | 44.8 ± 1.3 | **3.6 ± 3.0** | 44.8 ± 1.3 | **8.9 ± 3.3** |
> > > | | GRAN | 89.7 ± 2.7 | 59.4 ± 2.0 | 55.0 ± 1.8 | 45.7 ± 1.9 | 59.4 ± 2.0 |
> > > | | ESGG | *79.2 ± 4.3* | **31.9 ± 5.0** | 17.7 ± 2.3 | **31.9 ± 5.0** | 22.0 ± 2.1 |
> > >
> > > As in the paper, **Bold** indicates best performance, *italic* indicates second best performance (lower is better). Overall, all three kernels yield informative but almost always looser bounds on the JSD than the TabPFN-based PGS estimates. Consequently, this does not change our choice of descriptor or overall PGS parameterization choice for the main results.
> > >
> > > We address individual concerns as a response to specific reviewers.

---

### Meta-Review · Area_Chair_UXUB · 2026-01-05

**Summary:**

The paper provide a framework to evaluate graph generative models, from a spirit of classifier two sample test (C2ST), that extend from and subsume MMD-based approaches. The classifier is to be used to distinguish the generated samples from the observation; and constructed based on the (symmetric) Jensen-Shannon divergence The paper is noted to be well-written by the reviewers and related works are addressed. Additional experimental findings are provided and updated text includes concerns raised during the discussion period.

**Reviewer Concerns:**

In general, the response shows vibrant discussion and additional results and clarifications have been added to both the manuscript and the chat. Specifically,

- Sensitivity of PGS (addressed) for the high variance setting based on the data variance

- The PGS sensitive to the graph descriptor (addressed)

- Comparison of graph kernels (addressed) by adding new results in the appendix

- aggregation of multiple graphs (partially addressed) clarified the various way of aggregation while it might be still open questions for the assessment of graph samples, e.g. single large graph or aggregating block features in SBM example; the description of single graph fit and test could be further clarified.

**Reviewer Scores:**

The reviewer concerns are generally acknowledged. It seems that Reviewer VKfC's concern is also addressed and the score may increase if there is the full participation opportunity.

---

### Decision · Program_Chairs · 2026-01-26

Accept (Poster)